 **eLIFE**

# A map of human PRDM9 binding provides evidence for novel behaviors of PRDM9 and other zinc-finger proteins in meiosis

Nicolas Altemose[1,2†*], Nudrat Noor[1‡], Emmanuelle Bitoun[1], Afidalina Tumian[2§], Michael Imbeault[3#], J Ross Chapman[1], A Radu Aricescu[1¶], Simon R Myers[1,2*]

[1]The Wellcome Trust Centre for Human Genetics, University of Oxford, Oxford, United Kingdom; [2]Department of Statistics, University of Oxford, Oxford, United Kingdom; [3]Global Health Institute, École Polytechnique Fédérale de Lausanne, Lausanne, Switzerland

**\*For correspondence:**
altemose@berkeley.edu (NA);
myers@stats.ox.ac.uk (SRM)

**Present address:** †Department of Bioengineering, University of California, Berkeley, Berkeley, United States; ‡TH Chan School of PublicHealth, Harvard University, Cambridge, United States; §Department of Computer Science, International Islamic University Malaysia, Lumpur, Malaysia; #Department of Genetics, University Of Cambridge, Cambridge, United Kingdom; ¶MRC Laboratory of Molecular Biology, Cambridge, United Kingdom

**Competing interests:** The authors declare that no competing interests exist.

**Abstract** PRDM9 binding localizes almost all meiotic recombination sites in humans and mice. However, most PRDM9-bound loci do not become recombination hotspots. To explore factors that affect binding and subsequent recombination outcomes, we mapped human PRDM9 binding sites in a transfected human cell line and measured PRDM9-induced histone modifications. These data reveal varied DNA-binding modalities of PRDM9. We also find that human PRDM9 frequently binds promoters, despite their low recombination rates, and it can activate expression of a small number of genes including *CTCFL* and *VCX*. Furthermore, we identify specific sequence motifs that predict consistent, localized meiotic recombination suppression around a subset of PRDM9 binding sites. These motifs strongly associate with KRAB-ZNF protein binding, TRIM28 recruitment, and specific histone modifications. Finally, we demonstrate that, in addition to binding DNA, PRDM9's zinc fingers also mediate its multimerization, and we show that a pair of highly diverged alleles preferentially form homo-multimers.
DOI: https://doi.org/10.7554/eLife.28383.001

## Introduction

In humans and mice, PRDM9 determines the locations of meiotic recombination hotspots (*Baudat et al., 2010*; *Myers et al., 2010*; *Parvanov et al., 2010*). PRDM9 is expressed early in meiotic prophase (*Sun et al., 2015*), during which its C2H2 Zinc-Finger (ZF) domain binds DNA at particular motifs and its PR/SET domain trimethylates surrounding histone H3 proteins at lysine 4 (H3K4me3; *Hayashi et al., 2005*) and at lysine 36 (H3K36me3; *Wu et al., 2013*; *Eram et al., 2014*; *Powers et al., 2016*; *Davies et al., 2016*; *Grey et al., 2017*; *Yamada et al., 2017*). At a subset of PRDM9 binding sites, SPO11 is recruited to form Double Strand Breaks (DSBs) (*Neale and Keeney, 2006*; *Smagulova et al., 2011*). These DSBs undergo end resection and the resulting single-stranded DNA ends are decorated with the meiosis-specific protein DMC1 (*Neale and Keeney, 2006*).

In vivo experiments to date have mapped the locations of intermediate events in recombination by performing Chromatin ImmunoPrecipitation with high-throughput sequencing (ChIP-seq) against the H3K4me3 mark and the DMC1 mark in testis tissue from mice and humans (*Baker et al., 2014*; *Smagulova et al., 2011*; *Brick et al., 2012*; *Pratto et al., 2014*; *Davies et al., 2016*), or by sequencing DNA fragments that remain attached to SPO11 after DSB formation in mice (*Lange et al., 2016*). Recent studies have also published direct PRDM9 ChIP-seq results using a custom antibody in mouse testes (*Baker et al., 2015a*; *Walker et al., 2015*; *Grey et al., 2017*). To study the DNA-binding properties of mouse PRDM9, one study sequenced genomic DNA fragments bound in vitro

**eLife digest** Human cells have two copies of each chromosome: one from the mother, and one from the father. When cells divide to form sex cells, such as sperm or egg cells, the maternal and paternal chromosomes line up next to each other and swap some of their DNA. This process, known as genetic recombination, creates different versions of genes and ensures that we are all unique – or genetically diverse.

Recombination is a complex process that is largely controlled by a protein called PRDM9. This protein binds DNA at particular spots on the chromosome and directs other proteins to carry out recombination nearby. However, not all of PRDM9's binding sites are known, and not all regions that PRDM9 binds to undergo recombination. Until now, it was not understood why this is the case at fine scales.

To investigate this further, Altemose et al. activated the human version of PRDM9 in human kidney cells grown in the laboratory. The results showed that PRDM9 often bound near the start sites of genes, although these regions rarely undergo recombination in humans. When PRDM9 bound near these sites, it sometimes turned the gene on, which suggests that it may also help to regulate the activity of genes.

Moreover, a specific group of DNA-binding proteins, called KRAB-ZNF proteins, appear to suppress recombination wherever they bind, which explains why some PRDM9 binding sites do not recombine. Lastly, Altemose et al. discovered that the part of PRDM9 that binds to DNA can also bind to other copies of PRDM9 proteins. This self-binding ability might play a role in bringing together the maternal and paternal chromosomes at the correct spots during recombination.

Together, these results shed new light on the recombination process, which is a driving force in the formation of new species and essential for fertility. A next step will be to study these results further in tissues of the reproductive organs. This will provide a better understanding of the forces that shape human evolution.

DOI: https://doi.org/10.7554/eLife.28383.002

by recombinant proteins containing only the PRDM9 ZF array (*Walker et al., 2015*). In humans, recombination hotspots identified by DMC1 mapping and by Linkage Disequilibrium (LD) mapping have enabled the discovery of human PRDM9 binding motifs (*Myers et al., 2008, 2010*; *Hinch et al., 2011*; *Pratto et al., 2014*; *Davies et al., 2016*). However, these published motifs are neither sufficient nor necessary to predict genome-wide PRDM9 binding, DSB formation, or recombination events (*Myers et al., 2010*; *Pratto et al., 2014*), and it has been suggested that binding might be influenced by chromatin features in cis (*Walker et al., 2015*). Moreover, not all PRDM9 binding sites become hotspots (*Baker et al., 2014*; *Grey et al., 2017*), and the reasons for this remain unclear. In particular, apart from PRDM9 motifs themselves, there are no specific DNA sequence features that have been shown to modulate recombination rate in cis in mammals.

The H3K4me3 mark has been associated with meiotic recombination initiation in budding yeast (*Borde et al., 2009*), which lack PRDM9, as well as in PRDM9 knockout mice (*Brick et al., 2012*). Recent work has suggested that this histone mark is bound by CXXC1, a protein that also binds to PRDM9's KRAB domain and to the axis-associated protein IHO1 (*Imai et al., 2017*). Because the H3K4me3 mark is also found at active gene promoters (*Santos-Rosa et al., 2002*), PRDM9 has been hypothesized to play a role in meiotic gene regulation, in addition to its role in initiating recombination (*Hayashi et al., 2005*; *Mihola et al., 2009*). In fact, PRDM9 was shown to activate transcription in a reporter gene assay (*Hayashi et al., 2005*), and its SET domain has been shown to de-repress a subset of genes when tethered to their promoters (*Cano-Rodriguez et al., 2016*). However, recent experiments demonstrate full fertility in transgenic mice with completely remodeled PRDM9 binding landscapes (*Baker et al., 2014*; *Davies et al., 2016*), suggesting that PRDM9 has no essential role in gene activation. This does not preclude the possibility that PRDM9 may play a secondary gene regulatory role in meiosis. PRDM9 has also been shown to bind to itself and form multimers in transfected cells, while maintaining its ability to bind DNA and trimethylate histones (*Baker et al., 2015b*). However, it is not known which domains of PRDM9 mediate this multimer formation activity nor whether PRDM9 allelic variation impacts multimerization.

To investigate the properties of PRDM9's zinc-fingers in humans as they relate to the questions posed above, we expressed several engineered versions of PRDM9 in a mitotic human cell line (HEK293T), then performed various high-throughput sequencing experiments. While this approach cannot reproduce cell-type-specific phenomena found only in spermatocytes and oocytes, it nevertheless enables us to infer some of the fundamental rules governing the behavior of PRDM9 in the nucleus. Indeed, as we describe below, this system replicates many of the key properties of PRDM9 binding in vivo. In these cells, we performed ChIP-seq against human PRDM9, H3K4me3, H3K36me3, and chimp PRDM9, as well as ATAC-seq (Assay for Transposase-Accessible Chromatin with high-throughput sequencing) to examine nucleosome positioning and DNA accessibility, and RNA-seq to examine gene expression. Importantly, by comparing data from transfected and untransfected cells (in which there is weak endogenous *PRDM9* expression), we can observe the same genomic sites with and without the effects of PRDM9 overexpression. This approach also allows us to rapidly engineer and test various different alleles and truncations of PRDM9 to explore the properties of its individual domains. Further, our results are complemented by previously published data on LD-based recombination hotspots (*Frazer et al., 2007*), DSB hotspots decorated by DMC1 (*Pratto et al., 2014*), H3K4me3 in human testes (*Pratto et al., 2014*), and histone modifications across human cell types (*Kundaje et al., 2015*), which we jointly analyze to understand the regulation of recombination outcomes downstream of PRDM9 binding. As described below, our results implicate a widespread role for *other* zinc-finger genes in suppressing, rather than activating, meiotic recombination in humans.

## Results

### A map of direct PRDM9 binding in the human genome

We performed ChIP-seq in HEK293T cells transfected with the human PRDM9 reference allele (the 'B' allele) containing an N-terminal YFP tag that was targeted for immunoprecipitation. To identify regions bound by PRDM9, we modeled binding enrichment relative to a measure of local background coverage at each position in the genome (detailed in Appendix 1), which accounts for local differences in sequencing coverage, including differences attributable to the known aneuploidy of this cell line (*Graham et al., 1977*; *Bylund et al., 2004*; *Lin et al., 2014*). This yielded 170,198 PRDM9 binding peaks across the genome ($p<10^{-6}$), demonstrating that PRDM9 can bind with some affinity to many sites outside of recombination hotspots, which number in the tens of thousands (*Myers et al., 2005*; *Pratto et al., 2014*). This large number of peaks likely results from the high expression level of PRDM9 in this system, providing sensitivity to detect even weak binding interactions, although it may be attributable in part to the chromatin organization of this cell type.

We compared our ChIP-seq data with a set of 18,343 published in vivo human DSB hotspot peaks from DMC1 ChIP-seq experiments in testis samples (*Pratto et al., 2014*). We found evidence for binding at 74% of DSB hotspots (at $p<10^{-3}$) after correcting for chance overlaps (see Materials and methods). The proportion bound in our system is greater (up to 82%) at DSB hotspots >15 Mb from telomeres, which show elevated recombination rates in human males (*Dib et al., 1996*; *Pratto et al., 2014*; *Figure 1—figure supplement 1a*). Overlap probabilities increase with both PRDM9 binding strength and DMC1 heat (*Figure 1b*; *Figure 1—figure supplement 1b*). Furthermore, at PRDM9 binding sites, we observed peaks in LD-based recombination rates (HapMap CEU map, *Frazer et al., 2007*), which increase with PRDM9 binding strength (*Figure 1c–d*), as does DMC1 enrichment (*Figure 1—figure supplement 2c*). Therefore, despite cell-type differences between our HEK293T expression system and the chromatin environment of early meiotic cells, our binding peaks capture the majority of biologically relevant recombination hotspots and reveal many additional non-hotspot sites bound by PRDM9 in HEK293T cells.

### PRDM9 can bind multiple motifs with different internal spacings

Next, we leveraged the large number and high resolution of our ChIP-seq peaks to search for sequence motifs at PRDM9 binding sites using a Bayesian de novo motif-finding algorithm (described in *Davies et al., 2016* and in Materials and methods). Rather than yielding a single motif described by a position weight matrix (PWM), this algorithm allows binding sites to be described by a mixture of multiple motifs enriched in peak centers. The algorithm identified seven non-

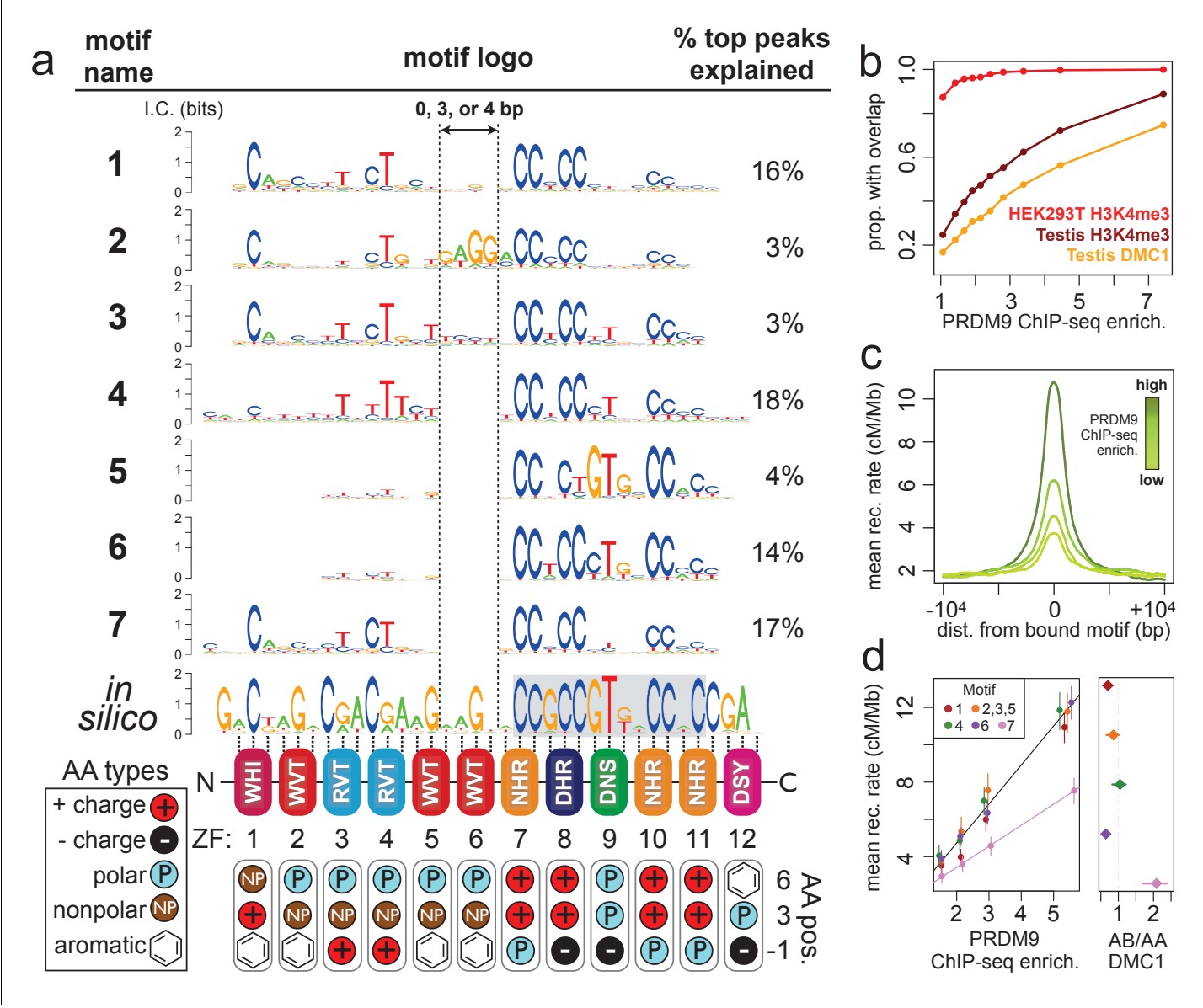

**Figure 1.** Comparison of seven distinct motifs bound by human PRDM9 (B allele). (a) Seven motif logos produced by our algorithm (applied to the top 5,000 PRDM9 binding peaks ranked by enrichment, after filtering out repeat-masked sequences) were aligned to each other and to an in silico binding prediction (*Myers et al., 2010*; *Persikov et al., 2009*; *Persikov and Singh, 2014*, maximizing alignment of the most information-rich bases. The position of the published hotspot 13-mer is indicated by the gray box overlapping the in silico motif (*Myers et al., 2008*). On the right is the percentage of the top 1,000 peaks (ranked by enrichment without further filtering) containing each motif type. Zinc-finger residues at 3 DNA-contacting positions (labeled −1, 3, 6) are illustrated below each ZF position, classified by polarity, charge, and presence of aromatic side chains. ZFs 5 and 6 lack positively charged amino acids and contain aromatic tryptophan residues, and they coincide with a variably spaced motif region (indicated by vertical dotted lines). Motif 4 is truncated here. (b) H3K4me3 ChIP-seq data from PRDM9-transfected HEK293T cells (this study) and H3K4me3/DMC1 data from testes (*Pratto et al., 2014*) were force-called to provide a p-value for enrichment of each sample in a 1 kb window centered on each PRDM9 peak (filtered to remove coverage outliers and those overlapping H3K4me3 peaks in untransfected cells). PRDM9 enrichment values are unitless (equal to the estimated signal divided by background, minus 1 and set to 0 if negative, at the base with the smallest p-value within each peak). Peaks were split into deciles according to their PRDM9 enrichment values, and the proportion of peaks with a force-called H3K4me3 or DMC1 p-value <0.05 is plotted within each decile. (c) Peaks were stratified into quartiles based on increasing PRDM9 enrichment (light green to dark green) after filtering out promoters. Mean recombination rates (from the HapMap LD-based recombination map, *Frazer et al., 2007*) at each base in the 20 kb region centered on each bound motif are plotted for each quartile, with smoothing (ksmooth, bandwidth 25). (d) *Left plot*: Peak enrichment quartiles (filtered to remove promoters as in c) were separated by motif type (Motifs 2, 3, and 5 were combined due to low abundance), and the mean HapMap CEU recombination rate overlapping peak centers was plotted against median PRDM9 enrichment in each quartile, with lines of best fit added for Motif 7 (pink) versus all

*Figure 1 continued on next page*

*Figure 1 continued*

other motifs. *Right plot*: Fold enrichment of each motif in AB-only DMC1 peaks versus AA-only DMC1 peaks (*Pratto et al., 2014*). Error bars indicate two standard errors of the mean (left plot) or 95% bootstrap confidence intervals (right plot).

DOI: https://doi.org/10.7554/eLife.28383.003

The following source data and figure supplements are available for figure 1:

**Source data 1.** List of all ChIP-seq samples.
DOI: https://doi.org/10.7554/eLife.28383.008
**Source data 2.** PWMs for all motifs, in MEME format.
DOI: https://doi.org/10.7554/eLife.28383.009
**Figure supplement 1.** DMC1, H3K4me3, and H3K36me3 signals surrounding human PRDM9 peaks.
DOI: https://doi.org/10.7554/eLife.28383.004
**Figure supplement 2.** Comparison of PRDM9 and H3K4me3/DMC1 enrichment values.
DOI: https://doi.org/10.7554/eLife.28383.005
**Figure supplement 3.** All motifs found in human PRDM9 peaks.
DOI: https://doi.org/10.7554/eLife.28383.006
**Figure supplement 4.** Motif 7 represents a binding mode favored by the B allele.
DOI: https://doi.org/10.7554/eLife.28383.007

degenerate motifs, representing distinct PRDM9 binding modes. These explain 75% of the strongest 1000 binding peaks, falling to 53% of all peaks (*Figure 1a*). The remaining peaks contain mostly degenerate, GC-rich sequences (*Figure 1—figure supplement 3*), similar to DMC1 hotspots in transgenic mice containing this same human PRDM9 allele (*Davies et al., 2016*) and interpretable as binding to clusters of individually weaker motif matches in mostly GC-rich regions.

While each of the seven motifs has a close internal match to the published 13-mer found in human recombination hotspots (*Myers et al., 2008*), allowing for multiple binding modalities revealed that the zinc fingers predicted to bind upstream of this 13-mer (ZFs 1–6) can comparably high sequence specificity (*Figure 1a*). We aligned our seven motifs to each other and to an in-silico motif prediction (based on the zinc-finger domain's amino acid sequence alone; *Myers et al., 2010*; *Persikov et al., 2009*; *Persikov and Singh, 2014*), revealing differences across motifs driven mainly by variable internal spacings (*Figure 1a*) alongside smaller differences in base-pair preferences (e.g. Motif 5). The region corresponding to ZF5 and ZF6 is predicted to span 6 bp, but in Motifs 4–7 this region spans only 2 bp, and in Motif 1 it spans only 5 bp. Interestingly, we only observed these three particular spacings, and the expected 6 bp binding footprint is observed only for Motifs 2 and 3, which explain a relatively small proportion of peaks (6%). This alternative spacing cannot be captured in a single motif, possibly explaining why ZFs 1–6 have shown weak sequence specificity in previously published hotspot motifs (*Myers et al., 2008*, *2010*; *Hinch et al., 2011*; *Pratto et al., 2014*).

Alternative spacing within motifs could explain how long zinc-finger arrays like PRDM9's are able to consecutively bind DNA despite theoretical physical constraints (*Persikov and Singh, 2011*), similar to multivalent CTCF binding (*Nakahashi et al., 2013*). Our results are also consistent with recent findings that truncated mouse PRDM9 alleles can stably bind discontinuous submotifs, though at reduced specificities, with subsets of zinc fingers (*Striedner et al., 2017*). ZF5 and ZF6, which overlap the variably spaced region, have large, aromatic tryptophan residues at the DNA-contacting '−1' position (*Figure 1a*). They also lack the positively charged DNA-contacting residues found in the most sequence-specific zinc fingers in the array (consistent with an electrostatic attraction to the negatively charged DNA). We speculate that these bulky, uncharged middle zinc fingers might fail to bind DNA strongly and may act more like a linker between the more strongly binding zinc fingers found upstream and downstream.

Interestingly, we observed a lower mean LD-based recombination rate (*Frazer et al., 2007*) around Motif 7 peaks, not explained by differences in PRDM9 binding enrichment, promoter overlap, repeat overlap, or H3K4me3 enrichment (*Figure 1d*, *Figure 1—figure supplement 4*). We hypothesized that Motif 7 might be favorably bound by the B allele and thus underrepresented in LD-based recombination maps, which are dominated by historical recombination events initiated by the more common A allele of PRDM9, which differs at a single DNA-contacting amino acid in ZF5 (*Baudat et al., 2010*). To test this hypothesis, we searched for our seven motifs in DSB hotspots

unique to an individual with an A/B PRDM9 genotype, then compared these to DSB hotspots found in homozygous A/A individuals (*Pratto et al., 2014*). We found that Motif 7 is two-fold enriched in A/B-only hotspots relative to A/A hotspots, while all other motifs are found in more similar proportions between the two sets (*Figure 1d*). Motif 7 also resembles, but extends, a motif previously identified in A/B-only hotspots (*Pratto et al., 2014*). We conclude that the B allele must bind Motif 7 with greater affinity than does the A allele, demonstrating distinguishable binding preferences between these highly similar PRDM9 alleles.

## PRDM9 deposits H3K4me3 essentially everywhere it binds

We investigated the histone methylation activity of PRDM9 by performing ChIP-seq against the H3K4me3 mark in transfected and untransfected cells. After subtracting sites overlapping 'pre-existing' H3K4me3 peaks (those present in untransfected cells), we found that 95% of PRDM9 binding peaks show H3K4me3 following transfection (p<0.01), and this proportion increases to 100% with increasing PRDM9 binding enrichment (see *Figure 1b*). That is, PRDM9 makes the H3K4me3 mark essentially everywhere it binds, regardless of the pre-existing chromatin substrate, with H3K4me3 signal strength increasing with PRDM9 binding strength ($r$=0.48, *Figure 1—figure supplement 1c*, *Figure 1—figure supplement 2*). As observed in mice (*Davies et al., 2016*; *Powers et al., 2016*; *Grey et al., 2017*), we also observe localized H3K36me3 deposition at bound sites (see *Figure 1—figure supplement 1d*).

Apart from depositing H3K4me3/H3K36me3 locally around its binding sites, PRDM9 has been shown to phase surrounding nucleosomes in vivo in mice (*Baker et al., 2014*). To investigate this behavior in transfected HEK293T cells, we performed ATAC-seq and found that full-length PRDM9 appears to phase surrounding nucleosomes even in this completely different cell type and expression system (see *Figure 2—figure supplement 3a*). However, when we transfected a truncated version of PRDM9 including only the zinc-finger domain, we saw no evidence of nucleosome phasing around PRDM9 binding sites (see *Figure 2—figure supplement 3b*). Instead, its ATAC-seq coverage pattern appears similar to that of unstransfected cells or of cells transfected with a truncated version of PRDM9 excluding the zinc-finger domain (*Figure 2—figure supplement 3c,d*). We confirmed that this 'ZF only' truncated protein localizes to the nucleus (see *Figure 2—figure supplement 4*), and previous studies have shown that PRDM9's ZF array is sufficient to bind DNA (*Walker et al., 2015*; *Striedner et al., 2017*). This suggests that PRDM9's nucleosome phasing behavior stems not only from the binding of its ZF array to DNA, but may involve steric effects of the non-ZF region or require histone methylation.

## Human PRDM9 frequently binds promoters

A study in mice has shown that, in the absence of PRDM, DSBs localize to active promoters marked with H3K4me3, suggesting that PRDM9 may serve to provide alternative H3K4me3 sites to compete with and direct recombination away from promoters (*Brick et al., 2012*). However, our ChIP-seq data revealed that, surprisingly, of the 12,982 protein-coding genes with H3K4me3 surrounding their Transcription Start Site (TSS) in our untransfected cells (p<$10^{-5}$), 81% have a PRDM9 binding peak center within 500 bp of the TSS, compared to only 6% expected by chance overlap (yielding a corrected overlap fraction of 79%). At promoters with little or no prior H3K4me3, the proportion bound by PRDM9 decreases to 15% (corrected for chance overlaps, *Figure 2a*), though this difference could potentially be explained by increasing power to detect weak binding events at more active genes. If we concentrate only on the strongest quartile of PRDM9 binding enrichment at promoters, we see that roughly 10% of promoters are strongly bound, regardless of H3K4me3 enrichment (*Figure 2a*).

Previous datasets in humans have been unable to detect this affinity for promoters because they relied on H3K4me3, DMC1, or LD mapping as proxies for inferring PRDM9 binding sites (*Baker et al., 2015b*; *Pratto et al., 2014*; *Myers et al., 2010*). Since active promoters contain PRDM9-independent H3K4me3 peaks, they are filtered out from H3K4me3 analyses, and since DSBs are suppressed at promoters (at least in the presence of PRDM9, as shown by *Brick et al., 2012*), promoters are underrepresented in DMC1 and LD-based recombination hotspots. One recent study mapped binding of the human PRDM9 B allele in HEK293T cells by ChIP-exo, yielding a conservative set of 839 peaks after stringent filtering (*Imbeault et al., 2017*). Of these 839 peaks, 87% overlap

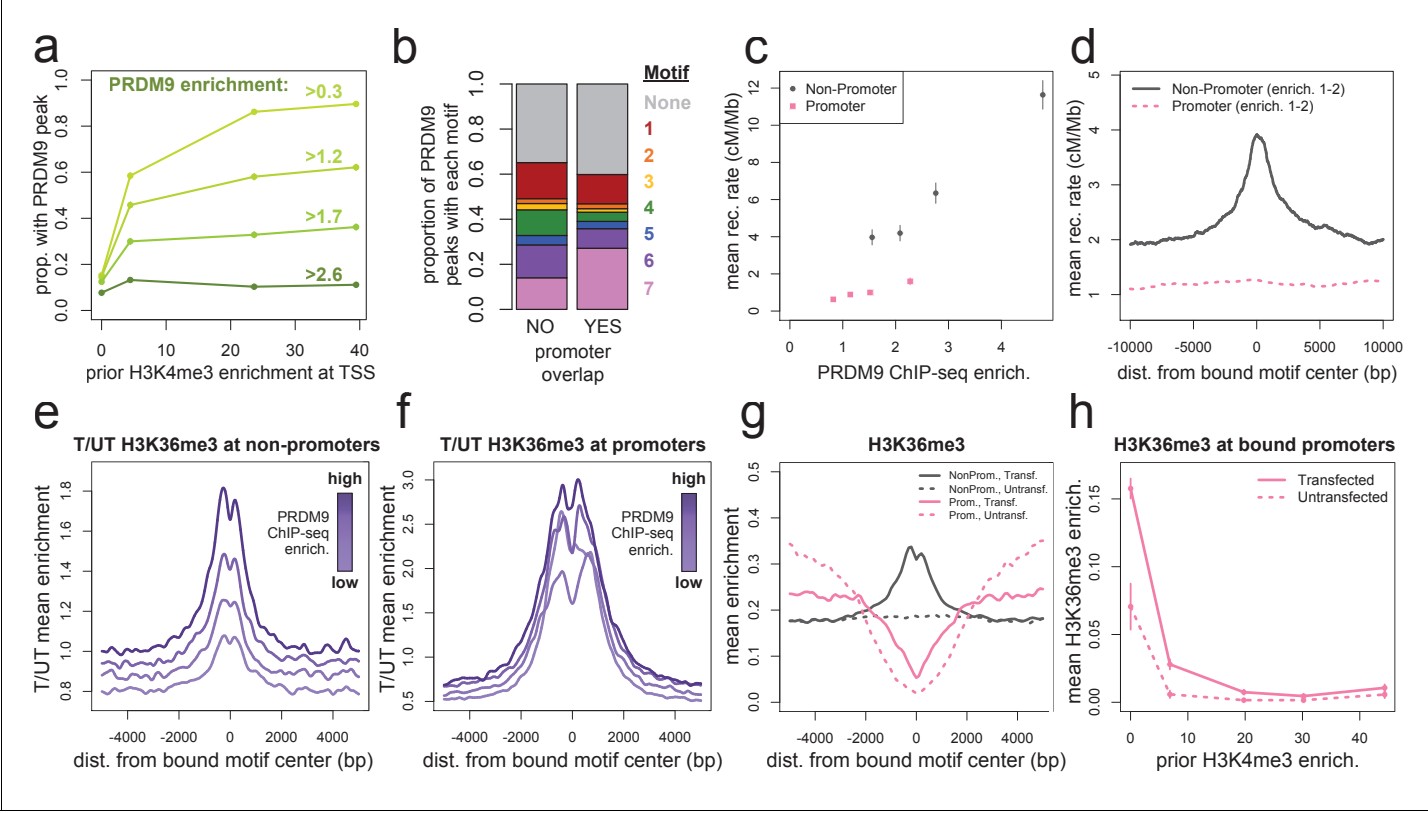

**Figure 2.** Human PRDM9 can bind promoters, though recombination is suppressed. (a) The chance-corrected proportion of protein-coding genes that have a PRDM9 peak center occurring within 500 bp of the TSS, stratified by different PRDM9 enrichment value thresholds (shades of green, with thresholds listed), in each quartile of force-called H3K4me3 enrichment surrounding the TSS in untransfected cells. The power to detect weaker binding events increases at more active promoters (as measured by H3K4me3), though strong PRDM9 binding events appear at roughly 10% of all promoters regardless of activity. (b) Barplot illustrating the proportion of promoter or non-promoter PRDM9 peaks assigned each of the 7 motifs (or no motif, in gray). Motif 7 appears 2-fold enriched in promoter peaks. (c) Mean HapMap CEU recombination rates are reported for promoter (pink squares) and non-promoter (gray circles) human PRDM9 peaks split into quartiles of PRDM9 enrichment (filtered not to overlap repeats or occur within 15 Mb of a telomere; error bars represent two standard errors of the mean). Both median enrichment values and recombination rates are greater for non-promoter peaks, even in overlapping ranges of PRDM9 enrichment. (d) Mean recombination rate in 20 kb windows centered on bound motifs, for promoter (pink) and non-promoter (gray) peaks further filtered only to include peaks with PRDM9 enrichment values between 1 and 2 (smoothing: ksmooth bandwidth 200). (e) Mean H3K36me3 enrichment in transfected cells divided by mean H3K36me3 enrichment in untransfected cells at 36,000 non-promoter PRDM9 binding sites split into quartiles of PRDM9 enrichment (shades of purple). (f) same as e but for 10,000 promoter PRDM9 binding sites split into quartiles of PRDM9 enrichment. (g) The absolute mean enrichment values used to generate plots e and f, split into transfected (solid) and untransfected (dotted) samples at promoter (pink) and non-promoter (gray) PRDM9 binding sites in the top quartile of PRDM9 enrichment. There is a depletion of H3K36me3 coverage surrounding promoters in untransfected cells, but the magnitude of this depletion decreases in transfected cells. (h) At 4,000 protein-coding genes with a strong PRDM9 binding peak within 500 bp of the TSS (PRDM9 enrichment >2 and <10), we show the relationship between force-called H3K4me3 enrichment and force-called H3K36me3 enrichment in the 1 kb surrounding each TSS, for both transfected and untransfected cells (solid and dotted lines). Error bars indicate two standard errors of the mean H3K36me3 enrichment within each quintile of H3K4me3 enrichment. H3K36me3 enrichment increases in transfected cells at all strongly bound promoters, but this effect diminishes almost to 0 as promoter activity increases (which forces H3K36me3 close to 0 in all cells). This effect cannot be accounted for by the modest decrease in PRDM9 enrichment at more active promoters (mean PRDM9 enrichment decreases from 4.3 in the first H3K4me3 quintile to 3.1 in the fifth quintile).

DOI: https://doi.org/10.7554/eLife.28383.010

The following figure supplements are available for figure 2:

**Figure supplement 1.** Chimp w11a PRDM9 binds a T-rich motif away from human binding sites.

DOI: https://doi.org/10.7554/eLife.28383.011

**Figure supplement 2.** Human PRDM9 can bind promoters, though DSBs do not occur.

DOI: https://doi.org/10.7554/eLife.28383.012

**Figure supplement 3.** ATAC-seq profiles showing nucleosome phasing around PRDM9 binding sites.

DOI: https://doi.org/10.7554/eLife.28383.013

**Figure supplement 4.** PRDM9s ZF domain is necessary and sufficient for nuclear localization.

*Figure 2 continued on next page*

*Figure 2 continued*

DOI: https://doi.org/10.7554/eLife.28383.014

our 170,198 peaks, and they are similarly enriched in promoters (18% occur within 500 bp of a TSS, versus 6% when shifted 5 kb, compared to 15% and 7% with our peaks, respectively).

To exclude the possibility that PRDM9 binding peaks observed at promoters were false positives (*Jain et al., 2015*), we performed two ChIP-seq replicates on cells transfected with a PRDM9 construct in which we replaced the human ZF domain with the ZF domain from the chimpanzee w11a allele, which is not predicted to bind the GC-rich DNA commonly found at promoters (*Auton et al., 2012*; *Schwartz et al., 2014*). We found that the chimp allele binds a T-rich motif (*Figure 2—figure supplement 1c*), and only 5% of chimp PRDM9 peaks occur within 500 bp of a human peak center, below the 8% expected by chance (*Figure 2—figure supplement 1b*). In contrast to results for human PRDM9, only 3% of promoters fall within 500 bp of a chimp PRDM9 peak, versus 9% expected by chance overlap, confirming that the promoter peaks we observe for the human allele are unlikely to be ChIP-seq artifacts.

Furthermore, motif identification at human PRDM9's promoter binding sites identified the expected binding motifs at similar frequencies to non-promoter peaks, except for a twofold enrichment of Motif 7 (*Figure 2b*). Interestingly, Motif 7 is also the B-allele-enriched motif, so PRDM9's promoter affinity might also differ between common human alleles. We suggest that these GC-rich motifs, together with accessible chromatin, enable human PRDM9 to consistently bind to promoter regions in HEK293T cells (*Figure 2a*). Notably, however, PRDM9 peaks in promoters tend to have lower mean enrichment estimates across a range of motif FIMO scores (*Figure 2—figure supplement 2d*). It is also worth noting that in vivo mapping of PRDM9 binding will be required to confirm that promoter binding occurs in meiotic cells, although it is difficult to understand how this sequence-dependent binding could be cell-type-specific across all promoters.

## PRDM9-induced H3K36me3 is depleted at promoters

Although there is widespread binding of human PRDM9 to promoters in HEK293T cells, we observe little to no elevation in local recombination rate or testis DMC1 enrichment at these binding sites (*Figure 2c,d*, *Figure 2—figure supplement 2e,f*). In the absence of PRDM9, DSBs localize to promoters in mice (*Brick et al., 2012*), but in light of our results, it remains difficult to explain how recombination might be suppressed at promoters despite direct PRDM9 binding. A second mark, H3K36me3, is also deposited by PRDM9 at many of its binding sites in vivo (*Powers et al., 2016*), and it shows a similar pattern to H3K4me3 around DSB sites in mice (*Yamada et al., 2017*).

At both non-promoter and promoter PRDM9 peaks, we observed a similar enrichment of H3K36me3 in transfected relative to untransfected cells (*Figure 2e,f*), confirming that PRDM9 indeed binds these sites. However, a very strong depletion of H3K36me3 around promoters in untransfected cells means that *absolute* levels of H3K36me3 remain low in promoters, relative to non-promoter binding sites (*Figure 2g*). Interestingly, the amount of H3K36me3 deposited by PRDM9 at promoters negatively correlates with the amount of H3K4me3 enrichment at those promoters in untransfected cells, and this cannot be explained by differential PRDM9 binding (*Figure 2h*). This suggests that at highly active promoters, PRDM9 is less able to deposit H3K36me3, or this mark is actively removed. This difference between promoter and non-promoter binding sites could in principle explain the lack of recombination at promoters, if the simultaneous presence of both H3K36me3 and H3K4me3 influences recombination initiation, as has been suggested by *Powers et al. (2016)* and shown to be consistent with DSB data by *Yamada et al. (2017)*. In humanized mice, in vivo DSB hotspot sites favor motif positions with lower PRDM9-independent H3K4me3 levels than genomic background (*Davies et al., 2016*), and this seems highly concordant with our human results.

## PRDM9 can activate transcription of some genes, including *VCX* and *CTCFL*

We have shown that human PRDM9 binds promoters and deposits the H3K4me3 mark wherever it binds in HEK293T cells, which raises the possibility that PRDM9 may affect gene expression, given

that H3K4me3 is highly enriched at active promoters (*Santos-Rosa et al., 2002*). Tethering PRDM9's SET domain to other promoter-binding proteins has been shown to de-repress gene expression in a context-dependent manner (*Cano-Rodriguez et al., 2016*), leading us to hypothesize that full-length human PRDM9 might also be able to activate gene expression. We therefore performed RNA-seq in cells transfected with human PRDM9, along with control samples that were either untransfected, transfected with the chimp allele, or transfected with a construct containing only the human zinc-finger domain (and incapable of H3K4me3 deposition; referred to as 'ZF only'; all constructs illustrated in Figure 5a).

Seven transcripts showed overwhelming evidence of being differentially expressed in cells transfected with the human allele versus all other samples, with all seven being upregulated by PRDM9 presence. Five overlap known genes: *MEG3*, *ONECUT3*, *LGALS1*, *VCX*, and *CTCFL*. Interestingly, the latter two genes are normally expressed only in spermatogenesis (*Lahn and Page, 2000*; *Sleutels et al., 2012*). We validated expression induction at these two genes using qPCR (*Figure 3*).

*CTCFL* is a variant of chromatin regulator *CTCF*, and in mice it has been shown to be expressed exclusively in pre-leptotene spermatocytes (*Sleutels et al., 2012*). Male knockout mice show greatly reduced fertility due to meiotic arrest (*Sleutels et al., 2012*), and variants at *CTCFL* influence genome-wide recombination rates in human males (*Kong et al., 2014*). CTCFL may be involved in organizing the meiotic chromatin landscape and regulating the transcription of meiotic genes (*Sleutels et al., 2012*). We found that *CTCFL* RNA levels increase 28-fold after transfection with the human allele, from a nearly undetectable baseline transcription level (*Figure 3*; we note this may underestimate the true relative expression level given that transfection efficiency is not 100%). PRDM9 binds strongly to a GC-rich repeat near the *CTCFL* TSS and deposits H3K4me3, which is absent in untransfected cells (*Figure 3*). The chimp PRDM9 allele, in contrast, does not bind near the TSS and does not show elevated transcript levels after transfection (*Figure 3*).

*VCX* encodes a small, highly charged protein of unknown function and has been previously studied for its involvement in PRDM9-related non-homologous recombination events and X-linked ichthyosis (*Myers et al., 2008*; *Van Esch et al., 2005*). We found that PRDM9 does not in fact bind near the annotated *VCX* TSS, but instead in the middle of the gene and very strongly at a minisatellite array of PRDM9 binding motifs (*Myers et al., 2008*) near the terminus of the gene (*Figure 3—figure supplement 1*). PRDM9 adds the H3K4me3 mark throughout the gene's coding regions in a pattern similar to that seen in testes (*Figure 3—figure supplement 1*). RNA-seq coverage suggests normal splicing, but use of an alternative promoter that excludes the first, untranslated exon (*Figure 3—figure supplement 1*).

We note that this result does not establish whether human PRDM9 is necessary or sufficient for *CTCFL* and *VCX* expression in vivo, but still PRDM9 is demonstrably able to trigger the transcription of these genes in a way that depends on the binding of its zinc fingers. Previous work has shown that *Prdm9* expression begins in pre-leptotene cells in mice (*Sun et al., 2015*), concurrent with *Ctcfl* expression (*Sleutels et al., 2012*) and thus supports the possibility that PRDM9 may promote *CTCFL* transcription in vivo. The failure of the chimp allele to bind to or activate the expression of human *CTCFL* further suggests that this behavior may not be essential across organisms, although the chimp allele might in principle still bind the *CTCFL* promoter in the chimp genome. Similarly, there is no evidence that human PRDM9 alleles with very different binding preferences, such as the C allele, would bind the same promoter. Also notably, the motif bound at the *CTCFL* promoter is Motif 7, so the A and B alleles may bind this locus with different affinities.

43 additional genes showed weaker evidence of being activated by human PRDM9 binding near their annotated transcription start sites, with 41 showing increases, as opposed to decreases, in expression (*Figure 3—source data 2*). We lack power to detect small changes in gene expression, especially decreases in expression (*Trapnell et al., 2012*). Nonetheless, it is likely that effects of similar magnitude to *CTCFL* and *VCX* are quite rare. Our data do make it clear that PRDM9 binding and histone trimethylation near a promoter can trigger or enhance gene expression in some cases. Furthermore, this effect on gene expression is not likely to result from PRDM9 binding alone but from its trimethylation activity, given that transfection with the zinc fingers alone does not trigger expression. Further work will need to establish if promoter-binding PRDM9 alleles are able to regulate gene expression in vivo, whether as an accidental side effect of binding or specifically functional, though this work may remain challenging in humans.

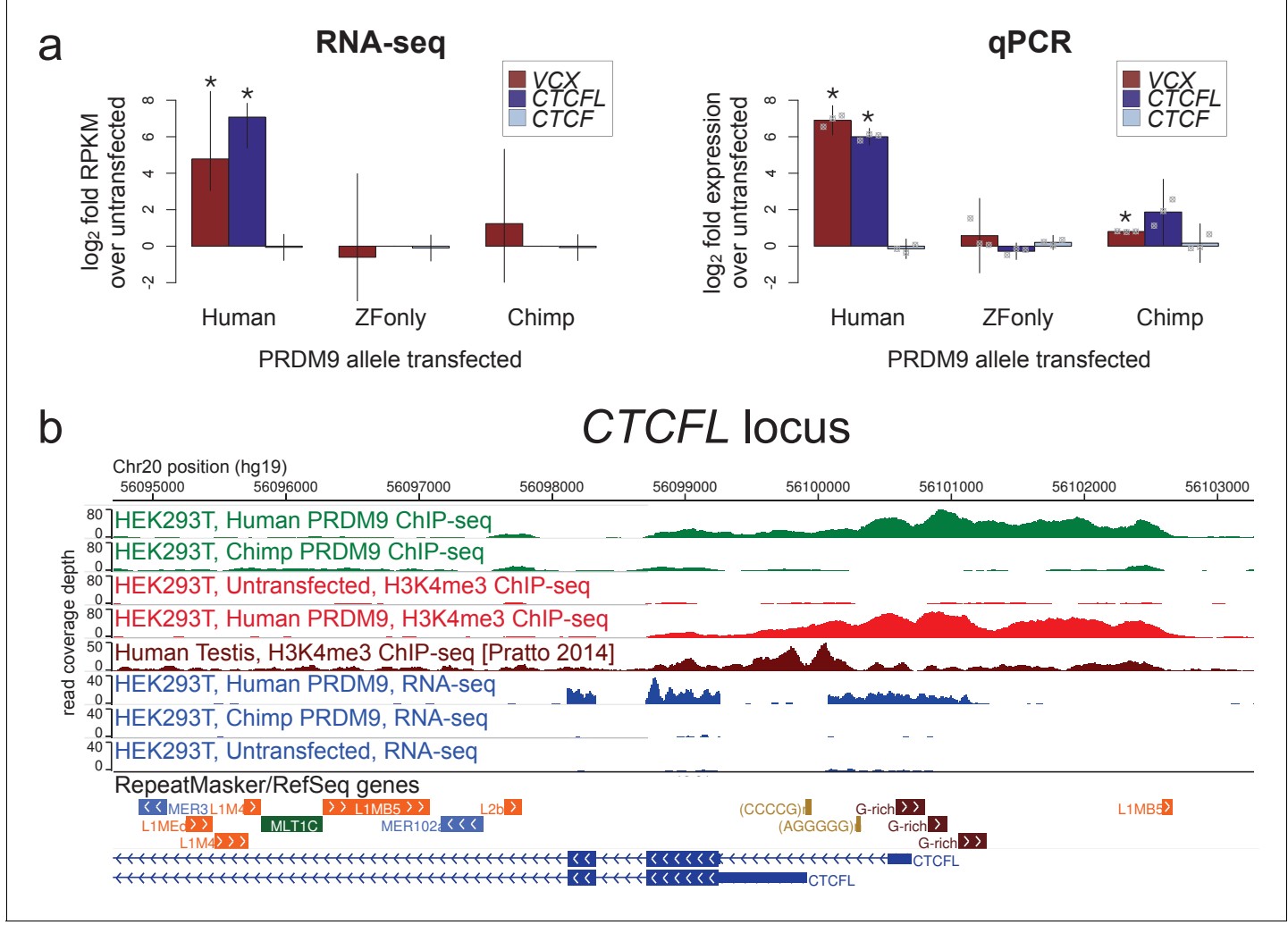

**Figure 3.** Spermatogenesis-specific genes *VCX* and *CTCFL* are activated by human PRDM9 in HEK293T cells. (**a**) left: Bar plots showing the $\log_2$ fold change relative to untransfected HEK293T cells in computed FPKM values (fragments per kilobase of transcript per million mapped RNA-seq reads) for HEK293T cells transfected with the human allele, the chimp allele, or a construct containing only the human Zinc-Finger domain, for *CTCFL* and *VCX*, with *CTCF* as a negative control. Error bars conservatively represent maximum ranges of the ratios given confidence intervals for FPKM values computed by cufflinks (*Trapnell et al., 2012*). Asterisks indicate significant differential gene expression, as reported by CuffDiff (p<0.0001). right: qPCR validation results for the same genes from 3 independent biological replicates. Y-axis values are $\log_2$ ratios of $\Delta\Delta$ $C_t$ values for each gene relative to the untransfected sample (normalized to the *TBP* housekeeping gene; see Materials and methods). Error bars represent 95% confidence intervals from 3 biological replicates (t distribution; gray points represent individual replicate values), and asterisks indicate p<0.001 (one-tailed t test). (**b**) A browser screenshot (*Zhou et al., 2011*) from Chr20 near the promoter region of *CTCFL* with custom tracks indicating ChIP-seq and RNA-seq raw read coverage data. Human PRDM9, but not chimp PRDM9, (green) binds a G-rich repeat near the TSS, adding an H3K4me3 mark (light red) where none is present in untransfected cells. RNA-seq coverage (blue) spikes in the coding regions in transfected cells, while it is nearly flat in untransfected cells or chimp-transfected cells. Testis H3K4me3 coverage (dark red, from *Pratto et al., 2014*) peaks at a slightly different locus, corresponding to an alternative TSS.
DOI: https://doi.org/10.7554/eLife.28383.015

The following source data and figure supplement are available for figure 3:

**Source data 1.** qPCR primers, Ct values, and calculations.
DOI: https://doi.org/10.7554/eLife.28383.017
**Source data 2.** PRDM9-bound genes with differential expression.
DOI: https://doi.org/10.7554/eLife.28383.018
**Figure supplement 1.** Raw coverage values surrounding the *VCX* promoter.
DOI: https://doi.org/10.7554/eLife.28383.016

# Analysis of THE1B repeats reveals non-PRDM9 motifs influencing recombination

Although our seven motifs (*Figure 1a*) improve our understanding of PRDM9 binding, even the top-scoring 0.1% of motif matches genome-wide have only a 50% chance of overlapping an actual PRDM9 binding peak (see *Figure 2—figure supplement 2a*). Moreover, at best we only observe a 55% correlation between H3K4me3 and DMC1 enrichment values from testis data surrounding our PRDM9 binding sites (*Figure 1—figure supplement 2f*). Therefore, other influences such as wider sequence and chromatin contexts must impact both binding and downstream recombination outcomes. The only specific known mammalian sequence feature so far identified as influencing either PRDM9 binding, or downstream recombination events, is the PRDM9 binding motif itself. Thus, it is uncertain which factors prevent or promote hotspot occurrence, whether these act in cis or trans, and what these might be. A powerful approach to identify factors that might influence PRDM9 binding and subsequent hotspot formation is to search for sequence motifs predicting these outcomes. Identified motifs are likely to have a causal influence, so they can help address whether particular histone modifications associated with those motifs have a genuinely causal role themselves.

We hypothesized that sequence motifs unrelated to PRDM9 binding might have strong local effects on recombination outcomes, but these motifs might evade detection if they operate only at a minority of recombination hotspots. To attempt to overcome this and control for the effects of local genetic context, we focused on hotspots centering within one family of retrotransposon elements, called THE1B repeats, which are the most strongly hotspot-enriched among all human repeats (*Myers et al., 2008*). PRDM9 binds directly to a subset of THE1B repeat copies containing matches to its target motif (*Figure 4a*), in a known region of the repeat (*Myers et al., 2008*, see Appendix 2), and THE1B-centered hotspots contribute a substantial fraction of all human A- and B-allele controlled recombination (4.6% measured by DMC1 mapping; *Pratto et al., 2014*). We analyzed over 20,000 THE1B repeats throughout the human genome, which share highly similar sequences perturbed by random mutations. These mutations allowed us to precisely dissect the impact of particular sequence motifs on PRDM9 binding, and on downstream DSB formation (as measured by DMC1 mapping, from *Pratto et al., 2014*) and crossover activity (as measured by LD mapping, from *Frazer et al., 2007*). We used conditional association testing to identify collections of motifs that independently correlate with PRDM9 binding or recombination (see Appendix 2).

Seventeen distinct motifs (*Figure 4a*) were found to influence PRDM9 binding to THE1B copies in HEK293T cells (*Figure 4—source data 1*). All map within the predicted PRDM9 binding region and span the entire region, confirming that all of PRDM9's zinc fingers are involved in binding. Motifs promoting PRDM9 binding associated with higher H3K4me3 enrichment in testes (data from *Pratto et al., 2014*) and with increasing LD/DMC1 hotspot probability, so the same motifs must operate in vivo (*Figure 4a*; detailed in Appendix 2). Importantly for the results described below, binding of PRDM9 does not associate strongly with any sequence motifs outside the directly bound region, so it might act as a local 'pioneer' protein at least on this background, despite results in mice (*Grey et al., 2017*).

We then independently tested for the presence of motifs influencing recombination hotspot formation *conditional* on presence of a PRDM9 binding site in HEK293T cells. We identified an initial seven such motifs (*Figure 4a*; detailed in Appendix 2; *Figure 4—source data 1*). Only three of these map within the PRDM9 binding region and correspond to stronger/weaker PRDM9 enrichment. The remaining four motifs show no association whatsoever with PRDM9 binding in HEK293T cells, and map well outside the PRDM9 binding motif (*Figure 4a*). We refer to these as 'non-PRDM9 recombination-influencing motifs'. The strongest signal is for the motif ATCCATG (joint $p=2.8\times10^{-9}$ for LD-hotspots, OR = 0.32), whose presence within a THE1B repeat produces a 2.5-fold reduction in the surrounding recombination rate at PRDM9-bound THE1B repeats (*Figure 4b*). ATCCATG presence also reduces the local recombination rate around THE1B repeats not bound by PRDM9, implying a more general, PRDM9-independent mechanism of recombination suppression (*Figure 4b*). Notably, this suppression extends beyond the boundaries of the THE1B repeat itself.

We observed strong testis H3K4me3 enrichment at THE1B repeats containing PRDM9 binding motifs regardless of whether 'ATCCATG' was present, and after conditioning on the strength of the PRDM9 motif match (*Figure 4b*). Therefore, this motif must suppress recombination downstream of PRDM9 binding in vivo. In fact, presence of the modifier motif ATCCATG actually modestly

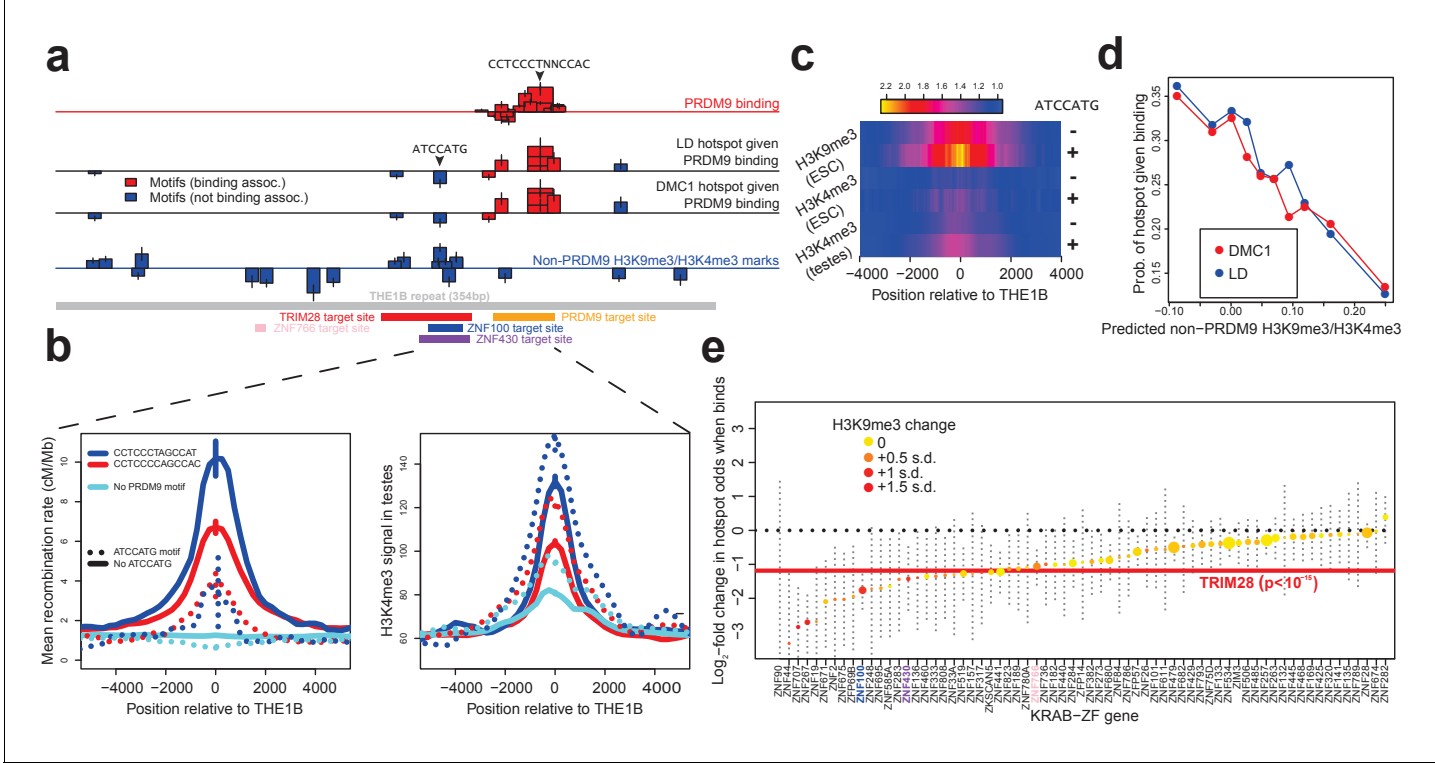

**Figure 4.** Influences on recombination in cis downstream of PRDM9 binding. (**a**) Analysis of THE1B repeats shows the positions along the THE1B consensus (bottom, gray) of motifs influencing PRDM9 binding (top row), motifs influencing recombination hotspot occurrence at bound sites (middle two rows), and motifs influencing H3K4me3/H3K9me3 in testes and somatic cells (bottom row). Rectangle widths show motif size, and heights show log-odds-ratio or effect size (two standard errors delineated). Rectangles below the lines have negative effects. Motifs associated with PRDM9 binding are in red; others in blue. Binding motifs for labeled proteins are at the plot base. (**b**) Left plot shows LD-based recombination rates around the centers of THE1B repeats containing different approximate matches to the PRDM9 binding motif CCTCCC[CT]AGCCA[CT] (colors) and the motif ATCCATG (lines dotted if present). Right plot is the same but shows mean H3K4me3 in testes (from *Pratto et al., 2014*). ATCCATG presence reduces recombination and increases H3K4me3. (**c**) Impact of ATCCATG presence (+) or absence (-) on normalized enrichment values around the centers of THE1B repeats, of H3K4me3 and H3K9me3 in different cells (labeled pairs of color bars, normalized to equal 1 at edges). H3K9me3 shows the strongest signal increase. (**d**) Predicted non-PRDM9 H3K9me3/H3K4me3 versus probability DMC1-based or LD-based hotspots occur at PRDM9-bound sites. For the x-axis repeats were binned according to an additive DNA-based score, using the bottom row of part A and the combination of motifs they contained. (**e**) Estimated impact on whether a hotspot occurs of co-binding by individual KRAB-ZNF proteins (labels; *Imbeault et al., 2017*) near a PRDM9 binding peak (genome-wide, not only within THE1B repeats, after filtering out promoter regions). For each KRAB-ZNF protein, a GLM was used to estimate the impact of KRAB-ZNF binding (binary regressor) on hotspot probability. We show the estimated log$_2$-odds, with 95% CIs. Colors indicate H3K9me3 enrichment increase at co-bound sites. Horizontal line shows the results for TRIM28. Features below the horizontal dotted line have a negative estimated impact on downstream recombination.

DOI: https://doi.org/10.7554/eLife.28383.019

The following source data and figure supplements are available for figure 4:

**Source data 1.** Detailed information on all THE1B motifs.
DOI: https://doi.org/10.7554/eLife.28383.022
**Figure supplement 1.** Features associated with recombination outcomes given PRDM9 binding.
DOI: https://doi.org/10.7554/eLife.28383.020
**Figure supplement 2.** Large-scale recombination rate affects testis DMC1 but not H3K4me3.
DOI: https://doi.org/10.7554/eLife.28383.021

*increased* the testis H3K4me3 signal, even at THE1B copies not containing a PRDM9 motif and not bound by PRDM9 in HEK293T cells (*Figure 4b*), which we return to below. Similar results were observed for the other three non-PRDM9 recombination-influencing motifs.

## Recombination-influencing motifs associate with H3K9me3 and H3K4me3 across many cell types

We hypothesized that the recombination-influencing motifs described above might be bound by chromatin-modifying proteins. To examine this possibility, we independently searched for motifs that could predict chromatin states within THE1B elements. Specifically, we searched de novo for motifs associated with 15 previously identified chromatin states, and individual histone modifications, across each of 125 somatic cell types (*Kundaje et al., 2015*). Strikingly, we observed that the motif ATCCATG (independently identified above as the strongest non-PRDM9 recombination-influencing motif) is also the strongest single predictor of the 'heterochromatin' state, marked by enriched H3K9me3. THE1B repeats containing ATCCATG are heterochromatin-enriched in over half of cell types, especially in embryonic stem cells, and exhibit a strong localized increase in H3K9me3 (*Figure 4c*). More surprisingly, we also observed a weak, but significant, localized increase in H3K4me3 signal ($p=7.5\times10^{-13}$; *Figure 4c*). We also saw the same weak H3K4me3 peak in testes, after restricting analysis to THE1B repeats not bound by PRDM9 (*Figure 4b,c*), indicating this modification operates fully independently of PRDM9. This weak increase might reflect genuine partial co-occurrence of H3K9me3 and H3K4me3 at the same locus (but possibly on different alleles, or in different cells), or in theory it could be explained by non-specificity of experimental antibodies for these two histone modifications.

We reasoned that we might more generally exploit the subtle H3K4me3 signal elevation (whatever its underlying cause) as a potential marker also of H3K9me3 elevation in germline tissues by examining H3K4me3 in testes (*Pratto et al., 2014*). We performed de novo motif finding to identify PRDM9-independent 7-mers associated with testis H3K4me3 in THE1B repeats definitively not bound by PRDM9 (detailed in Appendix 2). This identified eighteen motifs significantly associated with non-PRDM9 H3K4me3 (after Bonferroni correction, *Figure 4a*). The motif ATCCATG remained the most strongly associated ($p<10^{-25}$), with eight other motifs clustered around it (*Figure 4a*). Confirming that these motifs also predict H3K9me3 levels, we observed almost perfect positive correlation ($r = 0.93$) between H3K4me3 signal strength in testes and H3K9me3 (as well as H3K4me3) in particular ROADMAP ESC lines (*Figure 4—figure supplement 1c*). Therefore, these 18 motifs predict both H3K9me3 and H3K4me3, broadly observable across somatic cells and (at least for the latter mark) testes also, and so we refer to this set as 'non-PRDM9 H3K9me3/H3K4me3 motifs.'

In addition to the top-scoring motif, ATCCATG, many or all of the remaining 17 non-PRDM9 H3K9me3/H3K4me3 motifs evidently impact meiotic recombination (*Figure 4—source data 1*; $p<0.00036$ for effect size correlation). All four of the non-PRDM9 recombination-influencing motifs we found overlap at least one of these 18 independently derived non-PRDM9 H3K9me3/H3K4me3 motifs (*Figure 4a*; note that power differences account for the smaller size of the former motif set). Summing these 18 motif influences to produce a score for each THE1B repeat using only its DNA sequence, we see more than a threefold difference in the probability of observing a recombination hotspot across PRDM9-bound THE1B copies between the top and bottom 10% quantiles of the score (*Figure 4d*). Given that we are only able to examine the region within each 1–2 kb recombination hotspot corresponding to the 354 bases of the THE1B element, this likely underestimates the true impact of local sequence on whether hotspots occur or not.

Notably, our testing for association with other histone-defined chromatin states (e.g. states enriched for H3K27me3) in ROADMAP-studied cell types identified many more sequence motifs. These included the known binding targets of two proteins, DUX4 and ZBTB33, that were previously shown to bind to THE1B elements, with DUX4 showing strong expression in testes (*Young et al., 2013*; *Wang et al., 2012*). However, only those motifs associated with heterochromatin and H3K9me3/H3K4me3 overlapped our non-PRDM9 recombination-influencing motifs. Thus, only a particular subset of chromatin modifications correspond to suppressed recombination, in THE1B repeats at least.

Overall, this analysis of thousands of human hotspots reveals that in cis, it is not simply PRDM9 binding that influences whether hotspots occur. Multiple sequence motifs exist that do not prevent PRDM9 binding, but instead modify the average amount of recombination that occurs *downstream* of binding, over two-fold for a single motif (ATCCATG). Given this diversity even within THE1B-centered hotspots, completely different motifs might operate to modulate recombination activity in other hotspots, either centered in different repeats or in non-repeat DNA. In contrast to this

complexity, examination of histone modifications reveals a common signature across recombination-influencing motifs, with strong alterations in the specific histone mark H3K9me3 and weaker signals for H3K4me3. This suggests that the mechanism of action across motifs might share fundamental similarities. Both H3K4me3 and H3K9me3 marks correlate negatively with recombination across all human hotspots (*Figure 4d*; *Figure 4—figure supplement 1b*), and reduced levels of non-PRDM9 H3K4me3 within hotspots has been observed in mice (*Brick et al., 2012*; *Davies et al., 2016*).

## KRAB-ZNF binding and TRIM28 recruitment predict low recombination near PRDM9 binding sites

The large class of human KRAB-ZNF genes represent an obvious set of motif-binding candidates that might explain H3K9me3 deposition within THE1B repeats and more broadly. In many such genes, the KRAB domain recruits TRIM28, which in turn recruits histone-modifying proteins including SETDB1, which lead to H3K9me3 deposition on nearby nucleosomes (*Schultz et al., 2002*; *Imbeault et al., 2017*). We therefore examined recent data measuring genome-wide binding of 222 KRAB-ZNF proteins in humans, and sites where TRIM28 is present in embryonic stem cells, for overlap with THE1B repeats (*Imbeault et al., 2017*; Appendix 2). Notably, although PRDM9 is a KRAB-ZNF protein, its KRAB domain does not interact with TRIM28 (*Imai et al., 2017*). We identified three KRAB-ZNF proteins (ZNF100, ZNF430 and ZNF766), as well as TRIM28, that are enriched for binding in THE1B repeats and also associate genome-wide with H3K9me3 deposition. We identified binding motifs for each of these four proteins within THE1B repeats. Strikingly, ATCCATG overlapped the second most significant motif for TRIM28 recruitment, and additional motif analysis for TRIM28 revealed a large (51 bp) motif, fully spanning a cluster of eight motifs associated with H3K9me3/H3K4me3 and recombination rate (*Figure 4a*), and presumably representing the binding target of one or more KRAB-ZNF protein(s) whose binding targets have not yet been experimentally characterized. The three ZNF proteins also all bind sites overlapping those implicated in impacting H3K9me3/H3K4me3 and meiotic recombination, two in the same region as the TRIM28 motif, but with differing sequence specificity (*Figure 4a*). Thus, while binding maps are not yet available for every human KRAB-ZNF protein, those that bind THE1B repeats consistently operate to reduce recombination, and TRIM28 recruitment can explain the strongest signals we see.

Across *all* our PRDM9 binding peaks (not only those in THE1B elements), 36.5% fall within 500 bp of a binding site of at least one of the KRAB-ZNF proteins with available data (*Imbeault et al., 2017*), suggesting that such repression might be important in regulating recombination more generally. To test this, we individually analyzed the KRAB-ZNF proteins with at least 30 instances of a KRAB-ZNF binding peak occurring near a PRDM9 binding peak (after excluding DNase HS regions and promoters, which are often bound by multiple different proteins), for their effect on whether a hotspot occurs at these PRDM9 binding peaks (Appendix 2). This revealed a universal negative trend (*Figure 4e*) typified by a twofold reduction in recombination locally at TRIM28-marked sites genome-wide, with every gene except one (ZNF282, which was non-significant) inferred to reduce hotspot odds. Binding of almost all KRAB-ZNF genes tested correlated positively with H3K9me3, and those genes with strongest H3K9me3 enrichment showed the strongest suppression of recombination locally (*Figure 4e*).

Together, our results indicate a mechanism of cis recombination repression affecting thousands of human PRDM9 binding sites. Binding of KRAB-ZNF proteins to specific sequence motifs within or nearby the PRDM9 binding site, followed by TRIM28 recruitment and H3K9me3 deposition, universally acts to strongly repress local recombination. Perhaps surprisingly, this can occur without preventing PRDM9 binding or H3K4me3 deposition. We suggest that this is the mechanism at play for the recombination-suppressing, H3K9me3-promoting ATCCATG motif, which we suspect is bound by a KRAB-ZNF protein whose binding sites have not yet been mapped. Many KRAB-ZNF genes bind to specific sets of retrotransposon repeats (THE1B repeats represent one example), so this repressive mechanism is likely to act to reduce recombination around many particular repeats.

## Genome-wide broad-scale rates vary independently of PRDM9 binding

Finally, we used our THE1B dataset to examine the relationship between PRDM9 binding and broad-scale recombination rates genome-wide while controlling for local genetic context. To do so, we partitioned THE1B repeats into quintiles of increasing recombination rate in the surrounding 1

Mb in males (independently measured by *Kong et al., 2002*). We observed that DMC1 enrichment increases >10-fold with surrounding recombination rate across both telomeric and non-telomeric regions, but H3K4me3 enrichment in testes, a proxy for meiotic PRDM9 binding, shows no association whatsoever (*Figure 4—figure supplement 2*). Therefore, in broad 'hotter' regions, double-strand breaks and crossovers occur at much higher frequencies, completely independently of the local sequence (which is similar in THE1B repeats genome-wide) or the local level of PRDM9 binding. This proves that, at least in human males, megabase-scale recombination rates throughout the genome are not associated with PRDM9's ability to bind and deposit H3K4me3, consistent with previous observations in the specific case of elevated human male recombination in telomeres (*Pratto et al., 2014*).

## Multimer formation is mediated primarily by the ZF array

Our results thus far have added to the already complex array of evolutionary forces buffeting *PRDM9*, relating to its ability to influence gene expression or to the co-binding of other zinc-finger proteins near its binding sites. Another dimension of evolutionary constraint may arise from PRDM9's ability to bind to itself and form functional multimers. Previous work has shown that PRDM9 as a whole can multimerize and that hetero-multimers of the human A and C alleles can bind the sequence targets of either allele and trimethylate surrounding histones (*Baker et al., 2015b*). However, it remains unknown which PRDM9 domain is responsible for this observed multimerization behavior. We sought to determine whether multimerization might involve PRDM9's ZF domain in any way, given other examples of ZF domains mediating protein-protein interactions (*McCarty et al., 2003*; *Lee et al., 2007*). To do so, we co-expressed PRDM9 constructs with different ZF domain properties and performed co-ImmunoPrecipitation (co-IP) experiments, thus extending our study from PRDM9's DNA-binding properties to its protein binding properties.

First, to confirm the ability of the PRDM9 alleles we study here to form multimers (*Baker et al., 2015b*), we performed co-IP experiments with full-length human B-allele PRDM9 constructs differentially tagged with HA and V5 epitopes and co-transfected into HEK293T cells. Following IP against the HA-tagged construct, we detected the V5-tagged construct very robustly; and conversely (*Figure 5—figure supplement 1*). This is consistent with human PRDM9 binding strongly to itself, as demonstrated previously in HEK293 cells (*Baker et al., 2015b*).

To narrow the PRDM9 domain(s) responsible for this self-binding behavior, we split the full-length human B-allele PRDM9 cDNA into two pieces: one containing only the C-terminal Zinc-Finger domain (the 'ZFonly' construct), and one containing everything else (the 'noZF' construct; illustrated in *Figure 5a*). We co-transfected these constructs and full-length PRDM9 into HEK293T cells in various combinations. The full-length human construct and the ZFonly construct localized to the nucleus, but the noZF construct localized throughout the cell, confirming a dominant role for the ZF domain in nuclear localization (*Figure 2—figure supplement 4*, *Collin et al., 2013*; *Wang et al., 2014*).

Interestingly, the ZF domain alone appears to be responsible for most of PRDM9's self-binding activity (*Figure 5b*). Following co-transfection of noZF-HA and noZF-V5, and despite very high expression levels visible in the input, only a very faint co-IP band is visible in the absence of the ZF array. Because the mock control lane is clean (*Figure 5—figure supplement 2a*), this band likely reflects a real but weak self-binding capability mediated by the non-ZF portion of PRDM9 (though we cannot rule out a role for the 'early zinc finger'). In complete contrast, we saw an intense co-IP band when co-transfecting ZFonly-HA with ZFonly-V5. Therefore, the zinc-finger domain of one PRDM9 protein can bind strongly to the zinc-finger domain of another, while the rest of the protein interacts more weakly.

We confirmed this result by co-transfecting full-length, V5-tagged human PRDM9 with either noZF-HA or ZFonly-HA, revealing that the ZFonly construct is sufficient to bind and pull down the full-length construct. This finding replicated in a repeat experiment, and when reversing the direction of the IP-western experiment (*Figure 5—figure supplement 2b*). No co-IP band is seen in a negative control experiment in which we co-transfected the noZF construct with the ZFonly construct (*Figure 5b*), ruling out an interaction between the ZF domain and the rest of PRDM9 or any interaction between the epitope tags used. Our results remained unchanged following complete DNA digestion by benzonase in the ZFonly-ZFonly co-IP experiment (*Figure 5—figure supplement 3a*), implying that DNA is not required for the observed interaction between ZF domains.

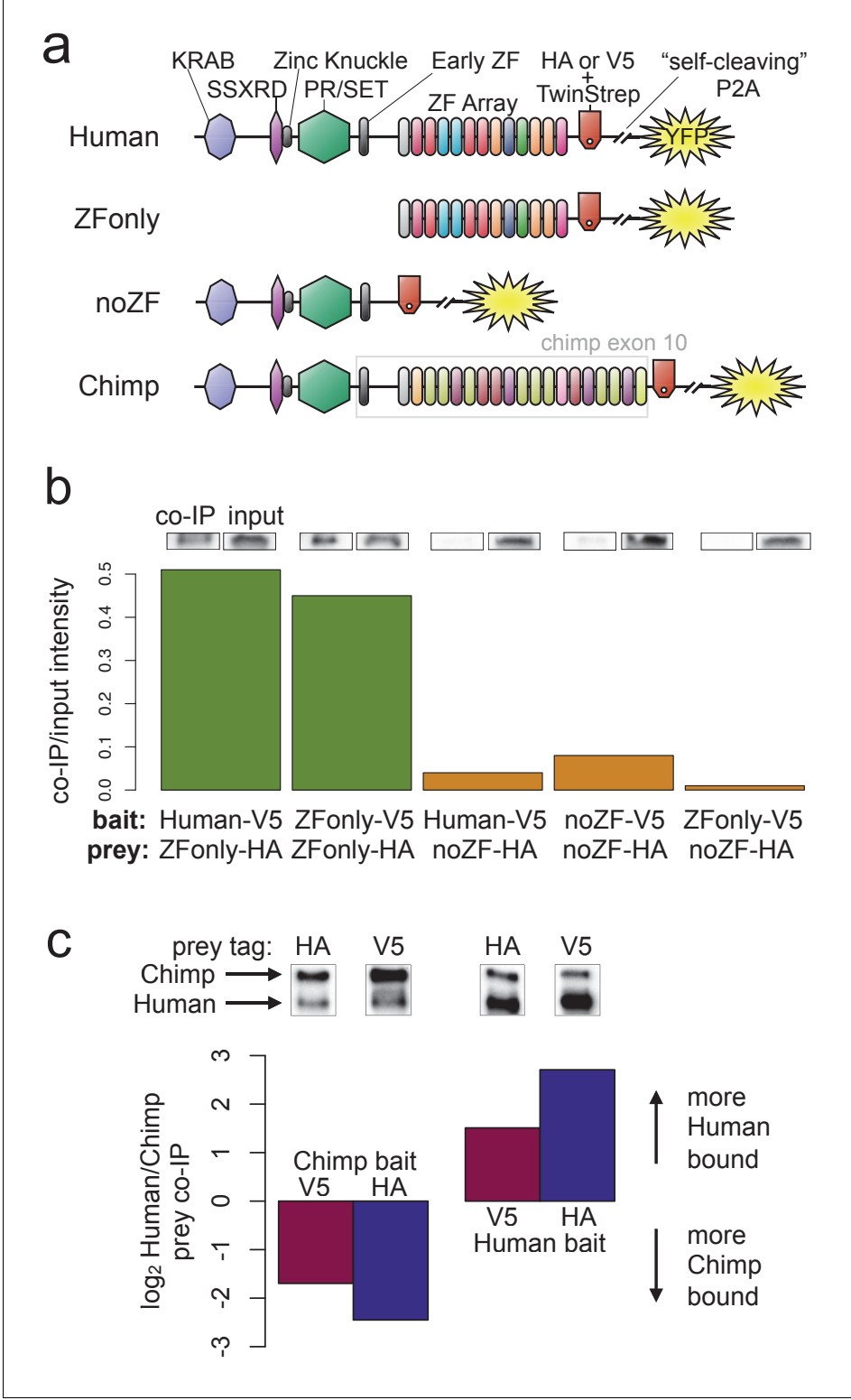

**Figure 5.** PRDM9 multimer formation is mediated by the ZF domain in an allele-biased manner. (**a**) Overview of the different C-terminally tagged PRDM9 constructs used. Both an HA and a V5 version of each construct were generated for co-IP experiments. (**b**) Barplot showing the relative intensity of western blot co-IP bands normalized to input bands (from 50-μg of total lysate protein) for each combination of bait and prey constructs. Whenever both bait and prey contain the zinc-finger domain (green bars), the co-IP signal is much stronger than when either or both constructs lack a ZF domain (orange bars). See *Figure 5—figure supplements 1* and *2* for complete

*Figure 5 continued on next page*

*Figure 5 continued*

westerns with mock controls. (**c**) Barplot showing the results of competitive co-IP experiments performed in cells transfected with both Human and Chimp as prey (with the same epitope tag) and either Human or Chimp as bait (with a complementary epitope tag). Bars indicate the relative co-IP band intensity for Human and Chimp prey constructs when pulled down with either Chimp or Human bait. When Human is used as bait, more Human prey is pulled down; when Chimp is used as bait, more Chimp prey is pulled down (and this holds for both directions of HA/V5 tagging).

DOI: https://doi.org/10.7554/eLife.28383.023

The following figure supplements are available for figure 5:

**Figure supplement 1.** Confirmation of PRDM9 multimer formation.
DOI: https://doi.org/10.7554/eLife.28383.024

**Figure supplement 2.** Multimerization is mediated primarily by ZF-ZF binding.
DOI: https://doi.org/10.7554/eLife.28383.025

**Figure supplement 3.** Benzonase treatment does not affect co-IP results.
DOI: https://doi.org/10.7554/eLife.28383.026

## Hetero-multimers of divergent ZF arrays form less efficiently

Finally, to examine the specificity of ZF array binding, we replaced the final exon containing the human ZF array with a synthesized cDNA matching the final exon of the chimpanzee reference PRDM9 allele (w11a) containing 18 zinc fingers (compared to 12 in the human allele, allowing us to resolve them as two distinct bands), and with different DNA-binding preferences. We refer to the resulting tagged constructs as Chimp-HA and Chimp-V5 (*Figure 5a*). To test the relative efficiency of homo- versus hetero-multimerization, we performed direct competition experiments. We transfected cells with three constructs: for example, Chimp-V5 plus Chimp-HA plus Human-HA. In this case Chimp-V5 would be the 'bait' pulled down by IP with anti-V5, and Chimp-HA and Human-HA would be the co-IP 'prey' detected by western blotting with anti-HA (we replicated by reversing the tags). The results show that Chimp PRDM9 pulls down Chimp PRDM9 more than twofold more efficiently than it pulls down Human PRDM9. Similarly, Human PRDM9 pulls down Human PRDM9 more than twofold more efficiently than it pulls down Chimp PRDM9 (*Figure 5c*). Thus, PRDM9 preferentially forms homo-multimers rather than hetero-multimers, at least for ZF arrays as highly diverged as Human and Chimp. These findings replicated after completely digesting DNA with benzonase (*Figure 5—figure supplement 3*). Because chimp and human PRDM9 ChIP-seq peaks almost never overlap (*Figure 2—figure supplement 1b*), we can rule out the possibility that heteromultimer formation between these two alleles results from co-binding to short DNA fragments that may be protected from benzonase digestion by PRDM9. That is, these results also confirm that PRDM9 multimer formation must be mediated by protein-protein interactions, not by protein-DNA interactions, though we still cannot formally rule out a role for DNA in enhancing this protein-protein interaction.

## Discussion

The extremely rapid evolution of PRDM9's zinc fingers, both within and between species, is one of the most striking features of this remarkable protein. Our results imply that over and above their role in positioning recombination sites and a role in chromosome synapsis (*Davies et al., 2016*), several other factors might influence this evolution. We showed here that PRDM9's zinc-finger domain can impact its ability to form multimers, its ability to activate gene expression, and its ability to initiate recombination, in particular if it binds near promoters or near targets of other zinc-finger proteins.

PRDM9's zinc-finger array has been regarded primarily as a DNA-binding domain with no other demonstrated functions, although studies of other zinc-finger proteins have shown that ZF domains can participate in highly specific protein-protein interactions, including with each other (*McCarty et al., 2003*; *Lee et al., 2007*). The mammalian gene with the most similar ZF-array to PRDM9 is ZNF133, whose zinc fingers have an almost identical consensus sequence, apart from at DNA-contacting bases, to PRDM9. ZNF133 has been shown to interact with PIAS1 (which interestingly is recruited to DNA damage sites; *Galanty et al., 2009*) via its zinc fingers, which can simultaneously bind its protein and DNA targets (*Lee et al., 2007*). Thus, it seems credible that

multimerization interactions involving PRDM9 might involve its zinc fingers, and it further seems plausible that PRDM9's zinc-finger domain might be able to mediate interactions with other proteins. Currently, we can only speculate about what function PRDM9 multimerization might serve if it occurs in meiosis. If biased multimerization occurs in vivo between different PRDM9 alleles (mediated by their variable zinc-finger domains), it could have important meiotic impacts in PRDM9 heterozygotes, although further study is needed, for example to determine if hetero-multimers form less efficiently between the human A, B and C alleles. Together with binding affinity differences, variable hetero-multimerization might impact PRDM9 dominance patterns, and dominance over less advantageous existing alleles could further increase the evolutionary advantage enjoyed by some newly arising alleles (*Baker et al., 2015b*) or potentially play a role in the dosage sensitivity of PRDM9 in causing hybrid infertility in mice (*Flachs et al., 2012*; *Ségurel et al., 2011*). One intriguing hypothesis is that multimer formation may play some role in PRDM9-mediated homologue pairing, which we previously identified as a potential mechanism to explain the role of PRDM9 in fertility and speciation in mice (*Davies et al., 2016*). In this case, a preference for homo-multimer formation would have obvious advantages.

Our results also highlight the key impact of zinc-finger variation on PRDM9 binding at both fine and broad scales. We observed no fewer than seven different modes of human PRDM9 binding with different internal spacings between several DNA-contacting zinc fingers (*Figure 1a*), a pattern not detected in previous studies. Binding is strongly impacted by all zinc fingers—as we observed in THE1B repeats and has been previously shown for mouse alleles (*Billings et al., 2013*)—and involves extensive sequence specificity not captured by a single shared motif. However, the chimpanzee w11a PRDM9 allele binds differently not only at fine scales but also broad scales (*Figure 2—figure supplement 1*) and avoids promoters. Similarly, a recent study in mice (*Grey et al., 2017*) found that two mouse PRDM9 alleles do not directly bind at promoters. When Spo11 was present to form DSBs, additional PRDM9 peaks appeared at a small number of promoters—hypothesized as due to indirect recruitment (*Grey et al., 2017*). An earlier study in mice with AT-rich PRDM9 binding motifs suggested that PRDM9 may direct recombination away from promoters by depositing competitive H3K4me3 marks (*Brick et al., 2012*).

In contrast to these alleles in chimp and mouse, we observed human PRDM9 directly binding to many promoter regions, previously unobserved due to filtering of PRDM9-independent H3K4me3 peaks and the evident suppression of DSB formation at these sites (*Pratto et al., 2014*; *Baker et al., 2015b*). Given the similarity of promoter composition and organization across cell types, the human A/B alleles likely bind to promoters in vivo as well, although we cannot exclude the possibility that such binding is prevented somehow, and further study will need to determine the promoter affinities of other human PRDM9 alleles. Our results imply that the suppression of recombination at promoters (including those that we show are bound by PRDM9) cannot simply be due to PRDM9 binding away from promoters. Interestingly, PRDM9 deposits less H3K36me3 at promoters compared to non-promoters, particularly at promoters with higher levels of PRDM9-independent H3K4me3 (*Figure 2*). We speculate that, if the co-occurrence of the H3K4me3 and H3K36me3 marks is essential for recombination initiation (as suggested by *Powers et al., 2016*; *Yamada et al., 2017*), then the relative lack of H3K36me3 at PRDM9-bound promoters could explain why these binding sites fail to initiate recombination. Of course, this does not explain why recombination tends toward promoters in the absence of PRDM9, be it in knockout mice (*Brick et al., 2012*) or lineages that have lost *PRDM9* (*Baker et al., 2017*), such as dogs (*Auton et al., 2013*). Together with the discovery of a fertile woman with two nonfunctional copies of *PRDM9* (*Narasimhan et al., 2016*), these results highlight the unresolved complexity surrounding PRDM9's role in meiosis.

Adding to this complexity is our finding that PRDM9 can influence the transcriptional activity of a subset of bound genes, such as the spermatogenesis-specific *CTCFL* and *VCX* genes, in transfected HEK293T cells. Speculatively, this pleiotropic effect may even help to explain why a single *PRDM9* allele predominates in many human populations. That is, while a multitude of alleles may function equally well in specifying sites of meiotic recombination initiation, perhaps a subset can positively affect fertility by binding to and enhancing the expression of meiotic genes such as *CTCFL*, and these alleles are consequently driven to high frequency by positive selection. We also observed that a predicted submotif shared by many western chimp *PRDM9* alleles (*Schwartz et al., 2014*) corresponds precisely to a group of chimp zinc fingers with the strongest influence on binding targets (*Figure 2—figure supplement 1c*), similar to the prior observation of a group of 'C-type' human

PRDM9 alleles that are diverse overall, but again overlap in the region identified to most strongly influence binding (*Hinch et al., 2011*; *Berg et al., 2011*; *Pratto et al., 2014*). This apparent sharing of binding specificities between alleles could potentially be driven by PRDM9's effects on transcription, its propensity to form multimers, and/or its ability to bind symmetrically to homologous chromosomes in heterozygotes (*Davies et al., 2016*). Further work will need to explore the extent to which these behaviors are functionally important in vivo.

Aside from recombination suppression at promoters, our results shed light on an additional level of recombination regulation occurring downstream of PRDM9 binding. Sequence-specific binding by the large collection of KRAB-ZNF genes is associated with localized recombination suppression at scales >1 kb, without suppressing nearby PRDM9 binding, or H3K4me3 deposition, either in transfected cells (this study) or in testes (*Pratto et al., 2014*, *Figure 4e*). This implies that hundreds of motifs exist that mark sites of local recombination suppression. In contrast, we observe no impact of the presence/absence of binding sites for proteins such as DUX4 (*Young et al., 2013*) on recombination, despite our observing clear effects of the DUX4 binding motif on local chromatin marks (*Figure 4—source data 1*). Instead, perhaps only certain chromatin modifications suppress recombination. At their binding sites, many KRAB-ZNF proteins recruit TRIM28 which in turn recruits histone remodeling proteins including SETDB1 and HP1, depositing the H3K9me3 modification (*Schultz et al., 2002*; *Imbeault et al., 2017*), which has been associated with suppression of meiotic recombination in mice (*Buard et al., 2009*; *Walker et al., 2015*; *Yamada et al., 2017*). It has been suggested that KRAB-ZNF-induced heterochromatin may serve to stabilize repetitive sequences by preventing non-allelic homologous recombination (NAHR) (*Vogel et al., 2006*; *Iyengar et al., 2011*). Furthermore, PRDM9 has been shown to interact with both readers and writers of H3K9me3 (*Parvanov et al., 2017*). Interestingly, we also saw a weak increase in H3K4me3 signal whenever H3K9me3 increased, and this signal is also observed in testes, implying the motifs we find can impact chromatin modifications in this tissue, and—unlike PRDM9—in many somatic cell types also.

Most KRAB-ZNF proteins bind repeats, and they constitute the largest family of transcription factors in mammals, with rapid evolution (*Imbeault et al., 2017*). Evidence suggests that the KRAB domain may have first evolved in an ancient ancestor of PRDM9 and then spread (*Birtle and Ponting, 2006*), so it is interesting that these partial descendants of PRDM9 appear to disrupt meiotic recombination. In general, KRAB-ZNF genes appear to emerge concomitantly with the spread of particular transposon families, and they play a role in repressing transposon activity (*Imbeault et al., 2017*; *Jacobs et al., 2014*; *Wolf et al., 2015*; *Rowe et al., 2013*). Paradoxically though, they often remain active long after their targets lose transpositional activity (*Imbeault et al., 2017*). Our results suggest that one possible reason might be an adaptive role for KRAB-ZNF genes in specifically suppressing meiotic recombination in and around repeats, which otherwise could be prone to mediating deleterious genomic rearrangements (as proposed by *Zamudio et al., 2015* regarding DNA methylation at transposons). If so, evolution of PRDM9 to bind new repeats might, in turn, lead to co-evolution of ZNF genes to suppress meiotic recombination at a subset of those repeats. We note that the meiotic effects of KRAB-ZNF proteins might be apparent even if they are not expressed in meiotic cells, as their chromatin marks might be transmitted epigenetically from precursor cells (*Rowe et al., 2013*). However, previous work has shown that KRAB-ZNF co-repressors are essential for normal gametogenesis in mice. Namely, the H3K9me3 methyltransferase SETDB1 is required to silence endogenous retroviruses in mouse primordial germ cells (*Liu et al., 2014*), and germline knockout of TRIM28 leads to sterility (*Weber et al., 2002*). Further study will need to determine which, if any, KRAB-ZNF proteins are active in human meiotic cells.

Another consequence of KRAB-ZNF-mediated meiotic recombination suppression is that not only PRDM9 binding sites, but potentially many other sites within hotspots, are predicted to cause DSB initiation asymmetry, and thus are likely to be subject to biased transmission—as seen previously for PRDM9 motifs and GC-biased gene conversion in hotspots (*Boulton et al., 1997*; *Coop and Myers, 2007*; *Myers et al., 2010*; *Baker et al., 2015a*; *Smagulova et al., 2016*; *Davies et al., 2016*). Unlike self-destructive drive at PRDM9 motifs, such drive would bias the evolution of features with broad impacts across cell types, towards *increased* KRAB-ZNF binding and hence constitutive silencing of hotspot regions, even if this silencing is selectively disadvantageous. Recent work by (*Yamada et al., 2017*) has demonstrated that as many as a third of meiotic DSBs occur within repetitive sequences in B6 mice, although DSB frequencies vary substantially among different classes of repeats, with most classes being depleted for DSBs. The authors hypothesize that PRDM9 may evolve to target

transposons for meiotic recombination so that the effects of hotspot death will rapidly inactivate them by driving mutations or deletions of the PRDM9 binding site to fixation (and this advantage might compensate for the risk of NAHR at those repeats; *Yamada et al., 2017*). Our work suggests that PRDM9 binding to transposable elements might also inactivate them in a second way: by accelerating their evolution towards constitutive silencing by KRAB-ZNF proteins. In this model, hotspot self-destructive drive would be mirrored by the rapid accumulation of new KRAB-ZNF binding sites within PRDM9-bound transposable elements—a prediction that should be examined empirically by future studies. On the other hand, given strong DSB suppression at promoters, nearby PRDM9 binding sites might be immune from the effects of hotspot death, which would otherwise act to abolish its binding and drive potentially deleterious mutations—including any which might weaken the promoter—to fixation in these regions. Indeed, the potentially destructive or repressive effects of hotspot death could explain why meiotic recombination is directed away from functional elements like promoters, and towards deleterious elements like transposons, at least in humans and mice.

## Materials and methods

### Cloning

A cDNA was custom synthesized to contain the full-length (2,685 bp) *PRDM9* transcript from the human reference genome (GRCh37), which is the B allele of *PRDM9*. 218 synonymous base changes were engineered into the exon containing the zinc-finger domain in order to distinguish the synthetic copy of *PRDM9* from the endogenous copy and to facilitate proper synthesis of this highly repetitive region. We cloned this cDNA into the pLEXm transient expression vector (*Aricescu et al., 2006*) by ligation with a Venus (YFP) tag at its N-terminus, fused using an AgeI restriction site. A similar synthesized construct was designed to match exon 10 of the chimp PRDM9 reference allele (the 'w11a' allele, 2,022 bp, codon optimized for human expression and non-repetitiveness). Exons 1–9 were amplified from the human construct, and the chimp allele was fused at the N-terminus with an XbaI site. The ZFonly and noZF alleles were amplified using internal primers designed inside the full-length human construct. For the C-terminally tagged constructs, a 198 bp HA and 213 bp V5 linker were synthesized (having the sequence linker-TwinStrep-linker-HA/V5-linker-P2A) and cloned between each respective PRDM9 allele and a YFP tag using KpnI and AgeI sites, respectively. C-terminally tagged constructs were cloned into the pLENTI CMV/TO Puro DEST vector (Addgene plasmid # 17293; *Campeau et al., 2009*), owing to its higher transient expression efficiency and to test the possibility of stable lentiviral transduction. Cloning into this vector was performed using the Gateway recombinase-based cloning system (Thermo Fisher Scientific, Waltham, MA). Constructs were cloned, amplified, and isolated using an Qiagen (Germany) EndoFree Plasmid Giga Kit to yield transfection-quality DNA, which was verified by restriction digestion and Sanger sequencing.

### Tissue culture and transfection

HEK293T cells were chosen owing to their high transfection efficiency, rapid growth rate, and low-cost media requirements. Cells were purchased directly from the ATCC (ATCC CRL-3216; RRID: CVCL_0063), with a certificate of analysis confirming cell line identity by Short Tandem Repeat profiling and confirming lack of mycoplasma contamination. All experiments were carried out on cells cultured for less than five passages from the purchased stock reference strain. Large-scale transfections of the N-terminal GFP-tagged Human PRDM9 construct were performed as described (*Aricescu et al., 2006*). Cells were grown in DMEM media (10% FCS, 1X NEAA, 2 mM L-Glut, Sigma D6546; Millipore Sigma, Burlington, MA) in 200 ml roller bottles at 37°C/5% $CO_2$. A transfection cocktail was prepared for each bottle by adding 0.5 mg of chloroform-purified construct DNA to 50 ml of serum-free DMEM (1X NEAA, 2 mM L-glut) and 1 mg polyethylenimine, followed by a 10 min incubation, and then addition of 375 μg of kifunensine. After the cells reached 75% confluence, the growth medium was removed from each roller bottle and replaced with 200 ml low-serum DMEM (2% FCS, 1X NEAA, 2 mM L-Glut) and 50 ml transfection cocktail. Cells were then incubated for 72 hr to enable expression of the transfected construct. Expression was verified by fluorescence microscopy, and we consistently observed visible fluorescence in at least 50% of cells for all samples prior to harvesting.

We performed all subsequent smaller-scale transfections of the C-terminally tagged constructs in the pLENTI vector using the FuGENE-HD transfection reagent according to manufacturer instructions (Promega, Madison, WI). HEK293T cells (ATCC CRL-3216; RRID:CVCL_0063) were thawed and incubated at 37°C with 5% $CO_2$ in DMEM (Sigma D6546) supplemented with 10% fetal bovine serum (Sigma F7524), 1X L-Glutamine (Sigma G7513), and 1X penicillin/streptomycin (Sigma P0781). The night before transfection, confluent cells were trypsinized (Sigma T3924), diluted in growth medium, and counted on an automatic hemocytometer (Bio-Rad TC20, Hercules, CA). For each replicate, 15 million cells were seeded in 30 ml growth medium in a T175 cell culture flask. The following morning, cells were transfected by mixing 30 µg total construct DNA into 800 µl OPTI-MEM (Thermo Fisher Scientific 31985062), then carefully adding 90 µl FuGENE-HD Transfection Reagent and flicking to mix, incubating at room temperature for 15 min, and then adding the mixture dropwise to each dish while swirling gently to mix. After 48 hr, cells were imaged briefly with a fluorescent microscope to confirm expression (and transfection efficiency >50%), and were subsequently harvested. As negative controls, additional cells were seeded at the same time but were not transfected.

## ChIP (N-terminal YFP-Human)

ChIP-seq was performed according to an online protocol produced by Rick Myers's laboratory (*Johnson et al., 2007*), which was used to produce much of the ENCODE Project's ChIP-seq data (*ENCODE Project Consortium, 2012*), with several optimizing modifications.

### Crosslinking

Bottles were removed from the incubator and shaken vigorously to detach cells. Fresh formaldehyde was added to a final concentration of 0.75% and cells were incubated at room temperature for 15 min. The crosslinking reaction was stopped by adding glycine to a final concentration of 125 mM. Cells were aliquoted to 50 ml conical tubes, centrifuged ($2000g$, 5 min), resuspended in cold 1X PBS, and centrifuged again. Pellets were snap frozen with dry ice, and then stored at −80°C.

### Lysis and Sonication

Frozen pellets were thawed and resuspended in cold Farnham Lysis Buffer (5 mM PIPES pH 8.0, 85 mM KCl, 0.5% NP-40, one tablet Roche Complete protease inhibitor per 50 ml; Roche, Switzerland) to a concentration of 20 million cells per ml, then passed through a 22G needle 20 times to further lyse and homogenize them. Technical replicates were processed in parallel from this point forward (with only one replicate performed for transfected H3K4me3). Lysates were centrifuged and resuspended in 300 µl cold RIPA lysis buffer (1X PBS, 1% NP-40, 0.5% sodium deoxycholate, 0.1% SDS, one tablet Roche Complete protease inhibitor per 50 ml) per 20 million cells to lyse nuclei. 300 µl samples were sonicated in a Bioruptor Twin sonication bath (Diagenode, Denville, NJ) in 1.5 ml Eppendorf tubes at 4°C for two 10 min periods of 30 s on, 30 s off at high power. Cell debris was removed by centrifugation (14,000 rpm, 15 min, 4°C), and supernatants were isolated and brought to a final volume of 1 ml with RIPA. These chromatin preps were snap-frozen in dry ice then stored at −80°C.

### Immunoprecipitation

Magnetic beads were washed by adding 200 µl Invitrogen Sheep Anti-Rabbit Dynabeads (Thermo Fisher Scientific) per sample to 800 µl cold PBS/BSA (1X PBS, 5 mg/ml BSA, one tablet Roche Complete protease inhibitor per 50 ml, filtered with 0.45 micron filter). Solutions were placed on a magnetic rack and resuspended in 1 ml PBS/BSA four times. 5 µl Abcam (United Kingdom) rabbit polyclonal ChIP-grade anti-GFP antibody (ab290; RRID:AB_303395) or rabbit polyclonal ChIP-grade anti-H3K4me3 antibody (ab8580; RRID:AB_306649) was added and solutions were incubated overnight at 4°C on a rotator. Antibody-coupled beads were washed three times with cold PBS/BSA and resuspended in 100 µl PBS/BSA, then added to 1 ml chromatin preps thawed on ice. One tube was prepared in parallel without adding beads, to yield a genomic background control sample from total chromatin. Tubes were incubated for 12 hr on a rotator at 4°C, then washed 5 times for 3 min each with cold LiCl Wash Buffer (100 mM Tris pH 7.5, 500 mM LiCl, 1% NP-40, 1% sodium deoxycholate, filtered with a 0.45 micron filter unit), then washed once with cold 1X TE (10 mM Tris-HCl pH 7.5, 0.1

mM Na$_2$-EDTA). Bead pellets were resuspended in 200 µl room-temperature IP elution buffer (1% SDS, 0.1 M NaHCO$_3$, filtered with a 0.45 micron filter unit) and vortexed to mix.

## Reverse crosslinking and DNA purification

Samples were incubated in a 65°C water bath for 1 hr with mixing at 15 min intervals to uncouple beads from protein-DNA complexes. Samples were centrifuged (14,000 rpm, 3 min) and placed on a magnet to pellet beads, and supernatants were isolated and then incubated in a 65°C water bath overnight to reverse crosslinks. DNA was purified using a Qiagen MinElute reaction cleanup kit and quantified using a Qubit High Sensitivity DNA kit (Thermo Fisher Scientific).

## ChIP (C-terminal-tagged constructs)

Slight modifications were made for the smaller-scale transfection experiments with C-terminally tagged constructs. Crosslinking was performed in 1% formaldehyde for 5 min. Input chromatin was 'pre-cleared' to remove chromatin bound non-specifically by the beads. For each sample, 50 µl of equilibrated magnetic beads were resuspended in 100 µl PBS/BSA and added to the chromatin samples for pre-clearing for two hours at 4°C with rotation. Beads were removed, and 100 µl of pre-cleared chromatin was set aside for the input control. 5 µl ChIP-grade rabbit polyclonal antibody (Abcam anti-HA ab9110 RRID:AB_307019, anti-V5 ab9116 RRID:AB_307024, anti-H3K4me3 ab8580 RRID:AB_306649, or anti-H3K36me3 ab9050 RRID:AB_306966) was added to the remaining pre-cleared chromatin and incubated overnight at 4°C with rotation. 50 $\mu$l beads were washed and resuspended as before, then incubated with the chromatin samples for 2 hr at 4°C with rotation. After washing and decrosslinking, samples were further incubated with 80 µg RNAse A at 37°C for 60 min and then with 80 µg Proteinase K at 55°C for 90 min.

## ChIP sequencing, mapping, and filtering

DNA was submitted to the Oxford Genomics Centre for library preparation, sequencing, and mapping. For the N-terminal YFP-Human experiments, ChIP and input chromatin DNA samples from transfected and untransfected cells were sequenced in multiplexed paired-end Illumina (San Diego, CA) HiSeq1000 libraries, yielding 51 bp reads. Samples from transfected cells were multiplexed across 3 lanes, yielding roughly 77–101 million properly mapped read pairs (i.e. fragments) per replicate. Samples from untransfected cells (processed independently) were multiplexed across 2 lanes, yielding roughly 60–99 million properly mapped fragments per sample. For the C-terminal tag experiments, ChIP and input chromatin DNA samples from transfected and untransfected cells were sequenced all together in 6 lanes of paired-end Illumina HiSeq2500 libraries (rapid mode), yielding 51 bp reads with 37 to 64 million reads per replicate. Coverage was chosen in each experiment to exceed recommendations for doing ChIP-seq with sufficient power to detect the majority of true binding events (*Landt et al., 2012*).

Sequencing reads were aligned to hg19 using BWA (v0.7.0-r313, option -q 10, *Li and Durbin, 2009*, RRID:SCR_010910) followed by Stampy (v1.0.23-r2059, option -bamkeepgoodreads, *Lunter and Goodson, 2011*, RRID:SCR_005504), and reads not mapped in a proper pair or with an insert size larger than 10 kb were removed. Read pairs representing likely PCR duplicates were also removed by samtools rmdup (v0.1.19–44428 cd, *Li et al., 2009*, RRID:SCR_002105). Pairs for which neither read had a mapping quality score greater than 0 were removed. For samples with only one replicate, fragments were split at random into two equally-sized pseudo-replicates. Fragment coverage from each replicate was then computed at each position in the genome using in-house code and the samtools (v0.1.19–44428 cd, RRID:SCR_002105) and bedtools (v2.23.0, genomecov -d, RRID:SCR_006646) packages (*Li et al., 2009*; *Quinlan and Hall, 2010*). Visualization (producing browser screengrabs) was done using the WashU Epigenome Browser (*Zhou et al., 2011*, RRID: SCR_006208). Details of the ChIP-seq samples are listed in *Figure 1—source data 1*. Our peak calling algorithm is fully described in Appendix 1.

We compared the C-terminal Human-HA/V5 data with the N-terminal YFP-Human data and found strong overlap between the peak sets (60%) but a poor correlation in raw coverage values or in our computed enrichment values (r = 0.3). We explored this further and noticed that the newer sequencing run had a strong increase in coverage of GC-rich regions (nearly two-fold higher input coverage in regions with >60% GC), perhaps owing to differences in the ChIP protocol or to downstream

differences in the library prep and sequencing steps (Illumina HiSeq 1000 versus Illumina HiSeq 2500). We also cannot exclude any effects due to the different placement of the tags. Due to this strong GC bias, we utilized the N-terminal YFP-Human dataset exclusively for most analyses of the human allele, except when directly comparing to data obtained using the C-terminal Human-HA/V5 constructs (ATAC-seq, RNA-seq, H3K36me3 ChIP-seq, Chimp ChIP-seq).

## Overlap correction

When comparing peak sets to determine overlap proportions, one must account for chance overlaps owing to the width and number of peaks being compared. For comparisons between single-base peak centers and DSB hotspot intervals, for example, we computed the expected number of chance overlaps $c$ between the $n$ peak centers and the $t$ hotspot intervals, each with width $w_i$, in a genome of size $g$ as

$$c = \sum_{i \in t} \left( 1 - \left( \frac{g - w_i}{g} \right)^n \right).$$  (1)

For more complicated comparisons, for example between two sets of intervals, we computed chance overlaps by randomly shifting the positions of one set of intervals uniformly in the interval $[-60000, 60000]$, then counted the resulting overlaps to estimate $c$.

Given $f$ observed overlaps between the sets of $n$ and $t$ peaks, we can compute the corrected overlap fraction, $o/t$ as follows. Let $o/t$ be the proportion of systematic overlaps, $c/t$ be the fraction of chance overlaps, and $f/t$ be the proportion of total overlaps. The probability of no overlap is simply the product of the complements of chance and systematic overlaps, as follows:

$$(1 - f/t) = (1 - o/t)(1 - c/t).$$

Solving for $o/t$ then yields:

$$o/t = 1 - \frac{1 - f/t}{1 - c/t}.$$  (2)

Note that this method is only suitable when the number of chance overlaps is smaller than the number of total overlaps.

## Motif finding

For each peak, a 300 bp sequence (centered on the called peak center) was extracted from the reference sequence (hg19). Ab initio motif calling was performed on sequences from the top 5,000 peaks (ranked by enrichment) that passed a set of stringent filters (p<$10^{-10}$, enrichment >2, C.I. width ≤50, no bases overlapping annotated repeats, number of input reads between 10%ile and 90%ile, and ≥30 reads from ChIP rep1 + ChIP rep2). Motif calling proceeded in two stages: seeding motif identification, and joint motif refinement. Each seeding motif was obtained by first counting all 10-mers present in all input sequences, and from the top 50 most frequently occurring 10-mers, the one with the greatest over-representation in the central 100 bp of each peak sequence was chosen. This seeding 10-mer was then refined for 100 iterations as described in (*Davies et al., 2016*), and all peak sequences containing matches to this refined motif were removed. From the remaining sequences, a new 10-mer was found and refined into a seeding motif, and this process was iterated up to 20 times. The 20 resulting seeding motifs were then refined jointly for 200 iterations as described (*Davies et al., 2016*). Three separate runs were performed for each sample to verify consensus. For the YFP-Human peaks, a run producing 17 final motifs was chosen, and of these the 7 motifs with ≥85% of matches occurring in the central 100 bp of each peak sequence were chosen as the final set in order to remove degenerate motifs (i.e. those with little base specificity at any position) as well as likely false positives (such as a match to the motif for the AP1 transcription factor). For the Chimp-HA/V5 peaks, only two motifs were produced, one of which was a degenerate CT-rich motif found in only 10% of peaks (but not centrally enriched), so it was filtered out (not shown). These final motifs were then force-called on the full set of peaks (without any peak filtering) by rerunning the refinement algorithm (*Davies et al., 2016*) with the option to not update the motifs with each iteration. The motif with the greatest posterior probability (of at least 0.75) of a match was

reported for each peak, along with position and strand. For identifying motif matches genome wide, we used FIMO (version 4.10.0; *Bailey et al., 2015*).

## ATAC-seq

ATAC libraries were prepared as described (*Buenrostro et al., 2013*). Briefly, 50,000 cells were lysed in 10 mM Tris-HCl pH 7.4, 10 mM NaCl, 3 mM MgCl$_2$, 0.1% IGEPAL CA-630 and the nuclei were pelleted at 500$g$ for 10 min. The transposition reaction was carried out for 30 min at 37°C using the Nextera DNA Sample Preparation Kit (Illumina) according to the manufacturer's instructions. The libraries were purified using the MinElute PCR Purification Kit (Qiagen), PCR amplified, multiplexed, and sequenced by the Oxford Genomics Centre on an Illumina HiSeq2500 (rapid mode) to produce 60–77 million sequenced fragments (51 bp, paired-end reads) per sample. Reads were mapped to the hs37d5 reference (*Abecasis et al., 2012*) using BWA (v0.7.0-r313, *Li and Durbin, 2009*) followed by Stampy (v1.0.23-r2059, with option –bamkeepgoodreads, *Lunter and Goodson, 2011*). PCR duplicates, mtDNA-mapped reads, reads not mapped in a proper pair, reads with mapping quality equal to 0, and pairs with an insert size larger than 2 kb were removed using samtools (v0.1.19–44428 cd, *Li et al., 2009*), leaving ~11 million fragments per sample. Using in-house code, fragments were split by size into inter-nucleosome (51–100 bp) and mono-nucleosome fragments (180–247 bp), and the position of the central base in each fragment was reported, as described (*Buenrostro et al., 2013*). This yielded ~1 million inter-nucleosome and ~3 million mono-nucleosome fragments per sample. Fragment center coverage was computed genome-wide using bedtools (*Quinlan and Hall, 2010*).

## RNA extraction and RT-qPCR

Total RNA was extracted using the RNeasy kit (Qiagen) from three biological replicates (independently transfected in separate wells in parallel) per sample. For quantitative PCR analysis, RNA was reverse-transcribed using Expand Reverse Transcriptase (Roche), according to the manufacturer's instructions. qPCR reactions were carried out in duplicate for each sample using Fast SYBR Green Master Mix (Applied Biosystems, Foster City, CA) on a CFX real-time C1000 thermal cycler (Bio-Rad), following the manufacturer's guidelines. Data were analyzed using the CFX 2.1 Manager software (Bio-Rad) and normalized to the Tata binding protein (*TBP*) gene. Relative gene expression levels were calculated using the $\Delta\Delta C_t$ method, after averaging the two technical replicates for each sample. Statistical analysis was carried using a one-tailed t test. Primer sequences (from *Hines et al., 2010* and *Lahn and Page, 2000*) and Ct values are given in *Figure 3—source data 1*.

## RNA-seq

Total RNA was submitted to the Oxford Genomics Centre for mRNA enrichment, library preparation, and sequencing. Samples were multiplexed and sequenced on an Illumina Hi-Seq2500 (rapid mode), yielding 71–98 million 51 bp read pairs per sample. We created a custom reference sequence by merging the hs37d5 reference (used by the 1000 Genomes Project to improve mapping quality *Abecasis et al., 2012*) with the construct and vector sequences transfected into our cells. Data were analyzed using the Tuxedo software package (*Trapnell et al., 2012*). Reads were mapped and processed using TopHat (version 2.0.13, options –mate-inner-dist=250 –mate-std-dev 80 –transcriptome-index = Ensembl.GRCh37.genes.gtf, RRID:SCR_013035); followed by Cufflinks, CuffQuant, and Cuff-Diff (version 2.2.1, RRID:SCR_014597, RRID:SCR_001647); then analyzed using CummeRbund (RRID:SCR_014568).

We searched for all genes with evidence of H3K4me3 within 500 bp of a TSS in the human-transfected sample (p<0.05, force-calling, requiring >5 input reads) and with defined FPKM values in the untransfected sample. Of the 14,667 genes passing these filters, 10,652 (73%) have a human PRDM9 binding peak within 500 bp of the TSS. Of these, 873 showed at least some evidence of differential expression between the human-transfected and untransfected samples (p<0.05), and of these, 76 are significant after correction for multiple testing, with 43 significant only in the human-transfected sample (p<0.05 after Benjamini-Hochberg correction).

## Cell culture and transfection for co-IP experiments

For each experiment, 10 million HEK293T cells (ATCC CRL-3216; RRID:CVCL_0063) were seeded in 20 ml growth medium in a 15 cm round cell culture dish. The following morning, cells were transfected by mixing 30 µg total DNA into 800 µl OPTI-MEM (Thermo Fisher Scientific 31985062), then carefully adding 90 µl FuGENE-HD Transfection Reagent and flicking to mix, incubating at room temperature for 15 min, and then adding the mixture dropwise to each dish while swirling gently to mix. After 48 hr, cells were imaged briefly with a fluorescence microscope to confirm expression and were subsequently harvested. As negative controls, additional cells were seeded at the same time but were not transfected.

## Cell lysis and immunoprecipitation for co-IP experiments

Dishes were aspirated to remove media and cells were washed with cold PBS. 2 ml of cold lysis buffer (1% Triton X-100, 150 mM NaCl, 50 mM Tris pH 8.0 plus 2X final concentration of Roche cOmplete Protease Inhibitor Cocktail Tablets) were added and cells were collected into 2 ml Eppendorf tubes using a cell scraper. Tubes were incubated on ice for 30 min and lysates were dounced 20 times in a 2 ml dounce homogenizer with a tight pestle to help shear nuclear membranes. Cells were spun at $2000g$ for 5 min to remove chromatin and cell debris. 100 µl of lysate was set aside as an input control, and the remainder was split evenly among experimental and mock IP conditions. 2 µg of primary antibody (Abcam ChIP-grade rabbit polyclonal anti-HA ab9110 RRID:AB_307019 or anti-V5 ab9116 RRID:AB_307024, or rabbit polyclonal IgG isotype control ab171870 RRID:AB_2687657) was added and lysates were incubated for 1 hr at 4°C with rotation. For each sample, 25 µl of magnetic beads (Invitrogen M-280 Sheep Anti-Rabbit Dynabeads) was equilibrated by washing 3 times in 1 ml cold PBS/BSA (1X PBS, 5 mg/ml BSA, filtered with 0.45-micron filter), then resuspending in 25 µl PBS/BSA. Beads were added to the lysates and incubated for an additional hour at 4°C. Tubes were spun down and placed on a magnetic rack for 1 min. Beads were pipetted up and down in 1 ml cold lysis buffer and rotated for 3 min at 4°C. Washing steps were repeated 4 more times, with all steps taking place in a cold room at 4°C.

## Western blotting

Beads were resuspended in 20 µl 2X Laemmli western loading buffer and boiled for 5 min at 100°C. Beads were removed on a magnetic stand and supernatants were diluted two-fold. The total protein concentrations of input lysates were estimated using a Pierce BCA Protein Assay Kit (Thermo Fisher Scientific 23227) and a NanoDrop spectrophotometer (Thermo Fisher Scientific). 4X Laemmli buffer was added to 50 µg of input protein to a final concentration of 1X then boiled for 5 min at 100°C. Samples were run on 10-well 7.5% Bio-Rad mini-Protean TGX pre-cast gels at 150 Volts in standard TGX running buffer for approximately 1 hr, using 5 µl of Full-Range Rainbow Ladder (VWR 95040–114, Radnor, PA) in one well. Gels were then assembled onto a Bio-Rad mini Trans-Blot transfer pack (with PVDF membrane) according to manufacturer instructions and run on a Trans-Blot Turbo machine on the Mixed MW setting (2.5A, up to 25V, 7 min). Membranes were quickly removed and transferred to 50 ml conical tubes, then blocked for 5 min with rotation in 10 ml Blocking Buffer (5% milk in PBS with 0.1% Tween-20), which was then poured off. Primary antibodies were diluted 1:5,000 in 5 ml blocking buffer and added to the membranes and incubated for 1 hr at room temperature with rotation. Membranes were washed 3 times for 5 min each in PBST (PBS with 0.1% Tween). Secondary antibody (Amersham ECL Donkey anti-Rabbit IgG, HRP-linked, NA934 RRID:AB_772206; GE Healthcare Life Sciences, Pittsburgh, PA) was diluted 1:30,000 in blocking buffer, then 5 ml was added to each membrane and they were incubated for 1 hr at room temperature with rotation. Membranes were washed an additional three times in PBST and one final time in PBS. Blots were imaged using a Bio-Rad Clarity ECL kit according to manufacturer instructions and placed between sheets of transparency film to prevent drying during imaging. Imaging was performed using a Bio-Rad ChemiDoc MP Instrument using chemiluminescence hi-sensitivity settings and signal accumulation mode for various exposure times. Image processing was performed in the Bio-Rad ImageLab software (RRID:SCR_014210), in which relative bands intensities were quantified by densitometry.

## Benzonase treatment of cell extracts followed by co-IP westerns

HEK293T cells (ATCC CRL-3216; RRID:CVCL_0063) were co-transfected for 48 hr with equimolar mixtures of pLenti constructs encoding V5-or HA-tagged full-length (FL) human (h) or chimp (c) PRDM9, or the zinc-Finger (ZF) domain only, using Fugene HD transfection reagent according to the manufacturer's guidelines (Promega). Cells were lysed for 30 min on ice in buffer containing 50 mM Tris-HCl pH 8.0, 150 mM NaCl, 1% Triton X-100 and a cocktail of protease inhibitors (Roche). Cell debris were pelleted by centrifugation at 4°C for 20 min at 20,000$g$. Protein extracts were incubated in the presence or absence of 125 U/ml benzonase (Sigma) and 2 mM MgCl_2 for 1 hr at 4°C with gentle rotation, and clarified again by centrifugation for 15 min at 16,000$g$. Note a pellet is visible after treatment with benzonase. Extracts were incubated for 1 hr at 4°C with 2 µg of anti-V5 antibody (Abcam ab9116 RRID:AB_307024) and a further 1 hr with 25 µl Dynabeads M-280 (Thermo Fisher Scientific). After 5 washes in lysis buffer, the immunocomplexes were eluted from the beads for 5 min at 100°C in 2x Laemmli sample buffer (Bio-Rad) and resolved on a 4–15% (ZF) or 7.5% (FL) Mini-PROTEAN TGX precast gel (Bio-Rad) alongside 50 µg of input extracts (measured by BCA assay, Thermo Fisher Scientific 23227). Proteins were transferred onto PVDF membranes and PRDM9 was detected by western blot following standard procedures. Blots were blocked overnight in PBS containing 0.1% Tween-20% and 5% milk, and incubated for 1 hr at room temperature with anti-HA (Abcam ab9110 RRID:AB_307019) or anti-V5 (Abcam ab9116 RRID:AB_307024) antibodies (1:5,000 dilution), and appropriate ECL HRP-conjugated IgG secondary antibodies (Amersham ECL Donkey anti-Rabbit IgG, HRP-linked, NA934 RRID:AB_772206) with 3 washes in PBS-Tween buffer in between. Protein signals were revealed using the ECL Prime western blotting detection reagent according to the manufacturer's recommendations (GE Healthcare). To assess benzonase digestion efficiency, input protein extracts were diluted 1:20 in 0.1% SDS, and DNA concentration was measured on a nanodrop. 2 µg of DNA from each sample was analyzed on a 2% agarose gel in the presence of 0.1% SDS.

## Immunofluorescence detection of PRDM9 protein variants

HEK293T cells (ATCC CRL-3216; RRID:CVCL_0063) were seeded onto glass coverslips pre-treated with Poly-L-Lysine (Millipore Sigma). Transfections with FL, ZF only and no ZF V5-tagged PRDM9 constructs were carried out for 24 hr, as described above. Cells were fixed for 20 min in chilled methanol, washed three times in PBS, permeabilized for 10 min in PBS containing 0.1% Triton X-100, washed again, and blocked for 1 hr at RT in PBS supplemented with 0.1% Tween 20% and 1% BSA. Cells were immunostained with an anti-V5 antibody (Abcam ab9116 RRID:AB_307024) overnight at 4°C, washed, and incubated for 1 hr at RT with an appropriate secondary antibody conjugated to the Alexa Fluor 594 dye (Thermo Fisher Scientific A21207 RRID:AB_141637). Coverslips were mounted in medium containing DAPI (Vectashield, Vector Laboratories, United Kingdom) and the cells were observed on a Olympus (Japan) BX60 microscope for epifluorescence equipped with a Sensys CCD camera (Photometrics, Tucson, AZ). Images were captured using the Genus Cytovision software (Leica Microsystems, Germany).

### Data availability

Sequencing reads, genome-wide fragment coverage depth, peak calls, and differential gene expression files are available with GEO accession https://www.ncbi.nlm.nih.gov/geo/query/acc.cgi?acc=GSE99407. Source code is available in the Github repository https://github.com/altemose/PRDM9-map (*Altemose, 2017*; copy archived at https://github.com/elifesciences-publications/PRDM9-map).

## Acknowledgements

We would like to thank Jonathan Flint for providing bench space and reagents, as well as Julian Knight, Benjamin Davies, Peter Donnelly, Anjali Gupta Hinch, Robert W Davies, and Catherine M Green for their helpful guidance and feedback. We thank the Oxford Genomics Centre for generating the sequencing data and Garreth McCrudden for proofreading the manuscript.

## Additional information

### Funding

| Funder | Grant reference number | Author |
|---|---|---|
| Wellcome | Investigator Award 098387/Z/12/Z | Simon R Myers |
| Cancer Research UK | Career Development Fellowship C52690/A19270 | J Ross Chapman |
| Howard Hughes Medical Institute | Gilliam Fellowship for Advanced Study | Nicolas Altemose |
| Medical Research Council | Grant MR/L009609/1 | A Radu Aricescu |
| Foreign and Commonwealth Office | Marshall Scholarship | Nicolas Altemose |
| Wellcome | Core Award 090532/Z/09/Z | Nicolas Altemose<br>Nudrat Noor<br>Emmanuelle Bitoun<br>J Ross Chapman<br>A Radu Aricescu<br>Simon R Myers |
| Wellcome | DPhil Studentship 086817/Z/08/Z | Nudrat Noor |

The funders had no role in study design, data collection and interpretation, or the decision to submit the work for publication.

### Author contributions

Nicolas Altemose, Designed all experiments, Performed all ChIP-seq and co-IP experiments, Analyzed and interpreted the data, Wrote and edited the manuscript; Nudrat Noor, Designed and performed YFP-Human ChIP-seq experiments, Contributed to ChIP-seq data analysis and interpretation, Contributed to funding acquisition; Emmanuelle Bitoun, Designed and performed ATAC-seq, RNA-seq, qPCR, and IF experiments, Analyzed qPCR and IF data, Performed co-IP replicates, Helped write and edit the manuscript; Afidalina Tumian, Contributed unpublished data; Michael Imbeault, Provided unpublished KRAB-ZNF data, Edited the manuscript; J Ross Chapman, Provided guidance, reagents, and equipment for cell culture and co-IP experiments; A Radu Aricescu, Helped design ChIP-seq experiments; Provided reagents and equipment for cell culture and ChIP-seq; Simon R Myers, Supervised the study, Acquired funding, Analyzed and interpreted data, Performed THE1B analysis, Wrote and edited the manuscript

### Author ORCIDs

Nicolas Altemose (ID) http://orcid.org/0000-0002-7231-6026
Michael Imbeault (ID) https://orcid.org/0000-0002-0073-0922
J Ross Chapman (ID) http://orcid.org/0000-0002-6477-4254
A Radu Aricescu (ID) https://orcid.org/0000-0003-3783-1388

### Decision letter and Author response

Decision letter https://doi.org/10.7554/eLife.28383.037
Author response https://doi.org/10.7554/eLife.28383.038

## Additional files

### Supplementary files

• Transparent reporting form
DOI: https://doi.org/10.7554/eLife.28383.027

## Major datasets

The following dataset was generated:

| Author(s) | Year | Dataset title | Dataset URL | Database, license, and accessibility information |
| --- | --- | --- | --- | --- |
| Altemose N, Myers SR | 2017 | Mapping PRDM9 binding and its effects in transfected HEK293T cells | https://www.ncbi.nlm.nih.gov/geo/query/acc.cgi?acc=GSE99407 | Publicly available at the NCBI Gene Expression Omnibus (accession no: GSE99407). |

The following previously published datasets were used:

| Author(s) | Year | Dataset title | Dataset URL | Database, license, and accessibility information |
| --- | --- | --- | --- | --- |
| Pratto F, Brick K, Khil P, Smagulova F, Petukhova G, Camerini-Otero R | 2014 | Recombination initiation maps of individual human genomes | https://www.ncbi.nlm.nih.gov/geo/query/acc.cgi?acc=GSE59836 | Publicly available at the NCBI Gene Expression Omnibus (accession no: GSE59836). |
| Imbeault M, Helleboid P, Trono D | 2017 | ChIP-exo of human KRAB-ZNFs transduced in HEK 293T cells and KAP1 in hES H1 cells | https://www.ncbi.nlm.nih.gov/geo/query/acc.cgi?acc=GSE78099 | Publicly available at the NCBI Gene Expression Omnibus (accession no: GSE78099). |

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

# Appendix 1

DOI: https://doi.org/10.7554/eLife.28383.028

## Peak calling algorithm

We developed a maximum-likelihood-based peak calling algorithm that takes as input the number of fragments overlapping a bin (a single base position or an interval) from two ChIP replicates and a genomic background control, as well as three constants describing the coverage ratios between these three inputs, which are estimated genome-wide in an initialization step. The Poisson distribution was chosen as a model of sequencing coverage given its support on all non-negative integers and simple parameterization. As specified, this model assumes that the coverage due to signal is proportional between the two ChIP-seq replicates across the genome and that the coverage due to background is proportional among all three lanes across the genome. We allow for local estimates of background and signal to account for sequence coverage biases and mappability differences across the genome. Ab initio single-base peak calling proceeds in three stages: (1) estimation of constants given coverage values in 100 bp non-overlapping bins genome-wide, (2) single-base maximum likelihood estimation given constants and single-base coverage values, (3) calling of peak centers in the likelihood landscape given a p-value threshold and a threshold on the minimum separation between peak centers.

## Definitions

Let $D_1(i)$, $D_2(i)$ and $G(i)$ be random variables representing the fragment coverage in bin $i$ from the two ChIP-seq replicates and the genomic control, respectively (and let $d_1(i)$, $d_2(i)$ and $g(i)$ represent the observed coverage in bin $i$). We model the coverage of each sequencing replicate $j$ at bin $i$ as a sample from a Poisson distribution with mean $\lambda_j(i)$,

$$D_1(i) \sim Poisson(\lambda_1(i)),$$
$$D_2(i) \sim Poisson(\lambda_2(i)),$$
$$G(i) \sim Poisson(\lambda_g(i)),$$

$$\lambda_1(i) = \alpha_1 b(i) + c(i),$$
$$\lambda_2(i) = \alpha_2 b(i) + \beta c(i),$$
$$\lambda_g(i) = b(i),$$

where $\alpha_1$ and $\alpha_2$ are constants defining how coverage due to background in the ChIP replicates compares to $b(i)$, a parameter representing the mean coverage in the genomic control lane at bin $i$; and $\beta$ is a constant defining how coverage due to binding enrichment in ChIP replicate two compares to $c(i)$, a parameter representing the coverage due to binding enrichment in ChIP replicate one at bin $i$. We wish to test the hypothesis that $c(i) \geq 0$ for each bin $i$.

## Estimating constants

To speed up this step and to provide smoother coverage estimates, we first computed coverage values in 100 bp bins across the autosomes. One can estimate $\alpha_j$ by assuming (conservatively) that when $d_1(i) = 0$ or $d_2(i) = 0$, $c(i) = 0$. That is, one can assume that if ChIP replicate $j$ has coverage 0 at bin $i$, then any coverage in the other replicate ($j'$) arises purely from background. Thus for all $i$ such that $d_j(i) = 0$

$$\lambda_{j'}(i) = \alpha_{j'} b(i),$$
$$\mathbb{E}_{genome}[\lambda_{j'}(i)] = \alpha_{j'} \mathbb{E}_{genome}[b(i)],$$

and thus one can estimate $\alpha_{j'}$ as

$$\hat{\alpha}_{j'} = \frac{\sum\limits_{i:d_j(i)=0} d_{j'}(i)}{\sum\limits_{i:d_j(i)=0} g(i)}. \tag{3}$$

Now an initial estimate of $\beta$ can be computed using genome-wide coverage means $\bar{d}_1, \bar{d}_2, \bar{g}$ as follows:

$$\begin{aligned} \bar{d}_1 &\approx \hat{\alpha}_1\bar{g} + \mathbb{E}_{genome}[c(i)], \\ \bar{d}_2 &\approx \hat{\alpha}_2\bar{g} + \beta\mathbb{E}_{genome}[c(i)], \end{aligned}$$

$$\hat{\beta} \approx \frac{\bar{d}_2 - \hat{\alpha}_2\bar{g}}{\bar{d}_1 - \hat{\alpha}_1\bar{g}}. \tag{4}$$

Next, maximum likelihood estimation and hypothesis testing are performed across all bins (see below), and $\hat{\beta}$ is re-computed as above, using coverage means from the subset of bins with $p<10^{-10}$, for which the ratio of coverage between the two replicates will be less affected by noise.

Finally, using the MLEs $\hat{b}(i)$ and $\hat{c}(i)$ for each bin (see subsection below), a genome-wide estimate of the proportion of reads from signal is computed as for replicate 2.

$$\frac{\sum\limits_i \hat{c}(i)}{\sum\limits_i (\hat{\alpha}_1\hat{b}(i) + \hat{c}(i))} \tag{5}$$

for replicate 1 and as

$$\frac{\sum\limits_i \hat{\beta}\hat{c}(i)}{\sum\limits_i (\hat{\alpha}_2\hat{b}(i) + \hat{\beta}\hat{c}(i))}. \tag{6}$$

## Hypothesis testing

With these estimates of $\alpha_j$ and $\beta$, one can compute Maximum Likelihood Estimators for the unknown parameters $b(i)$ and $c(i)$ at each bin $i$ from the coverage data $d_1(i)$, $d_2(i)$ and $g(i)$ (see below for derivation). Then, using these MLEs one can compute a log-likelihood ratio test statistic against a null model in which $c(i) = 0$:

$$\Lambda(i) = 2\log\frac{\max\limits_{b(i),c(i)\geq 0}[L(D_1(i)=d_1(i),D_2(i)=d_2(i),G(i)=g(i))]}{\max\limits_{b(i),c(i)=0}[L(D_1(i)=d_1(i),D_2(i)=d_2(i),G(i)=g(i))]}. \tag{7}$$

Under the null hypothesis, the test statistic $\Lambda(i)$ is distributed approximately as a $\chi^2$ distribution (with 1 degree of freedom due to the parameter $c(i)$ and an atom of probability at 0), yielding a p-value at each bin $i$ indicating the probability that the observed likelihood ratio could arise from background alone.

## Calculation of Maximum Likelihood Estimators

Recall that at each position the Poisson means for coverage in each lane are (dropping the $i$ notation for succinctness)

$$\begin{aligned} \lambda_1 &= \hat{\alpha}_1 b + c, \\ \lambda_2 &= \hat{\alpha}_2 b + \hat{\beta}c, \\ \lambda_g &= b, \end{aligned}$$

where $\hat{\alpha_1}$, $\hat{\alpha_2}$, and $\beta$ are constants estimated for the whole genome. To simplify calculations, we reparameterize using a new variable $y = c/b$ and rewrite the above equations as

$$\lambda_1 = \hat{\alpha_1}b + yb,$$
$$\lambda_2 = \hat{\alpha_2}b + \hat{\beta}yb,$$
$$\lambda_g = b.$$

Given the observed coverage values $d_1$, $d_2$, and $g$, the Poisson log likelihood function can be written as

$$
\begin{aligned}
\ell \;\propto\; & -\lambda_1 + d_1\log(\lambda_1) - \lambda_2 + d_2\log(\lambda_2) - \lambda_g + g\log(\lambda_g) \\
= \; & -\hat{\alpha_1}b - yb + d_1\log(\hat{\alpha_1}b + yb) - \hat{\alpha_2}b - \hat{\beta}yb + d_2\log(\hat{\alpha_2}b + \hat{\beta}yb) - b + g\log(b) \\
= \; & -b(\hat{\alpha_1} + \hat{\alpha_2} + 1) - yb(1 + \hat{\beta}) + d_1\log(\hat{\alpha_1}b + yb) + d_2\log(\hat{\alpha_2}b + \hat{\beta}yb) + g\log(b).
\end{aligned}
\tag{8}
$$

Now to maximize $\ell$ we first obtain the partial derivatives for $b$ and $y$

$$
\begin{aligned}
\frac{\partial \ell}{\partial b} \; = \; & -(\hat{\alpha_1} + \hat{\alpha_2} + 1) - y(1 + \hat{\beta}) + \frac{d_1(\hat{\alpha_1}+y)}{b(\hat{\alpha_1}+y)} + \frac{d_2(\hat{\alpha_2}+\hat{\beta}y)}{b(\hat{\alpha_2}+\hat{\beta}y)} + \frac{g}{b} \\
= \; & -(\hat{\alpha_1} + \hat{\alpha_2} + 1) - y(1 + \hat{\beta}) + \frac{1}{b}(d_1 + d_2 + g),
\end{aligned}
\tag{9}
$$

$$
\begin{aligned}
\frac{\partial \ell}{\partial y} \; = \; & -b(1 + \hat{\beta}) + \frac{d_1 b}{b(\hat{\alpha_1}+y)} + \frac{d_2\hat{\beta}b}{b(\hat{\alpha_2}+\hat{\beta}y)} \\
= \; & -b(1 + \hat{\beta}) + \frac{d_1}{(\hat{\alpha_1}+y)} + \frac{d_2\hat{\beta}}{(\hat{\alpha_2}+\hat{\beta}y)}.
\end{aligned}
\tag{10}
$$

Next, we set the partials to 0 and solve them as a system to obtain any potential local maxima. We start by solving for $b$ in **Equation 9** as follows:

$$
\begin{aligned}
0 \; = \; & -(\hat{\alpha_1} + \hat{\alpha_2} + 1) - y(1 + \hat{\beta}) + \frac{1}{b}(d_1 + d_2 + g); \\
b \; = \; & \frac{d_1 + d_2 + g}{\hat{\alpha_1} + \hat{\alpha_2} + 1 + y(1 + \hat{\beta})}.
\end{aligned}
\tag{11}
$$

Then, we substitute it into **Equation 10** and rewrite it as follows, with the aim of simplifying it into quadratic form:

$$
\begin{aligned}
0 = & -\frac{d_1 + d_2 + g}{\hat{\alpha_1} + \hat{\alpha_2} + 1 + y(1 + \hat{\beta})}(1 + \hat{\beta}) + \frac{d_1}{(\hat{\alpha_1}+y)} + \frac{d_2\hat{\beta}}{(\hat{\alpha_2}+\hat{\beta}y)}; \\
\frac{(d_1 + d_2 + g)(1 + \hat{\beta})}{\hat{\alpha_1} + \hat{\alpha_2} + 1 + y(1 + \hat{\beta})} & = \frac{d_1(\hat{\alpha_2}+\hat{\beta}y) + d_2\hat{\beta}(\hat{\alpha_1}+y)}{(\hat{\alpha_1}+y)(\hat{\alpha_2}+\hat{\beta}y)} \\
& = \frac{d_1\hat{\alpha_2} + d_1\hat{\beta}y + d_2\hat{\beta}\hat{\alpha_1} + d_2\hat{\beta}y}{\hat{\alpha_1}\hat{\alpha_2} + \hat{\alpha_1}\hat{\beta}y + \hat{\alpha_2}y + \hat{\beta}y^2} \\
& = \frac{y(d_1\hat{\beta} + d_2\hat{\beta}) + d_1\hat{\alpha_2} + d_2\hat{\beta}\hat{\alpha_1}}{\hat{\alpha_1}\hat{\alpha_2} + y(\hat{\alpha_1}\hat{\beta} + \hat{\alpha_2}) + \hat{\beta}y^2}.
\end{aligned}
\tag{12}
$$

To shorten notation, we substitute in the following variables for constant terms in **Equation 12**:

$$
\begin{aligned}
t_1 &= (g + d_1 + d_2)(1 + \hat{\beta}), \\
t_2 &= \hat{\alpha_1} + \hat{\alpha_2} + 1, \\
t_3 &= 1 + \hat{\beta}, \\
t_4 &= d_1\hat{\alpha_2} + d_2\hat{\beta}\hat{\alpha_1}, \\
t_5 &= d_1\hat{\beta} + d_2\hat{\beta}, \\
t_6 &= \hat{\alpha_1}\hat{\alpha_2}, \\
t_7 &= \hat{\alpha_1}\hat{\beta} + \hat{\alpha_2},
\end{aligned}
$$

yielding

$$\frac{t_1}{t_2+yt_3} = \frac{yt_5+t_4}{t_6+yt_7+\hat{\beta}y^2};$$

$$0 = t_1(t_6+yt_7+\hat{\beta}y^2) - (t_2+yt_3)(yt_5+t_4);$$

$$0 = t_1t_6 + yt_1t_7 + t_1\hat{\beta}y^2 - yt_2t_5 - t_2t_4 - y^2t_3t_5 - yt_3t_4; \tag{13}$$

$$0 = y^2(t_1\hat{\beta}-t_3t_5) + y(t_1t_7-t_2t_5-t_3t_4) + (t_1t_6-t_2t_4).$$

Now we can solve for $y$ in **Equation 13** using the quadratic formula, taking the positive root to be $\hat{y}$, the MLE for $y$, which we report as the 'enrichment' value for that bin. To obtain $\hat{b}$, we simply substitute $\hat{y}$ into **Equation 11** and, to return to the original paramaterization, $\hat{c}$ is simply computed as $\hat{y}\hat{b}$. Finally, to obtain $\hat{b}_0$, the MLE for $b$ under the background model, we can simply set $y$ to 0 in **Equation 11**, yielding

$$\hat{b}_0 = \frac{d_1+d_2+g}{\hat{\alpha}_1+\hat{\alpha}_2+1}. \tag{14}$$

## Peak calling and centering

Given a likelihood ratio value $\Lambda(i)$ for each base $i$ along a chromosome, along with a p-value threshold (which is converted to a lower bound on the likelihood ratio, $l$) and $m$, a threshold on the minimum separation between peak centers, initial peak centers are found by identifying all significant bases (bases for which $\Lambda(i)>l$) that are local maxima. Specifically, each significant base is scanned to test if

$$[\Lambda(i) > \max_{i-m-1}\Lambda(j)] \text{and} [\Lambda(i) \geq \max_{i+1+m}\Lambda(j)].$$

At each initial peak center satisfying these criteria, a confidence interval is computed by identifying the nearest position $j$ to the left and to the right (by a maximum of 1000 bp) where $(\Lambda(i) - \Lambda(j))>9.12$, which defines a 99% confidence interval for the peak center (using $\chi_2^2$, with one degree of freedom for the enrichment factor and one for the peak center position). All confidence intervals along a chromosome are then sorted from narrowest to widest, and in this order each confidence interval is added one at a time to the final peak set, provided it does not overlap any of the confidence intervals already included in the final peak set. This produces a final peak set with non-overlapping confidence intervals, favoring inclusion of stronger peaks with narrower confidence intervals. Finally, to refine peak centers in confidence intervals with multiple tied bases, the rounded mean position of all maximal bases is reported as the peak center. The resulting final peak set reports $\hat{y}$ and the p-value for $\Lambda$ at the peak center as the enrichment and p-value for that peak.

## Force-calling

This algorithm enables maximum likelihood estimation and hypothesis testing at any arbitrary bin in the genome, when provided with coverage values and estimates of $\alpha_1$, $\alpha_2$, and $\beta$. This enables us to 'force-call' enrichment and p-values at pre-specified locations in the genome, for example to determine what fraction of gene promoters show evidence of H3K4me3 enrichment in a 1 kb window centered on the transcription start site.

## Appendix 2

DOI: https://doi.org/10.7554/eLife.28383.029

## Details of THE1B analysis

We developed an approach to identify motifs associating with various cellular phenotypes generated by or studied in this paper, specifically in and around THE1B elements. THE1B repeats are homologous repeat elements found across the genome, are non-genic in general, and are centers of hotspot activity. We sought to characterize how (and if) naturally arising DNA sequence differences across the 20,696 autosomal THE1B copies impact both recombination and other measurable epigenetic features. Robustly identified associations are likely to be causal, because the underlying DNA sequences are not in general believed to be specifically and consistently altered by the presence/absence of epigenetic features but, instead, can influence these features. We used association testing to identify candidate sequence motifs, then leveraged conditional testing to successively identify independent signals. This accounts for the fact that overlapping motifs, and even non-overlapping motifs, are correlated in which THE1B elements possess them. We performed testing based on the exact occurrence of 7 bp motifs. This length was chosen as a balance between specificity within the THE1B sequence, and occurring relatively commonly across THE1B elements. First, for the 20696 autosomal THE1B LTR elements annotated by RepeatMasker software (hg19/Build 37, downloaded from the UCSC genome browser, and mapped to the positive strand relative to the THE1B consensus sequence) we produced a 20,696×16,384 matrix recording presence/absence of each motif of length seven in each THE1B copy, across the genome. All subsequent analyses were then restricted to the 2021 such motifs present in at least 500 different THE1B elements (i.e. at least 2.5% of THE1B copies, aiding statistical power to detect potential associations). For each matrix row, we can view the set of motifs present as characterizing a single THE1B repeat copy in terms of common 'variation' across such THE1B repeat copies. We annotated each THE1B repeat copy with various 'phenotypes' for example whether a recombination hotspot was present at that repeat copy. Then, we tested for association between each motif or groups of motifs, viewed as predictors, and the phenotype. This quantifies the impact of the set of common single or multiple base changes, against the 364 bp THE1B consensus sequence, on different recombination-related phenotypes. Motifs of interest were given a position relative to the 13 bp motif 'CCTCCCTAGCCACG' previously identified (*Myers et al., 2008*) as predicting hotspot status in THE1B repeats, and closely matching the C-terminal end of the PRDM9 binding consensus sequence. This motif maps to positions 261–274 in the THE1B consensus. To positionally map each motif, we used the mode of that motifs first base position, relative to the first base of the motif CCTCCC[CT]AGCCA[CT]G, within THE1B repeat copies containing these two motifs. Phenotypes/annotations were either 0–1 (e.g. hotspot status, binding peak overlap), or quantitative (in the form of counts, for the H3K4me3 signal strength, specifically the number of reads observed). For the conditional testing we therefore used generalized linear models (GLMs) with either a binomial, or quasi-Poisson, underlying model as appropriate, as implemented in the 'glm library in R. For association testing we used Fisher's exact test for association between 0–1 phenotypes and 0–1 motif occurrences, testing each motif separately. We performed different analyses catering for different phenotypes as appropriate, which we describe in subsequent sections.

## Identifying motifs associated with PRDM9 binding to THE1B elements

We used our human PRDM9 ChIP-Seq data to annotate each THE1B element as bound or not bound by PRDM9. Specifically, an element was defined as bound if it overlapped an identified PRDM9 binding peak region ($p < 10^{-5}$). A substantial fraction of human THE1B elements (4392 of 20696, 21%) were found to be bound. These PRDM9-bound copies fully explain THE1B enrichment among recombination hotspots identified by DMC1 mapping (1155 hotspots; $p < 10^{-15}$ by FET; odds ratio 10.8; *Pratto et al., 2014*), or LD mapping (1209 hotspots;

*Frazer et al., 2007*). Unbound THE1B repeats do not show significantly greater overlap with DMC1 hotspots than expected by chance (p=0.18 compared to a null set of THE1B repeat positions right-shifted 5 kb). Nevertheless, many strongly bound THE1B repeat copies still do not become hotspots.

Recording binding across elements as a 0–1 vector, we successively fit GLMs of increasing complexity in a stepwise fashion, testing association between sets of motifs as regressors, and PRDM9 binding/non-binding as a response. In each model, we added a second matrix of regressors with entries defining which of the previously identified motifs CCTCCCCAGCCATG (matching the THE1B consensus sequence), CCTCCCTAGCCACG, CCTCCCTAGCCATG, or CCTCCCCAGCCACG, were present. These motifs are known to influence PRDM9 binding in THE1B elements (*Myers et al., 2008*). Including these additional regressors avoids false positive associations due to motifs whose presence/absence associates with these previously known determinants of PRDM9 binding. We restricted testing to only THE1B elements containing an exact match to one of these motifs, to avoid complexities due to cases of unusual PRDM9 binding to diverged THE1B sequences. Specifically, beginning with the model having only the four motifs above as predictors, we successively added in that new motif (of all 2021 possible motifs) maximally increasing the likelihood (as measured by the model deviance in the fitted GLM) of observed peak/non-peak status. We restricted the set of possible next motifs to those not strongly correlated ($r^2<0.95$) with the current set of included predictors, to avoid statistical artifacts due to near-complete motif co-occurrence and correlations, and to ensure a set of sufficiently independent predictors. Motifs were added in successively, until the conditional p-value of the next candidate motif was not significant (p<0.05) after Bonferroni correction for 2021 motifs tested. This yielded a final set of 17 motifs. We used the final joint GLM fit to estimate the joint effect of each motif on the probability of seeing a PRDM9 binding peak – in the binomial model, this is interpretable as the increase in the log-odds of a hotspot given each motif occurs, and taking into account the other motifs effects. We note that

1. Each of the 17 identified motifs by construction shows very strong evidence of influencing binding status, significant after Bonferroni correction for multiple testing (p<0.05).

2. All identified motifs map in – or close to – the predicted binding target region of PRDM9 based on our new set of motifs (*Figure 4a*). Different motifs act either to increase or decrease binding probability.

The estimated positions, effects and standard errors of each motif are shown in *Figure 4a* (top row). The full list of motifs themselves and estimated effect sizes is provided in *Figure 4—source data 1*.

## Identifying motifs impacting hotspot status conditional on PRDM9 binding presence/absence

We annotated each THE1B element according to whether it overlapped a hotspot in a set of previously published human recombination hotspot positions (*Pratto et al., 2014*). That study examined meiotic DMC1 signal in male carriers of three different PRDM9 alleles (A, B, and C). Alleles A and B bind similar target sites, and the B allele is studied here. We accordingly measured overlap only for hotspots detected in individuals whose PRDM9 alleles were both either A or B. We also annotated each THE1B element according to whether it overlapped an LD-based human hotspot (*Frazer et al., 2007*). These two annotations were highly correlated (p<$10^{-15}$ by FET; odds ratio 25.6). Moreover, 1,676 THE1B repeats overlapped Pratto et al. hotspots (2,266 for LD-based hotspots), confirming that thousands of human hotspots localize in or near to THE1B elements. Having annotated THE1B repeats according to hotspot status, we used the same procedure as described above to test sequence motifs for association with hotspot status, separately for both hotspot sets. This analysis tests for evidence of association of different motifs with hotspot status, by influencing binding or other factors. We again used the same procedure, restricting to the set of THE1B elements defined as bound by PRDM9 above, to identify independent motifs associating with hotspot activity *conditional* on PRDM9 binding. We intersected motifs identified by these four analyses to identify a set of motifs robustly associating with hotspot occurrence, even given that measurable binding by PRDM9

occurs. (An initial comparison did not identify any evidence of motifs influencing one hotspot set differentially to the other, as might occur if e.g. female-specific influences on recombination rate exist within THE1B elements, and so we concentrate on this combined analysis). First, we identified seven motifs with independent, significant evidence (p<0.05 after Bonferroni correction) of association with whether an LD-based hotspot was observed, conditional on binding by PRDM9 in our ChIP-seq experiment. Separately, we identified four overlapping motifs with significant evidence of impacting the chance of being a Pratto et al. hotspot, conditional on binding by PRDM9 in our ChIP-seq experiment. Using the set of 9 unique motifs, we then fit a series of GLMs to jointly test for association of a 0–1 matrix with nine columns indicating motif presence/absence on (i) LD-based hotspot status, (ii) Pratto-based hotspot status in human males, and (iii)-(iv) the same conditional on PRDM9 binding, i.e. restricting testing to the set of THE1B elements overlapping a PRDM9 binding peak. In each model, we continued to include as regressors the previously identified 14 bp motifs influencing PRDM9 binding, and restrict testing to elements containing one of these motifs. Following this joint analysis, seven motifs show (a) p<0.05 (Bonferroni corrected p-value) for hotspot occurrence given binding, for at least one of the Pratto hotspot set and the LD-based hotspot set and (b) p<0.05 (nominal p-value) for all four tests, i.e. evidence of influencing hotspot status regardless of hotspot definition used, and both conditional and unconditional on PRDM9 binding. All but one of these motifs associate (p<0.05 after Bonferroni correction) with hotspot occurrence *unconditionally* also. We considered these seven motifs to form a set of independent, robust and consistently detected influences on hotspot status within THE1B repeats. For example, the motif 'ATCCATG' shows p<0.05 after Bonferroni correction for all of (i-iv) above. Specifically, testing this motif (conditional on previously identified 14 bp PRDM9 binding motifs) at all THE1B repeats, without conditioning on PRDM9 binding, showed $p=4.1\times10^{-11}$ for association with DMC1 hotspots and $p=5.9\times10^{-13}$ for association with LD-based hotspots and odds ratios of around 0.5. This means that its impact on hotspots cannot be mediated via any biases in our ability to measure binding in HEK293T cells. The other two motifs of nine may associate with hotspot status, but were conservatively excluded because they showed no evidence (p>0.05) for unconditional evidence of association with hotspot status. They were removed in case their effect is mediated through properties of PRDM9 binding, specific to HEK293T cells. The detailed results of this conditional testing are given in *Figure 4—source data 1*, and were used to produce the first two rows of *Figure 4a*.

## Identifying motifs associated with previously measured H3K4me3 signal strength in testes

A previous human study measured levels of H3K4me3 in testes (*Pratto et al., 2014*). Although PRDM9 deposits H3K4me3 on binding, other proteins are capable of inducing this mark, and H3K4me3 occurs, for example, at many human promoters independently of PRDM9. We sought to identify sequence features impacting male meiotic H3K4me3 in THE1B elements, whether bound by PRDM9 or not bound. We 'force-called' testis H3K4me3 as a quantitative phenotype at each THE1B element, and here test for association with the total number of reads observed across two replicates within 1 kb of the center of the element. We split the THE1B elements into two sets, those with potential PRDM9 binding (the 'bound set') and a set robustly evidenced to not be bound by PRDM9 (the 'unbound set'). For the bound set, we took the subset of THE1B elements containing an exact match to one of the 14 bp motifs CCTCCC[CT]AGCCA[CT]G, and overlapping a PRDM9 ChIP-seq peak. For the unbound set, we conservatively used the set of THE1B repeats remaining after removing as potentially bound by PRDM9 any repeat matching CCTCCC[CT]AGCCA[CT]G, or overlapping a PRDM9 binding site in our HEK293T cells, or overlapping an LD-based hotspot, or overlapping any Pratto et al. hotspot. The remaining THE1B elements contain no good match to the PRDM9 binding motif, and further show no evidence of any PRDM9-associated phenotype (binding or hotspot status). We then performed testing exactly as for the 0–1 annotations, to identify independent motifs associating with testis H3K4me3 level in each set. The only difference in each case was the GLM used (quasi-Poisson model). Notably, in the non-bound set of THE1B repeats, we are then testing for sequence features associating with H3K4me3 levels,

independent of PRDM9. Similarly to PRDM9 binding motifs, the identified motifs are likely to causally influence histone modifications including H3K4me3 levels (and as described in the main text and below, they also associate with H3K9me3 and H3K4me3 in somatic cells, and potentially other modifications), but through initially unknown biological mechanisms. In the bound set, both PRDM9-dependent and PRDM9-independent sequence features might be identified. The testing of non-bound regions identified 18 distinct motifs after Bonferroni correction of significance level, mapping throughout the THE1B consensus sequence and associated with both increases and decreases in measured H3K4me3 signal. The estimated positions, effects and standard errors of each motif were used to construct and *Figure 4d*. The full list of motifs themselves and estimated effect sizes is provided in *Figure 4—source data 1*. We note that all the motifs, except possibly one, map *outside* the PRDM9 target motif region, consistent with a role distinct from PRDM9. Further supporting this idea, 15/18 motifs show effects in the same direction for the bound set testing of the smaller, and so statistically less well powered, collection of PRDM9-bound repeats, suggestive of a continuing impact even if elements are also bound by PRDM9; although this reached significance in only four cases (p<0.05, with p<0.0001 for the motif with the strongest signal), this can be explained by the dominant impact of PRDM9 binding on H3K4me3 for this set, as well as the smaller sample size.

## Overlaps and correlations between recombination-related measures

The above procedures produced three partially overlapping sets of motifs that are highly significantly associated with PRDM9 binding, hotspot occurrence (measured by LD or DMC1) at sites bound by PRDM9, and testis H3K4me3 marks formed dependently and independently of PRDM9, respectively. We compared the sets of motifs identified – independently, using different phenotypic measures and often different sets of THE1B repeats – for overlaps. Given each set of motifs, we used the same procedures as described above to test the other measures, in order to examine whether the same features might have directional effects for the other measures and phenotypes. The results are shown in *Figure 4—source data 1* and described briefly in the main text. Overall, we found the following:

1. The determinants of PRDM9 *binding* we identified are found exclusively within the region directly contacted by the zinc fingers of PRDM9, or immediately adjacent (<10 bp). All influences on binding mapped within a region from −22 bp to + 14 bp relative to the motif CCTCCCTAGCCAC, in every case overlapping by the predicted PRDM9 binding motif within THE1B. While a previous report suggested influences on PRDM9 binding outside the binding region (*Grey et al., 2017*), these are not strongly evidenced here, although the motif from + 14 bp to 22 bp inclusive extends slightly beyond the region bound by PRDM9. Finally, the motif CCTCCTT (p=9.94×10$^{-5}$) is the most significant motif failing to reach Bonferroni significance, mapping just upstream of the region directly predicted to be within the binding region (−29 bp to −23 bp inclusive), suggesting there may be a weak role for sequence <10 bp away but not overlapping the identified motif itself.

2. Changes in DNA sequence throughout the roughly 40 bp PRDM9 binding target region (17 motifs) impact meiotic recombination, and recombination heat as well as H3K4me3 deposition seem to depend in a simple directional manner on binding. In general almost all (two exceptions discussed below) of 17 motifs impacting binding impact H3K4me3 at the bound sites in the same direction in human testes, i.e. during meiosis (where PRDM9 is expressed). Moreover, with the same 2 exceptions, all had a trend for measured recombination in the same direction when measured by LD and/or DMC1. For multiple motifs these associations were highly significant (*Figure 4—source data 1*). This finding is not unexpected but confirms the biological relevance of precisely and directly measuring binding, even in HEK293T cells.

3. As well as the above, and surprisingly, we identified a large number of motifs (18 reaching Bonferroni-corrected significance), associating with H3K4me3 signal strength in human testes at regions not bound by PRDM9. They map throughout the THE1B repeats, with only one

overlapping the PRDM9-bound region. These motifs each have rather weak signals for the H3K4me3 signal compared to (for example) PRDM9 binding. However as we discuss below, the same motifs each show (stronger) impacts on H3K9me3 deposition within a large collection of cell types, and so it may be that histone modifications other than H3K4me3 drive the links between these motifs and meiotic recombination (see below), and our H3K4me3 signals appear as secondary biological markers of this stronger effect. We therefore call these 'non-PRDM9' H3K9me9/H3K4me3 motifs.

4. We observed a strong, consistent, counter-directional correlation with non-PRDM9 H3K9me9/H3K4me3 motifs and hotspot activity. In THE1B elements, the sequence features increasing H3K9me9/H3K4me3 measured signals decrease recombination rate, in a seemingly simple linear fashion, and (less strongly) the opposite holds for decreases in H3K9me9/H3K4me3.

   First, of the seven new motifs identified to influence whether hotspots occur given binding in THE1B, three occur within the PRDM9 target motif, and are explained via direct changes on binding strength, in the expected direction. The remaining four motifs are outside the PRDM9 target motif. All of these are strongly associated ($p<10^{-60}$ for ATCCATG) with non-PRDM9 H3K9me9/H3K4me3, in the opposite direction to the recombination association (*Figure 4—source data 1*).

   Conversely, testing influence of the 18 non-PRDM9 H3K9me9/H3K4me3 motifs on (i) PRDM9 binding, and (ii) LD/DMC1 hotspot formation, we found no particular association with PRDM9 binding itself, and no overlap with the set of motifs identified to influence PRDM9 binding. However, for 17/18 motifs associated with increased/decreased H3K9me9/H3K4me3 levels, they were associated with decreased/increased probability of hotspot occurrence for each of LD-based hotspots and DMC1-based hotspots. The only exceptions in terms of direction showed non-significant trends, in different directions for the two sets of hotspots, so might be explained by statistical noise. Multiple motifs show significant evidence of altering hotspot probability (*Figure 4a*; *Figure 4—source data 1*).

   In particular, the most significant motif, associated with increased non-PRDM9 H3K9me9/H3K4me3, was again 'ATCCATG' ($p=4\times10^{-26}$). This motif has no association with PRDM9 binding in our experiments ($p>0.1$) but overwhelming evidence of reducing hotspot probability at these binding sites and is in the motif set identified independently as associating with hotspot occurrence ($p<10^{-4}$ for association with hotspot occurrence given binding, for each of DMC1 and LD hotspots).

5. As mentioned above, two motifs, 'TTGTGAG' and 'CCATGAT', have significant impacts on both PRDM9 binding and meiotic recombination, but in *opposite* directions. This unusual property might in principle reflect subtle differences in binding properties between PRDM9 alleles A/B or in different cellular environments (HEK293T cells vs. cells where PRDM9 is natively expressed). However a simpler explanation given the above is offered by the fact that both motifs have a weak positive association with non-PRDM9 H3K9me9/H3K4me3 independent of PRDM9 binding ($p<0.005$ in each case). Thus there may be competition for these motifs involving an increase in PRDM9 binding, but within an environment where other histone modifications they cause make a hotspot less likely, plausibly resulting in a predicted decrease in hotspot probability *given* binding, as observed. Thus the complex patterns we observe comparing thousands of sequence motifs across thousands of THE1B elements for four different recombination-related phenotypes may actually be highly parsimoniously explained by a simple but surprising phenomenon: PRDM9 binding and PRDM9-induced H3K4me3 deposition dramatically increase hotspot probability, but PRDM9-independent H3K4me3 and/or H3K9me3 (see below) dramatically inhibit recombination, downstream, even where PRDM9 is able to bind the THE1B repeat.

## Examining the impact on recombination of non-PRDM9 H3K9me3/H3K4me3 motifs in THE1B

To explore this signal, we plotted the estimated effect on H3K4me3 signal strength (log-fold increase on measured H3K4me3 signal) of each motif versus the average impact on recombination (measured by log-odds of a hotspot), in *Figure 4—figure supplement 1c*. This revealed a striking, essentially linear, negative trend ($p<10^{-16}$ by rank correlation; rank

correlation −0.85). Given these consistent marginal effects, we next examined how much influence these motifs have jointly, on whether hotspots occur or otherwise at THE1B repeats bound by PRDM9. Conceptually we imagine PRDM9-induced H3K4me3 increasing recombination, but other motifs that increase the non-PRDM9 H3K9me9/H3K4me3 signal, instead reducing recombination – in 'opposition'. Although we can use the H3K4me3 data in the appropriate tissue (testes), the signals obviously and unfortunately conflate, and cannot separate whether these data measure H3K4me3 deposited by PRDM9. However, we *can* separate them by using our identified motifs. We used (only) the DNA sequence of each THE1B repeat to predict the non-PRDM9 H3K9me3/H3K4me3 for that repeat. This is expected to negatively correlate with recombination from the above findings. It appears as if PRDM9 binding in general does not alter the effect of non-PRDM9 H3K9me3/H3K4me3 motifs (*Figure 4—source data 1*), so this DNA-sequence-based measure is likely to remain relevant in those repeats also bound by PRDM9. Indeed: in the column 'H3K4me3' at bound THE1B elements of *Figure 4— source data 1*, almost all the identified non-PRDM9 H3K9me9/H3K4me3 motifs have impacts in the same direction (rank correlation p=0.00036) for the unbound repeats, including e.g. the motif ATCCATG (p=$2\times10^{-5}$). In detail, for each element we calculated a separate 'positive' and 'negative' motif score (relative to a conceptual highly diverged THE1B element containing none of the motifs) for only motifs acting in those directions, summing the values given in column 'N' of *Figure 4—source data 1* across motifs present in that repeat copy. We fit a regression model (Poisson GLM as above) and found both scores to be highly significantly associated with hotspot occurrence (p=$9.9\times10^{-6}$ and p=$1.7\times10^{-7}$ respectively) in opposite directions, though with slightly different coefficients. We combined the scores by adding them, downweighting/tempering the negative part of the non-PRDM9 H3K9me9/H3K4me3 signal by 2.3637/3.4842, the ratio of regression coefficients. This yields a single prediction value of the non-PRDM9 component of H3K4me3 per THE1B repeat. To visualize the impact of non-PRDM9 H3K9me9/H3K4me3 signal on hotspots (*Figure 4d*), restricting our analysis to the set of elements defined as bound by PRDM9 as above, we binned their predicted non-PRDM9 component of the H3K9me3/H3K4me3 signal into 10 equal quantiles. For each quantile, we plotted the (log-fold) mean H3K9me3/H3K4me3 change, against the probability of a hotspot given binding. It should be noted that these correspond to a rather modest range of predicted H3K4me3 changes - for example the 95% upper quantile of the summed positive influences on H3K4me3 corresponds to just a 1.3-fold increase in signal over background. It is difficult to quantify how strong this is biologically given noise in the H3K4me3 assay, but a helpful comparison might be that the single motif CCTCCCTAGCCAC confers a >2-fold increase in H3K4me3 signal in testes within bound PRDM9 repeats even conditional on binding occurring, so it seems likely that H3K4me3 differences made by these motifs are modest – and require caution in interpretation, given the same motifs also associate with much stronger H3K9me3 level differences (see below). Strikingly and nevertheless, as a group these motifs produce a very large and consistent impact on hotspot probability, almost identical for the DMC1 and LD-based hotspot sets. Hotspot probability reduced almost 3-fold, from 35% to 13%, as non-PRDM9 H3K9me3/H3K4me3 increased. Thus, complex non-PRDM9 sequence factors operate in combination to collectively determine whether hotspots occur at THE1B repeats.

## General suppression of meiotic recombination but not PRDM9-associated H3K4me3 deposition, by the motif ATCCATG

We investigated whether non-PRDM9 H3K9me9/H3K4me3 sequence motifs reduce recombination by preventing PRDM9 from binding DNA and therefore recruiting DSBs, or instead act downstream of PRDM9 binding. For the most significant motif 'ATCCATG' we were able to test this by plotting mean LD-based and DMC1-based recombination rate, and H3K4me3 level in human testes, for a 10 kb region (500 bp window slide 250 bp across region) centered on the THE1B repeat. We calculated and plotted each mean separately, grouping THE1B repeats according to whether they contain different PRDM9-bound motifs of the form

CCTCCC[CT]AGCCA[CT] resulting in progressively stronger binding by PRDM9, and then either contain, or do not contain, the motif 'ATCCATG' (*Figure 4b*). As expected, the recombination signal increases steadily with closeness of the match to the PRDM9 consensus sequence CCTCCCTNNCCAC. Conditional on this closeness, presence of the motif ATCCATG always and strongly reduces mean recombination rate by around 2-fold. Even where no PRDM9 binding motif is present inside the THE1B repeat itself (*Figure 4b*, cyan lines) there is a statistically significant ($p < 10^{-10}$) suppression of mean recombination rate *below background* when the motif ATCCATG occurs, at a scale of approximately 1–2 kb in each direction. Thus, the motif ATCCATG within THE1B repeats appears to be a strong general local suppressor of human recombination, and is able to suppress recombination when PRDM9 binds the usual motif in THE1B, and nearby hotspot occurrence more widely. Moreover, this suppression acts over reasonably broad scales. In contrast to their different effects in recombination, while the H3K4me3 signal consistently increases with closeness of the match to the PRDM9 consensus sequence CCTCCCTNNCCAC, it is also higher when the non-PRDM9 motif ATCCATG is present, with no evidence that this motif suppresses PRDM9-dependent H3K4me3 deposition in vivo. It appears that PRDM9 binding, and ATCCATG-driven histone modifications, act additively and perhaps independently. Therefore, this single non-PRDM9 motif must play a strong suppression role in a high proportion of the THE1B repeats where it is present. Likely, this suppression acts in both males and females, because DMC1 rate estimates are for males only, while LD-based rate estimates are sex-averaged and reflect mainly ancient crossovers.

## Association testing the full landscape of histone modifications in THE1B repeats across ROADMAP cell lines

The ROADMAP consortium (*Kundaje et al., 2015*) previously measured multiple histone modifications and other molecular phenotypes across 125 diverse human somatic cell types. These were used to partition the genome into 15 different domains characterized by combinations of histone modifications: TssA , TssAFlnk, TxFlnk, **Tx**, **TxWk**, EnhG, **Enh**, **ZNF/Rpts**, **Het**, TssBiv, BivFlnk, EnhBiv, **ReprPC**, **ReprPCWk**, **Quies**. Eight of these states (in bold) occur over eight times across the 20696 THE1B repeats on average and were examined. We first identified the ROADMAP domain inference for each THE1B repeat in each of the studied cell types. For each domain type and each cell type, we identified de novo a set of motifs associating with that domain in that cell type, by exactly repeating the analysis approach we used for hotspot status, as described above. We used a p-value cutoff of $2.5 \times 10^{-8}$, to Bonferroni correct for the total of $125 \times 8 \times 2021$ tests performed. The full resulting set of 1571 identified ROADMAP motifs and details is given in *Figure 4—source data 1*. The motifs cover all eight domain types, and every cell type has at least three, and up to 36, different motifs. Thus, as in meiosis THE1B repeats possess a diverse set of independent motifs associated with many different histone modifications (including H3K9me3, H3K27me3, H3K4me3, H3K36me3, among others) in THE1B elements. Although our main focus here is on correlating results with recombination rates, the collection of motifs is of biological interest in itself. We grouped highly co-occurring (and typically overlapping) motifs, collapsing motifs whose correlation (in which THE1B element each motif occurred in) was >50% until no further grouping was possible. This resulted in a set of 67 distinct summary motif groups, whose results are summarized in *Figure 4—source data 1*, and which span much of the THE1B sequence. Previously, two papers have identified transcription factors DUX4 (*Young et al., 2013*) and ZBTB33 (*Wang et al., 2012*) as preferentially binding particular motifs within THE1B elements. Ordering motifs by how many cell types they are active in, of the top four motif groups identified, the top motif corresponds to the DUX4 consensus binding sequence and associates DUX4 binding with the two Tx (transcription) domains, associating the occurrence of this motif with only a signal of elevated H3K36me3 (*Kundaje et al., 2015*), ubiquitously across somatic cell types. Despite this, and interestingly, this motif was NOT identified as influencing H3K4me3 in testes, nor with any impact on meiotic recombination. Similarly, the fourth motif is a match to the ZBTB33 (Kaiso) target motif, associating this motif with the occurrence of both

Tx (i.e. H3K36me3) and 'ReprPCWk'; polycomb modifications, exhibiting enrichment of the H3K27me3 histone modification. The latter modification was previously associated with ZBTB33 binding, while the former represents a distinct modification associated with the same motif. The second motif group exactly matches the motif CCGCCAT which is the consensus binding target of YY1 and in THE1B repeats shows a similar enrichment signal to the DUX4 motif. The final motif of the top four identified was precisely the motif ATCCATG, which we identified above and found to strongly reduce recombination rate where present. Across 110 categories and cell types, this motif was identified, and unlike the above motifs, showed enrichment for both the 'Het' and 'ZNF/repeats' categories. These are characterized by elevated H3K9me3, which marks 'constitutive' heterochromatin or inactive DNA with widespread methylation of CpG dinucleotides, and in the second case, by elevated H3K36me3 also, which instead marks active regions, including transcribed regions. Given this, we compared all 18 motifs associating with H3K4me3 signal strength in human *testes* at regions not bound by PRDM9 (called non-PRDM9 H3K9me3/H3K4me3 motifs above) – and which show a consistent association with meiotic recombination in the opposite direction. Remarkably, 14 of the 18 motifs coincided with 14 of the 67 motif groups, indicating that these motifs (unlike PRDM9) appear to associate with histone modifications in somatic cells. Moreover, all 14 coinciding motifs lie within the subset of 34 motif groups associating with, in at least one cell type, the same heterochromatin category as the motif ATCCATG, a highly significant enrichment ($p=2.3\times10^{-5}$ by FET). This suggests a common cause for these diverse motifs – across many different cell types, they associate with increasing heterochromatin (H3K9me3, and as described above, and below, H3K4me3), while increases in H3K9me3 accompany increases in average H3K4me3 in testes, and decreases in meiotic recombination. Indeed, although we found 33 different motif groups associating with exclusively non-heterochromatin ROADMAP cellular domains, for example transcribed regions (Tx), or the polycomb repression-like state, none of these showed an impact on either H3K4me3 in testes, or meiotic recombination, despite (for example) high testes expression of DUX4 (*Young et al., 2013*). This implies a potential causal relationship between recombination and H3K4me3/H3K9me3, rather than the other marks studied by ROADMAP, within THE1B repeats. Looking across cell types, the overlap between motifs influencing THE1B H3K4me3 in testes and the heterochromatin state varies strongly between 0 and 10. The top cell types (*Figure 4—source data 1*) in increasing overlap were the following cell lines: ES-I3 Cells, hESC Derived CD184 + Endoderm Cultured Cells, hESC Derived CD56 + Mesoderm Cultured Cells, Primary monocytes from peripheral blood, Primary hematopoietic stem cells G-CSF-mobilized Male, Fetal Intestine Small, HUES48 Cells, HUES6 Cells, iPS-20b Cells and HUES64 Cells. This list is dominated by embryonic stem cells (ESCs), their derivatives, and induced pluripotent stem cells. These cell types therefore behave most similarly to the properties we observe for both meiotic recombination, and H3K4me3 in testes. Although the genomic 'domain' annotation is informative, we further directly analyzed histone modification enrichment values for all seven core 'ROADMAP' studied motifications (*Kundaje et al., 2015*) in two of the embryonic stem cell (ESC) types showing the strongest overlap; HUES6 Cells (E014 in *Figure 4—source data 1*) and HUES64 cells (E016). Using each histone modification in turn as a phenotype, we tested jointly (using the same Poisson GLM framework as previously) for an association of the set of 18 motifs influencing meiotic H3K4me3 on that modification in the ES cells. We tested whether (i) each motif showed a significant impact in ESC cells, and (ii) for correlation in the estimated *effect size* in ES cells to that in testes H3K4me3, to examine whether there is a concordant effect across cell types. Results for both ESC types were highly concordant (*Figure 4—figure supplement 1d*). For (ii), in HUES6 cells by far the strongest correlations in estimated effect size were seen with two marks; H3K4me3, and H3K9me3, with similar very strong positive rank correlations >90% ($p<10^{-16}$). These correlations are so high that within noise, it appears many or most motifs have identical impacts across these cell types. Nominally significant negative correlations of around −0.5 were also seen for alternative histone modifications at the same residues: H3K4me1 and H3K9ac ($0.01<p<0.05$), potentially explained by their absence when the other modifications are present. 9 of the 18 motifs were significant at $p<0.05$ for H3K4me3, and remarkably 15 of 18 are significant for H3K9me3 in

HUES6 cells, all in the same direction as testes H3K4me3 (*Figure 4—figure supplement 1d*), from these fully independent data. Taken together, these results overwhelmingly imply that all, or almost all, the motifs which are responsible for elevated H3K4me3 in THE1B in testes, operate similarly or identically to elevate H3K4me3 in other tissues and cell types, particularly embryonic stem cells. Further, they are also – and considerably more strongly (*Figure 4c*) – associated with H3K9me3 elevation in the same cell types. Therefore, we describe these motifs as non-PRDM9 H3K9me3/H3K4me3 motifs to reflect this. We note that this does not directly imply these marks are *established* in ESCs and other cells and they might be inherited in these cell types from progenitors. However these non-PRDM9 influences on recombination, unlike PRDM9-induced H3K4me3, clearly operate rather widely across cell types. It is perhaps surprising that H3K4me3 and H3K9me3 should show these consistent impacts in the same directions, and across diverse motifs within THE1B repeats; such a pattern was though seen previously across human repeats (*Kundaje et al., 2015*) and so might operate more widely. Unsurprisingly given our results, across all 20696 THE1B repeats we studied, the enrichment for these two marks is highly correlated (rank correlation 61% in HUES6 cells, the highest for any pair of marks), so the same individual THE1B repeats show (often weak) enrichment for both marks, although this does not necessarily imply co-occurrence in the same individual cells. Potential causes of these histone modifications are discussed in the main text.

## Identifying motifs associated with binding of KRAB-ZNF genes, and TRIM28 recruitment, at THE1B repeats

The above approach describes a method to identify sequence motifs within all or a subset of THE1B elements influencing 0–1 hotspot status. We applied the identical approach to attempt to identify binding motifs for three KRAB-ZNF proteins enriched for PRDM9 binding (*Imbeault et al., 2017*; Michael Imbeault, personal communication): ZNF100, ZNF430 and ZNF766. For each we first identified instances of binding peaks of each protein within 500 bp of the centers of THE1B elements, and then identified motifs. We did the same for TRIM28, a protein recruited by the KRAB domains of many KRAB-ZNF proteins, and assayed in H1 human embryonic stem cells (*Imbeault et al., 2017*). In the first three cases, the identified motifs cluster and could be mapped to specific regions of THE1B, shown in *Figure 4a* and also described below. In the case of TRIM28 the signal is expected to be a superposition of binding sites of different KRAB-ZNF proteins, complicating interpretation; indeed we identified 16 motifs, mapping throughout THE1B elements. The top-scoring motifs were TCCCTGC and CCATGTA. These heavily overlapped 2 of the 4 motifs altering (and in both cases decreasing) the probability of hotspot occurrence, including the highly significant motif ATCCATG. Therefore, we conditioned on the latter motif occurring and repeated our motif-finding for the resulting subset of THE1B repeat elements, reasoning that such TRIM28 peaks might be bound by a single protein with a well-defined target motif. Indeed, this analysis revealed a set of 7 motifs, all within a contiguous region of length 57 bp, mapping to the region 181–231 of the THE1B consensus sequence. The resulting extended 'TRIM28' target motif is shown in . There is some spacing variability in the first half of this motif among bound copies because of the variable number of copies of 'CT' found in this region. This motif incorporates and links the hotspot-influencing motifs ATCCATG and CTGCACA. Moreover, it overlaps several additional motifs associated with increasing non-PRDM9 H3K9me3/H3K4me3. Finally this motif is disrupted by several motifs associated with decreasing non-PRDM9 H3K9me3/H3K4me3. These overlaps are highlighted in *Figure 4—source data 1*, which gives results for all four motifs.

We also identified two similar target motifs for binding of ZNF766 mapping to different parts of the THE1B repeat consensus. The previously unknown extended 'TRIM28' motif is therefore a recombination coldspot motif, and simultaneously a motif for TRIM28 recruitment, H3K9me3 deposition, and weaker H3K4me3 deposition, at the same locations. Moreover it appears that binding in THE1B repeats and elsewhere by each of four further zinc-finger proteins ZNF430, ZNF100, ZNF766 is recruited by other motifs for decreased recombination rates, in a manner highly dependent on the cis sequences near PRDM9 binding sites inside THE1B repeats.

## Testing for a general association between KRAB-ZNF protein binding and TRIM28 recruitment and recombination at PRDM9-bound sites

Given that binding by KRAB-ZNF genes and TRIM28 recruitment offers an explanation for the ability of particular sequence motifs in THE1B to increase H3K9me3 and H3K4me3 and yet decrease recombination rates, while not preventing PRDM9 binding, we tested if this property were more general. Across 235 recently studied KRAB-ZNF genes and TRIM28, we first identified their ChIP-seq binding sites falling within 500 bp of our PRDM9 binding sites, after excluding PRDM9 binding sites at pre-existing H3K4me3 peaks, near TSS, or overlapping DNase HS sites (where our other results show hotspots to be less likely; including these regions strengthened but did not alter the below results). We then studied those proteins with at least 30 peaks overlapping our binding sites (other proteins showed similar overall patterns though we lacked statistical power to examine them individually). We used the binary GLM framework described above to perform association testing for each protein separately between occurrence of that protein binding the genome within PRDM9 binding sites, and whether those binding sites become hotspots. We included our measured PRDM9 binding strength, and local GC-content within the PRDM9 binding site, as co-regressors. The results are shown in *Figure 4e*; the estimated effect of KRAB-ZNF binding was negative in all but one case, and significantly negative impacts of binding on recombination ($p < 0.05$) were seen for 27 proteins (TRIM28 being the most significant) examined despite the typical low overlap of individual KRAB-ZNF genes with PRDM9 binding sites. Among the genes with significant negative impacts were each of the four analyzed above that bind THE1B repeats, and where we were able to identify connections to their binding target sequences. For each protein we also tested for association with H3K9me3 in HUES-64 cells, with identical predictors. Instead of hotspot status, the response variable was now mean H3K9me3 enrichment in the 1 kb surrounding the PRDM9 binding peak center, after quantile normalization and now fitting an ordinary linear model. The resulting values were used to color *Figure 4e*. The large majority of KRAB-ZNF genes examined show positive correlations between binding and H3K9me3 placement, as expected (*Grey et al., 2017*).

