## [Decision Letter]

Thank you for submitting your article "Human PRDM9 can bind and activate promoters, and other zinc-finger proteins associate with reduced recombination in cis" for consideration by *eLife*. Your article has been favorably evaluated by Kevin Struhl (Senior Editor) and three reviewers, one of whom is a member of our Board of Reviewing Editors. The reviewers have opted to remain anonymous.

The reviewers have discussed the reviews with one another and the Reviewing Editor has drafted this decision to help you prepare a revised submission. As you will see, the reviewers all had serious concerns. If you choose to resubmit to *eLife*, these points would need to be addressed in revisions.

The biggest concern was that the manuscript, as written, is long and winding, so it is hard to follow and hard to identify the main conclusions. In particular, all three reviewers were unsure what was learned from the analysis of the chimpanzee allele. Two also felt that the dimerization analysis could be put in a different paper, but we leave that decision to you. In any case, the manuscript needs to be thoroughly rewritten to focus on the main conclusions and motivate additional analyses more clearly.

The reviewers also brought up the need for an explicit comparison to Imbeault et al.'s PRDM9 data, and a discussion of Stevison et al. 2016.

Thirdly, there were concerns about the abnormal chromosomal make-up of the cell lines and how it may affect the number of PRDM9 binding sites, as well as whether transfection affects karyotypic stability. More generally, the reviewers felt that more discussion was needed of the possible limitations of the system, when it is not diploid, homeostatic mechanisms may not be operating as in vivo, and non-HEK data sets are used extensively in the analysis.

We include the specific comments below.

*Reviewer #1:*

I thought the paper was full of novel and interesting results, notably about the use of the various ZFs, the likely roles of pre-existing histone modifications and other KRAB-ZNFs in suppressing recombination.

However, the manuscript is very long, disjointed and therefore hard to follow. A number of specific sentences are also unclear, because they are not parallel structures or it is not obvious to what noun the pronouns refer; as just one example, what does "their" refer to in the eighth paragraph of the Discussion? I'd suggest a thorough revision, perhaps more clearly mirroring the sequence of points laid out in the Discussion.

In particular, it was not obvious to me why the chimp allele was investigated (is it simply to consider another, more diverged allele?), or what was specifically learned from that analysis? The authors argue that it tells us about epigenetic predictors of binding, but I failed to understand the logic. Also, and importantly, the discussion of the chimp data should really be referencing Stevison et al. 2016 Mol Bio Evol, in particular when concluding that the association of recombination rates to binding is weaker in chimp than in humans (as Stevison et al. argue that the estimates in Auton et al. are pretty poor and that is obviously an alternative explanation worth considering).

One point I think is missing is more of a discussion of the limitations of the use of this system, and in particular this cell line. Does the reliance on a comparison of transfected and untransfected make the aneuploidy of the cell line irrelevant in the analyses? Which properties do the authors think may plausibly differ in vivo?

*Reviewer #2:*

In this paper, Simon Myers and colleagues have mapped the genome-wide binding sites of the meiosis-specific PRDM9 protein, by expressing it ectopically in human cell lines. Although PRDM9 is normally only expressed in meiotic cells, and overexpressing the protein may have some experimental caveats, this experimental approach allows the authors to exploit all the available data from the Encode data sets that have been performed in human cell lines. Endogenous PRDM9 binding maps have recently been mapped from mice testes using an antibody against PRDM9.

The main findings are the identification of new consensus binding sites, thanks to an algorithm that allows spacing flexibility between the motifs recognized by each ZF of the ZF array. It emerges using this technique that some ZF are clearly not involved in DNA recognition, and at the opposite, the first ZF participate in DNA recognition, whereas they were previously though not to.

Using the case of THE1B repeats, which are bound by PRDM9 but show different DSB/recombination frequencies depending of where they are in the genome, they identify factors that modulate recombination at PRDM9 binding sites, positively such as broad chromosomal domains with high DSB/recombination frequency, and negatively, such as certain chromatin marks like endogenous H3K4me3 and H3K9me3. They also identify a motif around these THE1 repeat binding sites negatively associated with recombination. Interestingly, this motif corresponds to a binding site for KRAB-ZNF proteins that recruit heterochromatin marks (H3K9me3).

Additional data show that PRDM9 can activate transcription at some of its binding sites, predicted to vary depending on the PRDM9 allele.

Thanks to the thorough analysis of the set of PRDM9 binding sites and the correlation with other available genomic data, this paper brings interesting findings about the factors that control PRDM9 binding and its ability to promote recombination or not.

The paper is very long and contains many data that are somewhat heterogeneous and should perhaps be included in another paper, such as the part on the chimpanzee PRDM9 binding, or on the dimerization of PRDM9 through its ZF.

I have several comments that should be addressed:

1) It would be useful somewhere in the Introduction to indicate that a molecular function for H3K4me3 in promoting recombination has been proposed based on studies in budding yeast (2013), and seems conserved in mammals having PRDM9 since the mouse H3K4me3 "reader" interacts with an axis-associated DSB protein and with PRDM9 (Imai et al. 2017).

2) Subsection “A map of direct PRDM9 binding in the human genome”, first paragraph: I may have missed it, but do the PRDM9 binding sites that coincide with a DSB site have a higher binding frequency than those that do not?

3) Subsection “A map of direct PRDM9 binding in the human genome”, first paragraph: the larger number of peaks is not necessarily due to the overexpression, but may also be due to different chromatin properties in a cell line than in meiotic cells.

4) Subsection “Multimer formation is mediated primarily by the ZF array”, last paragraph: it is not clear why the authors did not examine PRDM9 multimer formation after DNase digestion. I think this should be done, if possible.

*Reviewer #3:*

In this manuscript, Altemose et al. describe PRDM9 binding preferences and its consequences in transfected human cell culture cells (HEK293) that normally do not express PRDM9. The experiment itself is relatively straightforward (however, see below for my concerns regarding novelty and choice of the system), and the authors have conducted an impressive number of in silico analyses on their data, combined with existing data sets. Unfortunately the manuscript is overly lengthy and not particularly coherent – in its current form it comes across more as a collection of individual observations and seems to lack a common thread. Some of the findings have already been described before; see in more detail below.

Substantive concerns:

1) It is not clear how much Altemose et al.'s data add to the observations of Imbeault et al. (Nature 2017). Imbeault et al. present ChIP-exo data on 222 human KRAB zinc finger proteins, including PRDM9, in HEK293 cell, and their paper also include the observation of PRDM9 binding near transcriptional start sites, i.e. promoters. Thus, it seems that one of the main observations of Altemose et al. (see subsection “A map of direct PRDM9 binding in the human genome”) is largely replicating Imbeault et al.'s data, albeit at a lower resolution (ChIP-seq vs. ChIP-exo).

2) Results of most wet-lab experiments are not shown or not described adequately. PRDM9 expression (by western blot) in transfected vs. untransfected cells? Transfection efficiency? Reproducibility of ChIP-seq results?

3) The experimental system used (PRDM9-transfected HEK293 cells) and its biological relevance for understanding meiotic recombination distributions. See also comment 4.

a) HEK293 cells have complex karyotypes and are not diploid (modal chromosome number more than 46) – how is this taken into consideration when analyzing ChIP-seq data?

b) HEK293 cells are most likely female. Should this be taken into account?

c) HEK293 cells are likely from a fetus of Western European ancestry. Shouldn't the recombination rates of only this population be considered? (Or if that is what was done, I cannot find a mention of it anywhere.)

d) What is the likely germline PRDM9 allele of the HEK293 donor fetus, i.e. the one that this particular genome would have evolved with?

e) Statement: "Because the human B allele binds promoters". Firstly, would be more accurate to state "because the human B allele binds promoters in HEK293 cells" and, secondly, it still does not follow that the same would be true for meiotic cells.

4) Inferring histone modifications for HEK293 cells – and especially for spermatocytes – from data K562 cells (subsection “Recombination outcomes depend on genomic context”, last paragraph) is problematic. "Imperfect proxy" strikes me as an understatement. K562 are myelogenous leukemia cells. I am not familiar with all available human (ENCODE etc.) datasets but quickly searching I found that there are data from Bradley Bernstein's lab on a number of histone modifications in HUVEC cells. These and all other available data sets (also Imbeault et al. Nature 2017) should be used to evaluate how appropriate it is to infer histone modification patterns across cell types. If the answer is that it is not, then the authors need to perform their own ChIP-seq for relevant modifications in HEK293.

5) Some conclusions and statements are not supported by the data presented in this manuscript. Note that they are, as such, likely valid and/or already well-established concepts in the field in the light all available evidence to date. These conclusions and statements would be appropriate in a review- or commentary-style paper, but not in the context of a primary research paper. Examples:

"Recombination outcomes depend on genomic context";

"KRAB-ZNF binding and TRIM28 recruitment suppress recombination".

6) What does experiment with the chimp PRDM9 reference allele add beyond published work (authors own data in Figure 4 compared Schwartz et al. 2014 and Auton et al. 2012)? What do we learn from this experiment beyond what was known? I also do not follow how this rather artificial experimental set-up (chimp PRDM9 acting on the HEK genome) leads to better understanding of "the epigenetic predictors of binding".

7) To me, one of the more interesting findings was the apparent transcriptional activation seen in PRDM9-tranfected cells (subsection “PRDM9 can activate transcription of some genes, including *VCX* and *CTCFL*” and Figure 5). If this were to be emphasized more in a future version of the manuscript, the data would need to be backed up by substantially stronger wet-lab protein-level evidence. If this observation is kept as just a side note, I am not requesting additional experiments.

[Editors' note: further revisions were requested prior to acceptance, as described below.]

Thank you for submitting your article "A map of human PRDM9 binding provides evidence for novel behaviors of PRDM9 and other zinc-finger proteins in meiosis" for consideration by *eLife*. Your article has been favorably evaluated by Kevin Struhl (Senior Editor) and two reviewers, one of whom is a member of our Board of Reviewing Editors. The reviewers have opted to remain anonymous.

The reviewers have discussed the reviews with one another and the Reviewing Editor. All agreed that the revisions have addressed all the substantive points raised in the initial review and that this version is much clearer and easier to follow. The manuscript could probably still be shortened somewhat without sacrificing content, however. For example, the section about Motif 7 and PRDM9B could be shortened to two sentences (and a reference added to Baudat et al. 2010 for the observation about A vs. B). Similarly, the various steps of the THE1B analysis could probably be shortened by half, and described in more details in the Methods.

Those are only suggestions, however. The only remaining concern for publication is about availability of the methods. A number of ingenious and useful approaches are developed in the paper (notably to find motifs and call peaks) but the codes do not appear to be publicly available. All main scripts should be (e.g., in GitHub), both because it is *eLife* policy and because it is necessary for the work to be reproducible.

---

## [Author Response]

The biggest concern was that the manuscript, as written, is long and winding, so it is hard to follow and hard to identify the main conclusions. In particular, all three reviewers were unsure what was learned from the analysis of the chimpanzee allele. Two also felt that the dimerization analysis could be put in a different paper, but we leave that decision to you. In any case, the manuscript needs to be thoroughly rewritten to focus on the main conclusions and motivate additional analyses more clearly.

We thoroughly restructured and simplified the entire text and changed and re-ordered the figures accordingly. We decided to emphasize the utility of the chimp allele experiments as an important control for some of our major conclusions about the human allele, while leaving the details of the chimp binding motif as Figure 2—figure supplement 1. We have also decided to maintain the co-IP results after performing additional replicating experiments with benzonase treatment to rule out a role for co-IP mediated by large DNA fragments. We have reworded the text to emphasize that protein-protein interactions represent an important novel function of PRDM9’s zinc-finger array, which may influence its already complicated evolution (subsection “Multimer formation is mediated primarily by the ZF array”, second paragraph, and the Discussion).

The reviewers also brought up the need for an explicit comparison to Imbeault et al.'s PRDM9 data, and a discussion of Stevison et al. 2016.

We have performed an explicit comparison to Imbeault et al.’s set of 839 conservatively called ChIP-exo peaks for PRDM9 to the 170,198 peaks we identified in the Results subsection “Human PRDM9 frequently binds promoters” (second paragraph). Because we have removed the figure and section discussing chimp recombination rates, we have not included a discussion of Stevison et al. (2016).

Thirdly, there were concerns about the abnormal chromosomal make-up of the cell lines and how it may affect the number of PRDM9 binding sites, as well as whether transfection affects karyotypic stability. More generally, the reviewers felt that more discussion was needed of the possible limitations of the system, when it is not diploid, homeostatic mechanisms may not be operating as in vivo, and non-HEK data sets are used extensively in the analysis.

The point is well taken. We comment on this in the Results subsection “A map of direct PRDM9 binding in the human genome” (first paragraph), emphasizing that because our peak-calling method normalizes using local background coverage, aneuploidy will not result in false peak calls, nor misestimated values of PRDM9 binding strength. (Increased coverage along a chromosome with higher copy number will, however, increase the power to detect weaker peaks on that chromosome.) We have also noted throughout the text some of the caveats of this experimental system and now suggest follow-up work that might explore these interesting phenomenain vivo(Introduction, Results, Discussion and Materials and methods).

We include the specific comments below.Reviewer #1:[…] However, the manuscript is very long, disjointed and therefore hard to follow. A number of specific sentences are also unclear, because they are not parallel structures or it is not obvious to what noun the pronouns refer; as just one example, what does "their" refer to in the eighth paragraph of the Discussion? I'd suggest a thorough revision, perhaps more clearly mirroring the sequence of points laid out in the Discussion.

As described above, we have extensively rewritten the text for clarity and aimed to improve the narrative flow.

In particular, it was not obvious to me why the chimp allele was investigated (is it simply to consider another, more diverged allele?), or what was specifically learned from that analysis? The authors argue that it tells us about epigenetic predictors of binding, but I failed to understand the logic. Also, and importantly, the discussion of the chimp data should really be referencing Stevison et al. 2016 Mol Bio Evol, in particular when concluding that the association of recombination rates to binding is weaker in chimp than in humans (as Stevison et al. argue that the estimates in Auton et al. are pretty poor and that is obviously an alternative explanation worth considering).

As described above, the chimp allele forms an important control for several of the main findings of the paper, including promoter binding, gene activation, and heteromultimer formation after benzonase treatment, and we now emphasize this better. To shorten and simplify the text, we have largely dropped the discussion of chimp recombination, although we still note that it is interesting that the most highly specific chimp zinc fingers, as determined empirically for the first time using our data, correspond to a shared submotif present in the majority of PRDM9 alleles segregating in western chimpanzees (Schwartz et al. 2014; Discussion, fifth paragraph).

One point I think is missing is more of a discussion of the limitations of the use of this system, and in particular this cell line. Does the reliance on a comparison of transfected and untransfected make the aneuploidy of the cell line irrelevant in the analyses? Which properties do the authors think may plausibly differ in vivo?

Please see above. Briefly, comparing transfected and untransfected cells does aid robustness. In terms of limitations of the experimental system, we have been able to check binding similarities using in vivo data (for example taking advantage of available meiotic H3K4me3 data, examining overlap with recombination hotspots, etc.), to overcome potential limitations of the HEK293T system. We believe it is unlikely that general broad properties of human PRDM9 we identify, e.g. binding to a large subset of promoters, and complex binding modes, differ in vivo. Although, overall, we observe PRDM9 binding to most human hotspots, it seems probable that individual binding sites may sometimes be stronger or weaker in meiotic cells, due to factors such as differential chromatin accessibility, and we have tried to make our analyses robust to this fact.

Reviewer #2:[…] The paper is very long and contains many data that are somewhat heterogeneous and should perhaps be included in another paper, such as the part on the chimpanzee PRDM9 binding, or on the dimerization of PRDM9 through its ZF.I have several comments that should be addressed:1) It would be useful somewhere in the Introduction to indicate that a molecular function for H3K4me3 in promoting recombination has been proposed based on studies in budding yeast (2013), and seems conserved in mammals having PRDM9 since the mouse H3K4me3 "reader" interacts with an axis-associated DSB protein and with PRDM9 (Imai et al. 2017).

We now reference these points in the third paragraph of the Introduction.

2) Subsection “A map of direct PRDM9 binding in the human genome”, first paragraph: I may have missed it, but do the PRDM9 binding sites that coincide with a DSB site have a higher binding frequency than those that do not?

In Figure 1, we show that the probability of overlapping a DSB site increases with PRDM9 binding enrichment and, in Figure 1—figure supplement 2, we show that PRDM9 binding enrichment positively correlates with testis DMC1 ChIP-seq enrichment. These results imply that the full set of PRDM9 binding sites overlapping DSB sites should have a higher mean PRDM9 enrichment than those that do not.

3) Subsection “A map of direct PRDM9 binding in the human genome”, first paragraph: the larger number of peaks is not necessarily due to the overexpression, but may also be due to different chromatin properties in a cell line than in meiotic cells.

We have added this caveat (subsection “A map of direct PRDM9 binding in the human genome”, first paragraph).

4) Subsection “Multimer formation is mediated primarily by the ZF array”, last paragraph: it is not clear why the authors did not examine PRDM9 multimer formation after DNase digestion. I think this should be done, if possible.

We agreed and performed these experiments. Please see above and Figure 5—figure supplement 3.

Reviewer #3:In this manuscript, Altemose et al. describe PRDM9 binding preferences and its consequences in transfected human cell culture cells (HEK293) that normally do not express PRDM9. The experiment itself is relatively straightforward (however, see below for my concerns regarding novelty and choice of the system), and the authors have conducted an impressive number of in silico analyses on their data, combined with existing data sets. Unfortunately the manuscript is overly lengthy and not particularly coherent – in its current form it comes across more as a collection of individual observations and seems to lack a common thread. Some of the findings have already been described before; see in more detail below.Substantive concerns:1) It is not clear how much Altemose et al.'s data add to the observations of Imbeault et al. (Nature 2017). Imbeault et al. present ChIP-exo data on 222 human KRAB zinc finger proteins, including PRDM9, in HEK293 cell, and their paper also include the observation of PRDM9 binding near transcriptional start sites, i.e. promoters. Thus, it seems that one of the main observations of Altemose et al. (see subsection “A map of direct PRDM9 binding in the human genome”) is largely replicating Imbeault et al.'s data, albeit at a lower resolution (ChIP-seq vs. ChIP-exo).

In response to the reviewer’s question regarding novelty, we were initially puzzled regarding the question: after a careful reading of the text of Imbeault et al. 2017 (including supplements), we could not find any mention of the extent of PRDM9 binding at promoters. However, we did find a column of their Supplementary Table 5 (which describes overlap of several hundred KRAB-ZNF genes with many different features) corresponding to PRDM9, and describing 18% overlap with promoters for their 839 conservatively called ChIP-exo peaks for PRDM9. In the revised manuscript, we highlighted the general agreement in terms of promoter overlap of the Imbeault et al. peaks with the set of 170,198 peaks we identified.

We think our results differ from the previous findings in several ways: 1) We find many more peaks, now accounting for the majority of the tens of thousands of known human hotspots. This allows us to, e.g., understand PRDM9 binding modalities more fully than possible from a reduced set of peaks, notwithstanding potentially greater resolution for individual peaks possible with ChIP-exo (although hundreds of our peak centers are localized to <10 bp, given our deep sequencing, and we identify the likeliest bound motif within each individual peak among our 7 motifs.) 2) We formally test whether promoter binding is unusual, then describe and aim to understand it, revealing, for example, that more strongly H3K4me3-marked promoters are more likely to show detectable binding, and that PRDM9 still binds promoters at the usual motifs. 3) We show recombination hotspots do not occur at PRDM9-bound promoters. 4) We show PRDM9 can activate expression of some, but not all, promoters it binds. 5) In the revised manuscript, we analyze H3K36me3 marking by PRDM9 at promoter binding sites, showing it differs at promoters and non-promoters.

We think the Imbeault et al. data, specifically at PRDM9, provide complementary independent evidence, but cannot replicate our main results. On the other hand, the knowledge of binding properties of many other ZF-genes provided by Imbeault et al. has helped us to understand the recombination impacts of these genes.

2) Results of most wet-lab experiments are not shown or not described adequately. PRDM9 expression (by western blot) in transfected vs. untransfected cells? Transfection efficiency? Reproducibility of ChIP-seq results?

Figure 5—figure supplement 1 includes a western blot showing expression of our PRDM9 constructs in transfected vs. untransfected cells. In early experiments with a commercially available PRDM9 antibody, we detected only a faint western band in transfected cells, and no visible band whatsoever in untransfected cells; however, we could not rule out sensitivity issues with this antibody and so we proceeded to use epitope tag antibodies and have not included these data. Our RNA-seq data do show low levels of endogenous PRDM9 expression in untransfected cells, though ~3200-fold lower than the levels of our transfected PRDM9 constructs. In the revision, we have added transfection efficiency estimates in the Materials and methods (subsection “ChIP (N-terminal YFP-Human)”) – we thank the reviewer for pointing this out. We had previously included a discussion of ChIP-seq reproducibility in the Materials and methods subsection “ChIP sequencing, mapping, and filtering”. We also note that we observe and describe nearly universal (>90%) overlap between our PRDM9 binding sites and H3K4me3 sites from transfected cells, which were measured in an independent experiment (Figure 1). We performed two replicate experiments to identify PRDM9 binding sites: our peak calling method benefits from combining information from these two replicates to provide joint peak calls, which therefore show binding evidence in both replicates.

3) The experimental system used (PRDM9-transfected HEK293 cells) and its biological relevance for understanding meiotic recombination distributions. See also comment 4.a) HEK293 cells have complex karyotypes and are not diploid (modal chromosome number more than 46) – how is this taken into consideration when analyzing ChIP-seq data?

Please see responses to reviewers 1 and 2 above; we do take chromosome copy number variation into account when calling peaks, which is, as pointed out, important given the variable karyotypes among HEK293T cells.

b) HEK293 cells are most likely female. Should this be taken into account?

We have excluded the Y chromosome from our analysis. Apart from this, the sex of the cell line should not affect our analyses.

c) HEK293 cells are likely from a fetus of Western European ancestry. Shouldn't the recombination rates of only this population be considered? (Or if that is what was done, I cannot find a mention of it anywhere.)

The LD-based recombination map that we use is the HapMap CEU map (HapMap 2007). We have noted this in the main text for clarity, and apologize for the earlier omission (subsection “A map of direct PRDM9 binding in the human genome”, last paragraph).

d) What is the likely germline PRDM9 allele of the HEK293 donor fetus, i.e. the one that this particular genome would have evolved with?

We have not determined this (which can be challenging with existing short-read data), but we would note that the LD-based measure of recombination rate that we use (HapMap 2007) reflects historical recombination events in the cell line’s ancestral population. That is, regardless of the cell line’s particular PRDM9 genotype, most of its historical recombination events will have occurred in ancestors many generations ago, carrying varying PRDM9 alleles dominated

by the most common alleles in its ancestral population (A-type alleles, in this case, given the cell line’s European ancestry). The PRDM9 genotype is only relevant when studying the recombination events occurring in a single individual’s meiotic cells (as in Pratto et al. 2014).

e) Statement: "Because the human B allele binds promoters". Firstly, would be more accurate to state "because the human B allele binds promoters in HEK293 cells" and, secondly, it still does not follow that the same would be true for meiotic cells.

We have changed the text as suggested and included a caveat about cell-type differences (subsection “PRDM9-induced H3K36me3 is depleted at promoters”, first paragraph). However, given strong evidence of sequence-dependent PRDM9 binding at a sizable fraction of promoters in our cell line by motifs that appear to be well shared between the A and B alleles (which from previous work bind heavily overlapping targets), it is difficult – though intriguing! – to think of a mechanism by which PRDM9 could be entirely blocked from binding at promoters in a different cell type. However, we now mention the possibility of meiotic cells somehow “blocking” PRDM9 binding in the Discussion (fifth paragraph). We also show in the revised manuscript that the levels of H3K36me3 deposited by PRDM9 are reduced at promoter binding sites. We suggest this may explain why PRDM9-bound promoters, if they are also bound in meiosis, might fail to initiate recombination (consistent with a hypothesis put forward by Powers et al.2016 about the necessity of H3K4me3/H3K36me3 co-localization).

4) Inferring histone modifications for HEK293 cells – and especially for spermatocytes – from data K562 cells (subsection “Recombination outcomes depend on genomic context”, last paragraph) is problematic. "Imperfect proxy" strikes me as an understatement. K562 are myelogenous leukemia cells. I am not familiar with all available human (ENCODE etc.) datasets but quickly searching I found that there are data from Bradley Bernstein's lab on a number of histone modifications in HUVEC cells. These and all other available data sets (also Imbeault et al. Nature 2017) should be used to evaluate how appropriate it is to infer histone modification patterns across cell types. If the answer is that it is not, then the authors need to perform their own ChIP-seq for relevant modifications in HEK293.

The point is well taken. To simplify and shorten the manuscript and highlight what we believe are our main findings, we have removed the section utilizing K562 ENCODE data.

5) Some conclusions and statements are not supported by the data presented in this manuscript. Note that they are, as such, likely valid and/or already well-established concepts in the field in the light all available evidence to date. These conclusions and statements would be appropriate in a review- or commentary-style paper, but not in the context of a primary research paper. Examples:"Recombination outcomes depend on genomic context";"KRAB-ZNF binding and TRIM28 recruitment suppress recombination".

We have modified nearly all section headers to emphasize the particular findings of our study, including the specific examples given.

6) What does experiment with the chimp PRDM9 reference allele add beyond published work (authors own data in Figure 4 compared Schwartz et al. 2014 and Auton et al. 2012)? What do we learn from this experiment beyond what was known? I also do not follow how this rather artificial experimental set-up (chimp PRDM9 acting on the HEK genome) leads to better understanding of "the epigenetic predictors of binding".

To simplify the manuscript, we have changed our discussion of the chimp dataset to emphasize its utility as a control (see above). However, we note that we have uncovered the first direct, empirically determined PRDM9 binding motif for any chimp allele. All previous studies relied on in silico predictions from the chimp ZF-array, which we find to be only partially accurate. It is interesting that the most highly specific chimpanzee zinc fingers correspond almost exactly to a submotif independently discovered by Schwartz et al. (2014) to be found in the *predicted* binding sites of the majority of western chimpanzee PRDM9 alleles (Discussion, fifth paragraph). Moreover, it is interesting that the chimp allele does not preferentially bind promoters, and has different fine- and broad-scale binding patterns compared to the human allele. This is in contrast to broad-scale similarity in recombination rates with humans, but consistent with the fact we don’t see more PRDM9 binding in general in regions with higher broad-scale rates, even outside telomeres, than in regions with lower broad-scale rates.

7) To me, one of the more interesting findings was the apparent transcriptional activation seen in PRDM9-tranfected cells (subsection “PRDM9 can activate transcription of some genes, including VCX and CTCFL” and Figure 5). If this were to be emphasized more in a future version of the manuscript, the data would need to be backed up by substantially stronger wet-lab protein-level evidence. If this observation is kept as just a side note, I am not requesting additional experiments.

We thank the reviewer for this comment and have restructured the text to emphasize this finding more. We also emphasize that previous work by Cano-Rodriguez et al. (2016) has demonstrated that PRDM9’s SET domain can derepress a subset of genes when targeted to promoters. However, we disagree with the reviewer that protein-level evidence is required to conclude that PRDM9 activates transcription of these particular genes in this cell line. We demonstrate transcriptional activation by both RNA-seq and by confirmatory RT-qPCR in triplicate, with negative controls. Whether these transcripts go on to make protein in transfected HEK293T cells is not essential to answer the question of whether PRDM9 activates their transcription. While a positive protein-level result (seeing VCX and/or CTCFL protein) could potentially corroborate our finding, a negative result would be uninterpretable, as these particular proteins might be quickly degraded in this cell type. We have though added additional caveats regarding cell-type differences to this section (subsection “PRDM9 can activate transcription of some genes, including *VCX* and *CTCFL*”, fifth paragraph).

[Editors' note: further revisions were requested prior to acceptance, as described below.]

The reviewers have discussed the reviews with one another and the Reviewing Editor. All agreed that the revisions have addressed all the substantive points raised in the initial review and that this version is much clearer and easier to follow. The manuscript could probably still be shortened somewhat without sacrificing content, however. For example, the section about Motif 7 and PRDM9B could be shortened to two sentences (and a reference added to Baudat et al. 2010 for the observation about A vs. B). Similarly, the various steps of the THE1B analysis could probably be shortened by half, and described in more details in the Methods.Those are only suggestions, however. The only remaining concern for publication is about availability of the methods. A number of ingenious and useful approaches are developed in the paper (notably to find motifs and call peaks) but the codes do not appear to be publicly available. All main scripts should be (e.g., in GitHub), both because it is eLife policy and because it is necessary for the work to be reproducible.

Thank you for your additional comments on our manuscript. I have taken steps to shorten and combine the suggested sections where possible, and I have shortened some of the figure legends and made small (stylistic or typographical) changes to some of the figure panels. I have also annotated and uploaded all of my source code to a GitHub repository (https://github.com/altemose/PRDM9-map), including re-tested implementations of our peak-calling and motif-finding algorithms. Additionally, to maximize transparency and reproducibility, I organized and included all of my code needed to regenerate the paper’s figures from intermediate files, as well as those intermediate files (and code to produce them, where practical). The link to this repository is now included in the Data Availability section of the manuscript as well as in the Transparent Reporting Form.